# Analyzing the Power of Chain of Thought through Memorization Capabilities

**Lijia Yu[1], Xiao-Shan Gao[2, 3]\*, Lijun Zhang[1, 3, 4]**
[1]Institute of AI for Industries, Chinese Academy of Sciences
[2]SKLMS, Academy of Mathematics and Systems Science, Chinese Academy of Sciences
[3]University of Chinese Academy of Sciences
[4]Key Laboratory of System Software of Chinese Academy of Sciences

## Abstract

It has been shown that the chain of thought (CoT) can enhance the power of large language models (LLMs) to solve certain mathematical reasoning problems. However, the capacity of CoT is still not fully explored. As an important instance, the following basic question has not yet been answered: Does CoT expand the capability of transformers across all reasoning tasks? We demonstrate that reasoning with transformers is essentially a memorization problem for reasoning datasets. Thus, examining the power of CoT across all reasoning tasks amounts to analyzing the memorization capabilities of CoT transformers. In this paper, we provide a complete description of the memorization capabilities of fixed-precision transformers, with or without CoT, and give a negative answer to the aforementioned question. Precisely, we first provide necessary and sufficient conditions for fixed-precision transformers with and without CoT to memorize a finite reasoning dataset and show that these two conditions do not imply each other. Then, we provide lower and upper bounds for the number of parameters needed for transformers, with or without CoT, to memorize a finite reasoning dataset with $N$ elements, which are $\overline{\Theta}(N)$ in all cases. This implies that there exist reasoning tasks for which CoT does not enhance the reasoning power of transformers, leading to a negative answer to the aforementioned question. Finally, we present the first results on memorizing infinite reasoning datasets by CoT transformers and demonstrate that some simple infinite datasets cannot be memorized by transformers, with or without CoT.

## 1   Introduction

Transformer-based LLMs [33] are the most powerful models in natural language processing, and autoregressive transformer-based models [27, 4, 28, 14, 42] are the predominantly used forms, which can solve a huge number of tasks by turning them into a sequence generation problem.

It has been shown that CoT [36] allows LLMs to generate a step-by-step "thinking" process, thus improving the mathematical reasoning power of LLMs. Theoretical studies reached the same conclusion. Log-precision transformers without CoT can only solve problem class $TC^0$ [22], but log-precision transformers with CoT can solve any **P** problem [21]. Transformers with CoT can solve arithmetic problems and dynamic programming problems that cannot be solved by transformers without CoT [8]. It was further shown that CoT also enhances the reasoning power of constant precision transformers [16, 7].

These findings clearly illustrate the advantages of CoT in boosting the capability of transformers to simulate certain algorithms by constructing transformers step by step according to the algorithm.

---

\*Corresponding author

39th Conference on Neural Information Processing Systems (NeurIPS 2025).

However, the capability of CoT is still not fully explored. First, many problems may not have an explicit algorithm, such as the decision of whether an algebraic differential equation has a rational solution [37]. Even if algorithms exist, they may be too complicated for constructing a simulating transformer, such as symbolic integration [3]. For these types of problems, the earlier method of incrementally building transformers following the algorithm appears to be ineffective. Whether CoT enhances the capability of transformers to solve such problems is not known. Second, experimental studies indicate that CoT has limited scalability, even worse performance compared to direct prompting in planning problems [31] and pattern-based in-context learning problems [43]. So we raise the following natural and basic question:

**Question 1.** Does CoT expand the capabilities of transformers across all reasoning tasks?

Since mathematical reasoning demands exact results, a key observation of this paper is that *reasoning with LLMs is essentially a memorization problem*. Following previous work, we formulate a reasoning task as a function or an algorithm: $y = R(x)$. For a sample set $S = \{(x, y) : y = R(x)\}_{x \in D}$ of $R$ over a given input domain $D$, the *memorization problem* asks whether there exists a transformer $\mathcal{F}$ such that $y = \mathcal{F}(x)$ for every $(x, y) \in S$. In the previous work on LLMs reasoning, memorizing reasoning datasets permits the use of information from the algorithm $R$. For example, the arithmetic problems considered in [8] can be viewed as the memorization of the reasoning dataset $\mathbb{Z}_{p,n}$.

As a capacity measure of neural networks, memorization has been widely studied [10, 9, 23, 41, 32]. Recently, optimal memorization capacities of transformers have been established for both the general dataset [19] and the dataset that satisfies a separability condition [12]. However, in these works, the autoregressive transformers that can generate intermediate steps or CoT were not considered. Furthermore, the precision of their model parameters depends on the input and/or model parameters, and the more realistic case of fixed-precision parameters was not considered.

In light of the above observations on the interplay between reasoning and memorization, alongside the limitations of existing research on memorization, we can rephrase Question 1 as follows:

**Question 2.** Does CoT expand the memorization capability of transformers for all reasoning datasets?

In this paper, we give a complete description of the memorization capabilities of fixed-precision transformers with or without CoT for general datasets and give a negative answer to Questions 1 and 2. Our main conclusions and contributions are as follows:

**1. Necessary and sufficient conditions for fixed-precision transformers with and without CoT to memorize a finite reasoning dataset are given.** It is shown that these two conditions do not imply each other. We further show that by using position embedding (adding more basic symbols), transformers without CoT (with CoT) can memorize all finite datasets.

**2. Lower and upper bounds for the number of parameters needed for fixed-precision transformers with or without CoT to memorize a finite reasoning dataset $S$ are given.** For a dataset $S$ with $N$ elements, all these bounds are $\overline{\Theta}(N)$, when omitting some smaller quantities. This implies that both CoT transformer and no-CoT transformer need exactly $\overline{\Theta}(N)$ parameters to memorize certain reasoning datasets.

As a consequence of the above results, we know that although CoT can enhance performance on some tasks, there exist reasoning tasks for which CoT does not enhance the reasoning power of transformers, leading to a negative answer to Questions 1 and 2.

**3. We give the first results on memorizing infinite datasets by CoT-transformers.** We show that both transformers with or without CoT cannot memorize some simple infinite datasets and arithmetic tasks in $\mathbb{Z}_p$ cannot be memorized by any CoT-transformer with positive confidence (refer to Proposition 5.7 for precise meaning).

In conclusion, our results provide not only new theoretical insights into the capabilities of CoT but also practical guidance. The tight bound for the number of parameters of memorization transformers and the effects of the position encoding and more symbols (Section 4.3) may serve as useful guidance for practitioners. Even the negative results, such as "CoT does not help general memorization" are useful in practice: when solving difficult problems or problems with no explicit algorithms, CoT probably has no effect, which is aligned with the experimental results in [31, 43].

## 2 Related work

**Memorization.** As a capacity measure of neural networks, memorization has been widely studied. It was shown that feedforward neural networks (FNNs) with $O(N)$ parameters and various activation functions can memorize any dataset with $N$ elements [10, 29, 9, 23, 41], and $O(N)$ parameters are also necessary for memorization [30]. A neural network with $O(N^{2/3})$ parameters can memorize a dataset satisfying certain separation conditions [25]. Furthermore, $O(\sqrt{N})$ parameters are enough for memorization [32]. Since the VC dimension of FNNs with $N$ parameters and ReLU activation function is at most $O(N^2)$ [1], the result of [32] is optimal. Robust memorization networks were constructed in [15, 39], which require essentially more parameters than memorization. Sample complexities for memorization FNNs were established [40] and it was shown that memorization FNNs are not generalizable for certain data distributions [15, 40]. It has been shown that memorization can enhance the generalization ability of large models by memorizing diversified features of the data [17, 2, 6, 40].

For memorization of the general dataset using transformers, $O(N)$ parameters are also sufficient and optimal [19, 20, 11]. $O(\sqrt{N})$ parameters are also optimal for memorization with transformers for datasets satisfying certain separation conditions [12, 13].

Our results are new in that autoregressive transformers are used and the parameters have fixed precision. For instance, in order to achieve $\overline{O}(\sqrt{N})$ parameters [12], it inevitably leads to an unbounded parameter precision.

**Reasoning with CoT.** CoT is a key method to enhance the reasoning capabilities of LLMs [36]. Recent studies have explored various aspects of CoT, including its theoretical foundation and practical applications. For the theoretical foundation, transformers were shown to be Turing complete when infinite precision parameters are allowed [26]. The log-precision transformer without CoT can only solve problems within the complexity class $TC^0$ [22]. With CoT, the log-precision transformers can solve any **P** problem [21]. CoT also enhances the reasoning power of constant precision transformers [16, 7]. The mutual expression relationship between Turing machines and LLMs was further studied in [24, 5]. CoT has proven to be highly effective in solving traditional mathematical problems such as arithmetic and dynamic programming [8]. For practical applications, a series of works have further improved the inference accuracy of transformers using CoTs [34, 44, 18]. However, some works showed that CoT also has certain limitations [35, 38].

It is important to note that these theoretical findings, such as [8, 21], neither conflict with nor imply our results. They showed that CoT can enhance the ability of transformers to solve polynomial-time problems, by simulating a known algorithm. We show that there exist reasoning problems for which CoT does not increase the power of transformers, by establishing memorization capabilities.

## 3 Prerequisite

In this paper, we use $O(A)$ to mean a value not greater than $cA$ for some constant $c$, and $\overline{O}$ to mean that small quantities, such as logarithms, are omitted. We use $\Omega(A)$ to mean a value not less than $cA$ for some constant $c$, and $\overline{\Omega}$ or $\overline{O}$ to mean that small quantities are omitted.

**Data.** Let $\Gamma = \{\gamma_i\}_{i=1}^T$ be a set of basic symbols or tokens. We call the sequence $(\gamma_{i_j})_{j=1}^k$ a length-$k$ sentence and use $\Gamma^+$ to denote the set of all sentences with positive lengths. For a sentence $x$, let $\mathrm{len}(x)$ be its length and $\mathrm{typ}(x) \subset \Gamma$ be the set of all the distinct symbols in $x$. Moreover, let $\gamma_0$ be the stop symbol that does not appear in the sentence or label, which is only used to stop the transformer. In this paper, we consider the following kind of data.

**Definition 3.1.** A subset $S = \{(x_i, y_i)\}_{i \in \Delta} \subset \Gamma^+ \times \Gamma$ is called a *language* based on symbols $\Gamma$, if $y_i \neq y_j$ implies $x_i \neq x_j$.

Here is a language used throughout the paper.

*Example* 3.2. Let $\mathbf{Arith}_p$ be the **arithmetic in** $\mathbb{Z}_p$, where $p$ is a given prime number. We have $\Gamma = \{\gamma_i\}_{i=1}^T$, where $T = p + 7$, and $\gamma_i = i - 1$ for $i \in [p]$, $\gamma_{p+1}$ to $\gamma_{p+7}$ are $=, +, -, \times, /, (,)$, respectively. $(x, y)$ is in $\mathbf{Arith}_p$ if and only if $x$ is an arithmetic expression in $\mathbb{Z}_p$ and $y$ is the result

of $x$. Further, let $\mathbf{Arith}_{p,n} = \{(x,y) \in \mathbf{Arith}_p : \mathrm{len}(x) \leq n\}$ be the set of arithmetic expressions with length not greater than $n$. Clearly, $\mathbf{Arith}_p$ is an infinite set and $\mathbf{Arith}_{p,n}$ is a finite set.

*Remark* 3.3. In Sections 4 and 5, only finite languages are considered, which can clearly be described by a Turing machine. The infinite languages considered in Section 6 are also computable by a Turing machine. Thus, all languages considered in this paper can be computed by a Turing machine and, therefore, are *reasoning datasets*.

*Remark* 3.4. As a first step in our study, only single-token labels are considered in this paper. Note that the majority of theoretical work on transformer reasoning has focused on the setting with a single-token label [26, 8, 22, 21]. Also note that even the single-token label case includes many meaningful tasks, such as language recognition and other single-token classification tasks, mathematical multiple-choice questions and other multiple-choice questions, single-token mathematical computation and other single-token generation problems.

**Autoregressive Transformer.** An autoregressive transformer $\mathcal{F}$ has three parts:

**Part One: Embedding.** The transformer first embeds each basic symbol $\gamma_i$ into a vector $v_i \in \mathbb{R}^d$, where $v_i$ serves as an adjustable parameter in the transformer based on different tasks, and $d$ is called the *embedding dimension*. Then the input sentence $x = (\gamma_{i_j})_{j=1}^n \in \Gamma^+$ is embedded into a matrix with $n$-rows and $d$-columns. See appendix A for details. We will use $x$ and its matrix representation interchangeably whenever this does not lead to confusion.

*Remark* 3.5. We do not use position encoding in the embedding layer in most of our results. See Appendix A for details on the embedding layer.

**Part Two: Hidden layer.** Firstly, we define the feedforward layer and the attention layer. For an input $x \in \mathbb{R}^{n \times d}$, the feedforward layer with width $W$ is $\mathrm{FNN}(x) = \mathrm{Relu}(xE_1 \oplus b)E_2$, where $E_1 \in \mathbb{R}^{d \times W}, b \in \mathbb{R}^{1 \times W}, E_2 \in \mathbb{R}^{W \times d}$ are the parameters, and $xE_1 \oplus b$ means adding the vector $b$ to each row of $xE_1$. For an input $x \in \mathbb{R}^{n \times d}$, the attention-layer with width $W$ and head $H$ is $\mathrm{ATT}(x) = \sum_{i=1}^H \mathrm{softmax}(xQ_iK_ix^t + M)xV_i$, where $Q_i \in \mathbb{R}^{d \times W}, K_i \in \mathbb{R}^{W \times d}, V_i \in \mathbb{R}^{d \times d}$ are the parameters. $M \in \{-\infty, 0\}^{n \times n}$ is a causal mask defined as $M_{i,j} = -\infty$ if and only if $j > i$. $M$ can ensure that the $i$-th row can only attend to the $j$-th row where $j \leq i$, which is commonly used in autoregressive generation.

Let the transformer have $D$ hidden layers or have depth $D$, $x^{(i)}$ be the output of the $i$-th layer, and $x^{(0)} = x \in \mathbb{R}^{n \times d}$ the input. Then the $i$-th hidden layer of the transformer is

$$x^{(i)} = x^{(i-1)}W_{i-1} + \mathrm{ATT}_i(x^{(i-1)}) + \mathrm{FNN}_i(x^{(i-1)}W_{i-1} + \mathrm{ATT}_i(x^{(i-1)})), i = 1, \ldots, D$$

where $\mathrm{ATT}_i$ and $\mathrm{FNN}_i$ are the $i$-th feedforward and attention layers defined above, and $W_i \in \mathbb{R}^{d \times d}$ is a residual matrix. It is easy to see that $x^{(i)} \in \mathbb{R}^{n \times d}$ for all $i$. $W$ and $L$ are called the *width and depth of the transformer*, respectively.

*Remark* 3.6. The residual matrix $W_i$ cannot improve the power of the transformer, which allows the transformer to use fewer parameters to express the target. Details can be found in Appendix A.2.

**Part three: Output layer.** The output layer performs a linear transformation for the last row of the last hidden layer, that is, $\mathcal{F}(x) = x^{(D)}_{\mathrm{len}(x)}W_o + b_o \in \mathbb{R}^{1 \times (T+1)}$, where $x^{(D)}_{\mathrm{len}(x)}$ is the last row of $x^{(D)}$, $W_o \in \mathbb{R}^{d \times (T+1)}$, and $b_o \in \mathbb{R}^{1 \times (T+1)}$.

Let $j = \arg\max_{i \in [T+1]}(\mathcal{F}(x))_i$. Then, the output of $\mathcal{F}(x)$, which is written as $\widehat{\mathcal{F}}(x)$, is $\gamma_j$ when $j \leq T$; and the output of $\mathcal{F}(x)$ is $\gamma_0$ when $j = T + 1$.

**Transformers with or without CoT.** For any sentence $x$, its label can be predicted using a transformer $\mathcal{F}$ by employing two methods:

**1. No CoT Transformer.** Just like FNN, use $y = \widehat{F}(x)$ as the prediction label of $x$. For convenience, we refer to this type of transformers as *no-CoT-transformers*.

**2. CoT Transformer.** The result is obtained as follows: Let $\mathrm{cot} = ()$.

(1) Let $x'$ be the concatenation of $x$ and $\mathrm{cot}$, and let $y = \widehat{\mathcal{F}}(x')$;

(2) If $y \neq \gamma_0$, insert $y$ as the last element of $\mathrm{cot}$, and return to (1);

(3) If $y = \gamma_0$, let the last element of cot be the prediction label of $x$ and be denoted as $\widehat{F}_{\text{cot}}(x)$ which is said to be obtained with *CoT-transformers*.

*Remark* 3.7. For the CoT-transformer, we require that the transformer terminates in a finite number of steps, and at termination, CoT is not empty.

Here is an illustrative example. If we input $3 + (2 - 1) \times 5$ into a no-CoT-transformer, it directly gives the answer 8. If we input $3 + (2 - 1) \times 5$ into a CoT-transformer, it can obtain the answer step by step, such as $= 3 + 1 \times 5 = 3 + 5 = 8\gamma_0$, where $\gamma_0$ is the stop symbol.

**Definition 3.8.** Let $H_{W,D,H}^q$ ($H_{W,D,H,\text{cot}}^q$) be the hypothesis space of no-CoT (CoT) transformers with width $W$, depth $D$, head $H$, and parameter precision $q$.

**Parameter precision and the number of parameters.** The transformer is said to have precision $q \in \mathbb{Z}_+$ if every parameter of the transformer is a $q$-digit decimal real number whose absolute value is not more than $10^q$. A transformer $\mathcal{F}$ in $H_{W,D,H}^q$ ($H_{W,D,H,\text{cot}}^q$) and based on the basic symbol set $\Gamma$ contains $\text{para}(W, D, H, T) = O(dT + DH(Wd + d^2))$ parameters.

# 4 Memorization expressive ability of transformers

In this section, we give necessary and sufficient conditions for CoT- and no-CoT-transformers to memorize a finite language and give upper bounds for the number of parameters of the memorization transformer. We further compare the expressive powers of CoT- and no-CoT-transformers. Proofs are given in the Appendix.

## 4.1 Memorization using no-CoT-transformer

A no-CoT-transformer (CoT-transformer) $\mathcal{F}$ is said to be a *memorization* of a language $S$ if $\widehat{\mathcal{F}}(x) = y$ ($\widehat{\mathcal{F}}_{\text{cot}}(x) = y$) for any $(x, y) \in S$. We have the following general memorization theorem for no-CoT-transformers.

**Theorem 4.1.** *Let $S$ be a finite language of basic symbols $\Gamma = \{\gamma_i\}_{i=1}^T$, $N = |S|$, $L = \max_{(x,y) \in S}\{\text{len}(x)\}$, and $q \in \mathbb{Z}_+$. Then $S$ can be memorized by a no-CoT-transformer if and only if $(x_1, y_1), (x_2, y_2) \in S$ and $\text{typ}(x_1) = \text{typ}(x_2) = \{\gamma_k\}$ for some $k \in [T]$ imply $y_1 = y_2$.*

*Furthermore, if the above condition is satisfied, then there exists a no-CoT memorization transformer $\mathcal{F}$ for $S$ in $H_{O(T),O(NLT\lceil L^2 \ln^2(NLT)/q \rceil),O(T)}^q$ and $\mathcal{F}$ can be computed in polynomial time about $N, T, L, q$. This gives an upper bound $\text{para}(\mathcal{F}) = O(NLT^3\lceil L^2 \ln^2(NLT)/q \rceil)$ for the number of parameters needed to memorize $S$.*

Theorem 4.1 gives a necessary and sufficient condition for a language to be memorized by a no-CoT-transformer and the required transformer size. It is clear that most commonly used languages satisfy the condition in Theorem 4.1. For example, the language $\mathbf{Arith}_{p,n}$ satisfies this condition and thus can be memorized by a non-CoT-transformer. This theorem indicates that even without CoT and position embedding, transformers have the power to memorize most languages. If the sentence lengths are uniform in the language, then the condition in Theorem 4.1 is inherently met. Thus, we obtain:

**Corollary 4.2.** *If a finite language $S$ satisfies $\text{len}(x) = L$ for all $(x, y) \in S$, then there exists a no-CoT memorization transformer $\mathcal{F}$ for $S$ in $H_{O(T),O(NLT\lceil L^2 \ln^2(NLT)/q \rceil),O(T)}^q$.*

The following result gives the reason behind the condition in Theorem 4.1.

**Proposition 4.3.** *Given $x_1, x_2 \in \Gamma^+$ such that $\text{typ}(x_1) = \text{typ}(x_2) = \{\gamma_k\}$ for a certain $k \in [T]$, it follows that $\widehat{\mathcal{F}}(x_1) = \widehat{\mathcal{F}}(x_2)$ for any no-CoT-transformer $\mathcal{F}$.*

## 4.2 Memorization using CoT-transformer

In this section, the memorization capacity of CoT-transformers will be discussed. We first introduce several notations. For a sentence $x$, let $x[i]$ be the $i$-th symbol of $x$ and $x_{[n]}$ be the sentence composed of the first $n$ symbols of $x$. For example, if $x = (\gamma_1, \gamma_2, \gamma_3)$, then $x[2] = \gamma_2$ and $x_{[2]} = (\gamma_1, \gamma_2)$. For any language $S$ and $(x, y) \in S$, we define a set $S_x \subset \Gamma^+$ as follows: $z \in S_x$ if and only if

(1) $z_{[\text{len}(x)]} = x$ and $z[\text{len}(z)] = y$;

(2) $|\text{typ}(z_{[\text{len}(x)+1]})| > 1$; and

(3) for any $(x_1, y_1) \in S$, if $\text{len}(x_1) > \text{len}(x)$ and $z_{[\text{len}(x_1)]} = x_1$, then $y_1 = y$.

Furthermore, let $S_x^1 = \{z[\text{len}(x)+1] : z \in S_x\}$. We have the following result.

**Theorem 4.4.** *Let $S$ be a finite language of $T$ symbols, $N = |S|$, $L = \max_{(x,y) \in S}\{\text{len}(x)\}$, and $q \in \mathbb{Z}_+$. Then $S$ can be memorized by a CoT transformer if and only if (1): $|S_x| > 0$ for any $(x,y) \in S$ and (2): $\cap_{(x,y) \in S, \text{typ}(x) = \{\gamma_j\}} S_x^1 \neq \emptyset$ for any $j \in [T]$ satisfying $\{(x,y) \in S : \text{typ}(x) = \{\gamma_j\}\} \neq \emptyset$.*

*Furthermore, if the above condition is satisfied, then there exists a CoT memorization transformer $\mathcal{F}$ for $S$ in $H^q_{O(T), O(NL^2 T \lceil L^2 \ln^2 (NLT)/q \rceil), O(T), \text{cot}}$, which can be computed in polynomial time about $N, T, L, q$. This gives an upper bound $\text{para}(\mathcal{F}) = O(NL^2 T^3 \lceil L^2 \ln^2 (NLT)/q \rceil)$ for the number of parameters needed to memorize $S$.*

Theorem 4.4 gives a necessary and sufficient condition for a language to be memorized by a CoT transformer and estimates the required transformer size. But the necessary and sufficient condition in Theorem 4.4 is not intuitive, and we give several easy-to-check sufficient conditions below.

**Proposition 4.5.** *Let $S$ be a finite language of symbol set $\Gamma$. Then each of the following conditions is sufficient for $S$ to be memorized by a CoT-transformer.*

1. *The set of the last elements of all sentences in $S$ is a proper subset of $\Gamma$, that is, $\{x[\text{len}(x)] : (x,y) \in S\} \subsetneq \Gamma$.*

2. *All sentences in $S$ have the same length, that is, $\text{len}(x) = L$ for all $(x,y) \in S$.*

By Proposition 4.5, most commonly used languages representing algorithms satisfy this condition. For example, by Condition 1 of Proposition 4.5, the language $\mathbf{Arith}_{p,n}$ defined in Example 3.2 satisfies this condition because the four arithmetic operators '+', '-', '$\times$', '/' cannot be the last symbol of an arithmetic expression.

### 4.3 CoT and no-CoT-transformers: comparison and more results

This section will address the distinctions between the two types of transformers. In particular, we will show that their memorization powers are different.

**The memorization powers for no-CoT-transformer and CoT-transformer are different.** From Theorems 4.1 and 4.4, the conditions for languages that can be memorized by no-CoT or CoT-transformers are rather stringent. While the memorization powers of CoT- and no-CoT-transformers are both strong enough to memorize most languages, the languages that can be memorized by CoT- and no-CoT-transformers are different, as shown by Proposition 4.7. We first define a language.

*Example* 4.6. For any basic symbol set $\Gamma = \{\gamma_i\}_{i=1}^T$, we define the language of length calculation problem LCP: $(x,y) \in \text{LCP}$ if and only if $x$ is a sentence and the label of $x$ is $y = \gamma_{t(x)}$, where $t(x) = \text{len}(x) \mod T$ and mod is defined as $(i + kT) \mod T = i$ for $0 < i \leq T$ and $k \in \mathbb{Z}_+$. In addition, let $\text{LCP}_n = \{(x,y) \in \text{LCP} : \text{len}(x) \leq n\}$, $\text{LCP}_n^{=1} = \{(x,y) \in \text{LCP}_n : |\text{typ}(x)| = 1\}$, and $\text{LCP}_n^{>1} = \{(x,y) \in \text{LCP}_n : |\text{typ}(x)| > 1\}$.

This is a simple language that counts the length of sentences. We have the following result.

**Proposition 4.7.** *For any symbol set $\Gamma$ such that $|\Gamma| \geq 2$ and $n \geq 2$, and precision $q \in \mathbb{Z}_+$, we have*

*(1) $\text{LCP}_n$ cannot be memorized by any no-CoT or CoT-transformer.*

*(2) $\text{LCP}_n^{=1}$ can be memorized by a CoT-transformer, but cannot be memorized by any no-CoT-transformer.*

*(3) $\text{LCP}_n^{>1}$ can be memorized by a no-CoT-transformer, but cannot be memorized by any CoT-transformer.*

From Proposition 4.7, CoT- and no-CoT-transformers can solve different parts of $\text{LCP}_n$, which confirms their different expressive abilities and demonstrates that **using CoT can change the range**

**of languages that transformers can memorize, but it is not strictly superior to those without CoT.** But transformers with CoT or without CoT cannot memorize $\text{LCP}_n$. In fact, it is not hard for transformers to memorize $\text{LCP}_n$. From Corollaries 4.9 and 4.12 given below, (CoT) no-CoT-transformers can memorize $\text{LCP}_n$ by (adding new symbols) using position embedding.

**Position embedding is important for no-CoT-transformer, but not for CoT-transformer.** Theorem 4.1 shows that no-CoT-transformers without position embedding can memorize almost every language but cannot memorize certain special languages. This limitation arises because the no-CoT-transformer cannot completely leverage the length information, which leads to Proposition 4.3. If position encoding is added, then there will be no such limitation, as shown below.

**Proposition 4.8.** *Let $S$ be a finite language with $N$ elements and $T$ basic symbols, $L = \max_{(x,y)\in S}\{\text{len}(x)\}$, and $q \in \mathbb{Z}_+$. Then $S$ can be memorized by a no-CoT-transformer $\mathcal{F}$ in $H^q_{O(T+\ln(L)),O(NLT\lceil L^2\ln^2(NLT)/q\rceil),O(T)}$, which uses position encoding for the first $L$ positions.*

As a consequence, we have

**Corollary 4.9.** *For any given $q \in \mathbb{Z}_+$, every finite language can be memorized by a no-CoT-transformer with precision $q$ and position encoding.*

On the other hand, position encoding is not as useful for CoT-transformers, as shown below.

**Proposition 4.10.** *Let $S$ be a finite language and $|\text{typ}(x)| > 1$ for any $(x,y) \in S$. If $S$ cannot be memorized by a CoT-transformer, then it also cannot be memorized by a CoT-transformer with position encoding.*

Position embedding can help memorize sentences $x$ satisfying $|\text{typ}(x)| = 1$, which is a weakness for the no-CoT-transformer. But based on the conditions in Theorem 4.4 and Proposition 4.5, the CoT-transformer is capable of managing these types of sentences in the majority of cases; thus, position embedding holds limited significance for CoT-transformers.

*Remark* 4.11. In most results in this paper, we do not use position encoding. Results just proved show that position encoding is important for transformers without CoT, but has no effect for transformers with CoT, which increases our understanding of positional encoding. See Appendix A for more details.

**More basic symbols are important for CoT-transformer, but not for no-CoT-transformer.** By condition 1 of Proposition 4.5, adding a few new symbols to $\Gamma$ enables CoT-transformers to memorize all finite languages, as illustrated below.

**Corollary 4.12.** *A finite language $S$ can be memorized by a CoT-transformer if $\cup_{(x,y)\in S}\text{typ}(x)$ is a proper subset of $\Gamma$.*

Corollary 4.12 is not applicable to non-CoT-transformers, highlighting the benefit of CoT's ability to exploit the basic symbols entirely. We will give additional clarification on this phenomenon. Let $S$ be a language based on $\Gamma = \{\gamma_i\}_{i=1}^{N+M}$ and $\text{typ}(x) \subset \{\gamma_i\}_{i=1}^{N}$ for any $(x,y) \in S$. Then when we classify $S$ by a no-CoT-transformer, the symbols $\{\gamma_i\}_{i=N+1}^{M+N}$ are useless; but when we classify $S$ by a CoT-transformer, $\{\gamma_i\}_{i=N+1}^{M+N}$ are useful because they can appear in the CoT. For example, in mathematical proofs, logical symbols like "$\cdot$", "$\because$", "$\rightarrow$" are often used in the proof, but are not commonly used in the problem description. So, if we do not use CoTs, such logical symbols are useless for generating proofs; but if we use the CoTs, these symbols are useful in generating the proofs, so adding more symbols to the basic symbols can better help the transformer generate a CoT.

## 5 Necessary conditions for memorization with transformers

In this section, we give some necessary conditions for memorization with CoT- and no-CoT-transformers, and show that, from the perspective of necessary conditions, CoT may not provide particularly significant assistance in some situations.

## 5.1 $\overline{O}(N)$ parameters are necessary and sufficient for memorization

In Section 4, we show that $\overline{O}(N)$ parameters are sufficient for both no-CoT-transformer and CoT-transformer to memorize any language with $N$ elements. In this section, we will show that $\overline{O}(N)$ parameters are also necessary for both kinds of transformers to memorize some languages.

**Theorem 5.1.** *For any $q, N, T \geq 3$ and basic symbols $\{\gamma\}_{i=1}^{T}$, there exists a finite language $S$ that satisfies the condition in Theorem 4.1 (Theorem 4.4) and $\max_{(x,y) \in S} \operatorname{len}(x) \leq O(\ln(N))$, such that for any given $W, D, H$, if $\operatorname{para}(W, D, H, T) < O(\frac{N \ln T}{q})$, then $S$ cannot be memorized by any transformer $\mathcal{F} \in H_{W,D,H}^{q}(\mathcal{F} \in H_{W,D,H,\mathrm{cot}}^{q})$.*

This theorem establishes a lower bound for the number of parameters of a transformer to memorize finite languages, even if $T, L \ll N, \overline{\Omega}(N)$ parameters are required for some languages, and the lower bounds for two kinds of transformers are essentially the same, which implies CoT cannot effectively reduce the number of parameters required for memorization for some languages.

*Remark* 5.2. From Theorems 4.1, 4.4, and 5.1, we see that there exists a gap between the lower bound and the upper bound for the number of parameters for memorization transformers. But when $q, T$ are constants and $N \gg L$, as shown in the Theorems 5.1, we can only consider $N$ as in most existing works. **Then $\overline{\Omega}(N)$ parameters are necessary and sufficient for both CoT- and no-CoT-transformers to memorize a language of size $N$, giving the optimal memorization capacity for both CoT- and no-CoT-transformers.** Note that in most cases, we have $N \gg L$, and in the actual situation, $q$ and $T$ are always constants, so the above discussion is meaningful.

## 5.2 The length of sentences affects memorization

In this section, we will show how the length of sentences affects memorization. Firstly, in Theorems 4.1 and 4.4, the memorization transformer depends on the sentence length $L$, which is due to the limitation on the parameter precision (i.e. $q \in \mathbb{Z}_{+}$). Without limitation (i.e. $q = \infty$) on the parameter precision, the structure of the transformer does not need to depend on $L$, as shown below; the proofs are given in the Appendix F.2.1 for no-CoT-transformer and Appendix F.2.2 for CoT-transformer.

**Proposition 5.3.** *Let $S$ be a finite language with $N$ elements and for $T$ basic symbols, which satisfies the condition in Theorem 4.1 (Theorem 4.4). Then there exists a no-CoT-transformer (CoT-transformer) $\mathcal{F} \in H_{O(T),O(N),O(T)}^{\infty}$ ($\mathcal{F} \in H_{O(T),O(N^2),O(T),\mathrm{cot}}^{\infty}$) which can memorize $S$.*

But when the precision is limited, the increase in length will bring more difficulties to memorization for transformers, and CoT cannot help to eliminate this difficulty. We can show that if the precision of transformers is limited, the number of parameters of the memorization transformer must depend on the length for some language, as shown below.

**Theorem 5.4.** *For any $P \in \mathbb{Z}_{+}$ and precision $q \in \mathbb{Z}_{+}$, there exists a $n \in \mathbb{Z}_{+}$, a basic symbol set $\Gamma$ such that $|\Gamma| \leq 5$ and a sub-language $S \subset \mathrm{LCP}_n$ with $|S| \leq 10$, such that $S$ satisfies the condition in Theorem 4.1 (Theorem 4.4), but $S$ cannot be memorized by any $\mathcal{F} \in H_{P,P,P}^{q}$ ($\mathcal{F} \in H_{P,P,P,\mathrm{cot}}^{q}$).*

The above theorem shows that as the length $n$ increases, any fixed structure transformer is not sufficient to memorize certain languages which just contain $O(1)$ sentences and $O(1)$ basic symbols. Although we do not know how to accurately calculate the dependence of parameters on sentence length, the above theorem actually implies that both types of transformers will face difficulties with languages with unbounded sentence lengths.

## 6 Memorization of infinite language is hard

In the preceding sections, we only considered finite languages. This section will explore the challenges transformers face in memorizing infinite languages, illustrated through two specific languages. If a transformer can memorize an infinite language like $\mathbf{Arith}_p$, then we can say that transformers truly have the ability to simulate an algorithm. For the expressive power of CoT-transformers, all results are for finite languages. For instance, $\mathbf{Arith}_{p,n}$ is considered in [8] and input with finite length is considered in [26, 21].

In this section, we discuss **whether a transformer can memorize infinite languages**, which is an important open problem, and we provide some negative results related to this problem. We

demonstrate that neither the no-CoT-transformer nor the CoT-transformer is able to memorize certain basic infinite languages, suggesting that CoT might not genuinely aid transformers in resolving these issues.

**Memorizing** LCP **with transformer.** By Theorem 4.1(Theorem 4.4), for any $S \subset \mathrm{LCP}_n$ that satisfies the condition in Theorem 4.1 (Theorem 4.4), there exists a no-CoT-transformer (CoT-transformer) that memorizes $S$. But for the infinite language LCP, this is not true, as shown below, which is a corollary of Theorem 5.4.

**Proposition 6.1.** *There exists a basic symbol set $\Gamma$ and an infinite sub-language $S$ of* LCP *based on $\Gamma$, such that $S$ satisfies the condition in Theorem 4.1 (Theorem 4.4), but $S$ cannot be memorized by any no-CoT-transformer (CoT-transformer) with any precision $q \in \mathbb{Z}_+ \cup \infty$.*

This proposition shows that both types of transformers, even without precision limitations, cannot memorize certain simple infinite languages, such as length counting.

**Memorizing** $\mathbf{Arith}_p$ **with transformer.** Since $\mathbf{Arith}_p$ is a very basic computation problem or algorithm, it is an interesting open problem to show whether there exists a CoT-transformer that can memorize $\mathbf{Arith}_p$. We will prove a negative answer to this problem under certain conditions.

We first define how to use transformers to solve problems in $\mathbf{Arith}_{p,n}$ and $\mathbf{Arith}_p$. We say that a no-CoT-transformer $\mathcal{F}$ *can solve* $\mathbf{Arith}_{p,n}$ *($\mathbf{Arith}_p$)* if $\mathcal{F}$ can memorize $\mathbf{Arith}_{p,n}$ ($\mathbf{Arith}_p$). We say that a CoT-transformer $\mathcal{F}$ *can solve* $\mathbf{Arith}_{p,n}$ *($\mathbf{Arith}_p$)* if $\mathcal{F}$ can memorize $\mathbf{Arith}_{p,n}$ ($\mathbf{Arith}_p$), and for any $(x, y) \in \mathbf{Arith}_{p,n}(\mathbf{Arith}_p)$, $\mathcal{F}(x)$ outputs a CoT as follows: $x = x_1 = \cdots = x_M = y\gamma_0$, where $x_i$ is a sentence in $\mathbf{Arith}_{p,n}(\mathbf{Arith}_p)$ obtained from $x_{i-1}$ ($x = x_0$) by performing several accurate arithmetic computations. We introduce a notion below.

**Definition 6.2.** For a transformer $\mathcal{F}$ and a sentence $x$, define the *confidence of $\mathcal{F}(x)$* to be $\mathcal{F}_i(x) - \max_{j \neq i} F_j(x)$ where $i = \arg\max_j F_j(x)$. We say that a no-CoT-transformer $\mathcal{F}$ can solve $\mathbf{Arith}_{p,n}$ ($\mathbf{Arith}_p$) with confidence $c \in \mathbb{R}_+$, if the confidence of $\mathcal{F}(x)$ is not smaller than $c$ for all $(x, y) \in \mathbf{Arith}_{p,n}(\mathbf{Arith}_p)$. We say that a CoT-transformer $\mathcal{F}$ can solve $\mathbf{Arith}_{p,n}(\mathbf{Arith}_p)$ with confidence $c$, if the confidence of each step in CoT is not smaller than $c$ for all $(x, y) \in \mathbf{Arith}_{p,n}(\mathbf{Arith}_p)$.

We explain the motivation behind the notion. In real computation on a computer, it is impossible to achieve arbitrary precision. Therefore, to ensure that $\mathcal{F}$ produces an accurate output for the input $x$ with positive certainty (the confidence level of the correct label should exceed the confidence level of the incorrect labels), the confidence of $\mathcal{F}(x)$ must exceed a specific constant $c$. We will show that, under such a confidence assumption, the transformer cannot solve $\mathbf{Arith}_p$.

**Proposition 6.3.** *For any $c > 0$ and precision $q \in \mathbb{Z}_+ \cup \infty$,*

*(1) there exists a no-CoT-transformer or a CoT-transformer with precision $q$ that can solve $\mathbf{Arith}_{p,n}$ with confidence $c$ for any fixed $n \in \mathbb{Z}_+$;*

*(2) there does not exist a no-CoT-transformer or a CoT-transformer with precision $q$ that can solve $\mathbf{Arith}_p$ with confidence $c$.*

This theorem reveals that, despite the absence of precision constraints for transformer parameters, under the realistic assumption of positive confidence, it is not feasible to tackle infinitely long arithmetic problems. However, finite-length arithmetic problems can be addressed effectively. This conclusion highlights the challenges that transformers face when tackling entire arithmetic problems. In fact, we propose the following conjecture.

*Conjecture* 6.4. $\mathbf{Arith}_p$ cannot be memorized by any fixed-precision CoT- or no-CoT-transformers.

*Remark* 6.5. In [26, 21], transformers have been shown to be Turing complete in the sense that a transformer can simulate a Turing machine with finite inputs, and a set of transformers can simulate an entire Turing machine. The main technical difficulty in limiting finite inputs is that the input length is used to construct the position encoding of the transformer. Conjecture 6.4 implies that a single transformer cannot simulate a Turing machine with an infinite number of inputs and thus is not Turing complete in the rigorous sense.

# 7 Conclusion

In this work, from a memorization-capacity perspective, we theoretically analyze the impact of CoT on autoregressive transformers. We have found and proven the necessary and sufficient conditions for a finite language to be memorized by the CoT-transformer or no-CoT-transformer, and estimated the number of parameters required for memorization from the perspective of upper and lower bounds. Thus, the relationship between the CoT-transformer and the no-CoT-transformer was thoroughly studied. Specifically, the languages that can be memorized by the no-CoT-transformer and CoT-transformer are different, and the no-CoT-transformer requires position embedding to enhance its expressive power, while the CoT-transformer needs only more basic symbols to enhance its expressive power. Finally, we proved that CoT cannot help the transformer solve some problems from the perspective of necessity, such as memorizing infinite languages. Although no experiments are provided, our results have some practical implications that are explained in the last paragraph of Section 1.

**Limitation and Future Work.** This paper only analyzes the power of CoT in memorization for data with one token as the label and ignores LayerNorm to simplify the model for analysis. Positional encoding is not considered in most results. Refer to Remark 4.11 for a detailed explanation of this issue. Finally, several key aspects of the power of CoT are still not well understood, including those connected to Conjecture 6.4.

## Acknowledgments

This paper is supported by the Strategic Priority Research Program of CAS Grant XDA0480502, a Huawei Grant TC20251015013, NSFC Grants 12288201 and 92270001, and the Robotic AI-Scientist Platform of Chinese Academy of Sciences, CAS Project for Young Scientists in Basic Research Grant YSBR-040, ISCAS New Cultivation Project ISCAS-PYFX-202201, and ISCAS Basic Research ISCAS-JCZD-202302. The authors thank anonymous referees for their valuable comments.

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

# A    Details about the structure of the transformer

## A.1    Position Encoding

**Embedding matrix.** For a sentence $x$, the embedding matrix of $x$ is $x_e = (v_{i_1}, v_{i_2}, \ldots, v_{i_n})^\tau$, which will be the input to the first hidden layer in the transformer. We call $n$ the *input length*. Unlike feedforward neural networks, the input length of the transformer does not need to be fixed, and a transformer can handle input sentences with any length.

**About Position Encoding.** With position encoding, the embedding of $x$ is $(v_{i_1} + E_1, v_{i_2} + E_2, \ldots, v_{i_n} + E_n)^\tau$, where $E_i$ is the position encoding for the $i$-th position. The position encoding provides location information. However, the use of position encoding may limit the length of the language that the transformer can process. In this paper, we aim for the transformer to handle languages of arbitrary length and do not limit the length of CoT, so we do not employ position encoding. For example, if we see position encoding as adjustable parameters, then encoding the first $N$ positions only will make the transformer able to process input with a length of no more than $N$. If we take position encoding as $n$ for the $n$-th position, then, at the later positions, the position encoding may exceed the precision limitation.

## A.2    The residual layer

In the definition of transformers, we use a transition matrix $W_{i-1}$ in the residual layer. In some other works such as [8], $W_{i-1} = I$ is the identity matrix.

In fact, the functions that the transformer can express under these two definitions are the same, as shown below. Firstly, we name the transformer defined by $W_{i-1} = I$ in each residual layer as $\mathcal{F}_I$. Then we have the following result.

**Proposition A.1.** *For any given basic symbol set $\Gamma$, we have*

*(1) For any transformer $\mathcal{F}_I$, there exists a transformer $\mathcal{F}$ with transition matrices such that $\mathcal{F}(x) = \mathcal{F}_I(x)$ for any $x \in \Gamma^+$.*

*(2) For any transformer $\mathcal{F}$ with transition matrices, there exists a transformer $\mathcal{F}_I$ such that $\mathcal{F}_I(x) = \mathcal{F}(x)$ for any $x \in \Gamma^+$, and $\mathrm{width}(\mathcal{F}_I) \leq 2\mathrm{width}(\mathcal{F})$, $\mathrm{depth}(\mathcal{F}_I) \leq 4\mathrm{depth}(\mathcal{F})$ and $\mathrm{head}(\mathcal{F}_I) \leq \mathrm{head}(\mathcal{F})$.*

*Proof.* (1) in the proposition is apparent: we just need to take the whole transition matrix in the residual layer of $\mathcal{F}$ as $I$, and other parameters are the same as $\mathcal{F}_I$.

We now prove (2). We will show the following at first: for any hidden layer in $\mathcal{F}$, it can be expressed by a combination of four hidden layers whose transition matrices in the residual layer are $I$.

Let a hidden layer $L$ of $\mathcal{F}$ with width $m$ and head $H$ be written as $L(x) = xW + \mathrm{ATT}(x) + \mathrm{FNN}(xW + \mathrm{ATT}(x))$, where $\mathrm{ATT}(x) = \sum_{i=1}^{H} \mathrm{softmax}(xQ_iK_ix^t + M)xV_i$ and $\mathrm{FNN}(x) = \mathrm{Relu}(xw_1 \oplus b)w_2$.

Then we define four layers with width not more than $2m$ and head $H$ whose transition matrix in the residual layer is $I$ as follows:

The first layer $L_0$ also uses $x \in \mathbb{R}^{n \times m}$ to obtain $(x, 0) \in \mathbb{R}^{n \times 2m}$, which is similar to that in the proof in Lemma B.2.

The second layer is $L_1(t) = t + \mathrm{Relu}(t(W^T, -W^T)^T)V$ where $V_{j,m+j} = 1, V_{m+j,m+j} = -1$ for all $j \in [m]$ and the other weights are 0.

The third layer is $L_2(t) = t + \mathrm{ATT}_2(t) + \mathrm{FNN}_2(t + \mathrm{ATT}_2(t))$.

The attention layer is calculated as $\mathrm{ATT}_2(t) = \sum_{i=1}^{H} \mathrm{softmax}(tQ_i'K_i't^T + M)tV_i'$, where $Q_i' \in \mathbb{R}^{2m \times 2m}$ is defined as: the $(j, k)$-th weight is the same as the $(j, k)$-th weight of $Q_i$, where $j, k \in [m]$, and other weights are 0. $K_i' \in \mathbb{R}^{2m \times 2m}$ is defined as: the $(j, k)$-th weight is the same as the $(j, k)$-th weight of $K_i$ where $j, k \in [m]$, and other weights are 0. $V_i' \in \mathbb{R}^{2m \times 2m}$ is defined as: the $(j, k+m)$-th weight is the same as the $(j, k)$-th weight of $V_i$, where $j, k \in [m]$, and other weights are 0.

The FNN layer is calculated as $\mathrm{FNN}_2(t) = \mathrm{Relu}(tw_1' \oplus b')w_2'$. $w_1', w_2'$ are defined as: the $(i+m, j+m)$-th weight is the same as the $(i, j)$-th weight of $w_k$, where $i, j \in [m]$, and other weights are 0. $b'$

is defined as: the $(j + m)$-th weight is the same as the $j$-th weight of $b$ but other weights are 0, where $j \in [m]$.

Then we have that, for any $x \in \mathbb{R}^{n \times m}$, we can obtain $x' = (x, 0) \in \mathbb{R}^{n \times 2m}$ by the first layer $L_0$. Then by the definition of $L_1$, we can calculate: $L_1(x') = (x, xW)$. Hence in the $L_2$, it holds $\text{ATT}_2((x, xW))) = \sum_{i=1}^{H} \text{softmax}(xQ_iK_ix^T + M) \cdot (0, xV_i) = (0, \text{ATT}(x))$. In the FNN, we have $\text{FNN}_2((x, xW) + \text{ATT}_2((x, xW))) = \text{FNN}_2((x, xW + \text{ATT}(x))) = (0, \text{FNN}(xW + \text{ATT}(x)))$. So we get $L_2(x') = (x, xw + \text{ATT}(x) + \text{FNN}(xw + \text{ATT}(x))$.

Let the last layer $L_l$ use $(x, xw + \text{ATT}(x) + \text{FNN}(xw + \text{ATT}(x))$ to obtain $xw + \text{ATT}(x) + \text{FNN}(xw + \text{ATT}(x))$.

So $L(x) = xW + \text{ATT}(x) + \text{FNN}(xW + \text{ATT}(x))$ is equal to $L_l(L_2(L_1(L_0(x)))) = L_l(L_2(L_1((x, 0))) = L_l(L_2(L_1((x, xW)))) = xW + \text{ATT}(x) + \text{FNN}(xW + \text{ATT}(x))$. And it is easy to see that the width, heads of $L_0, L_1, L_2$ and $L_l$ are not more than $2m$ and $H$, respectively.

So by the above result, for an $\mathcal{F}$ with width $W_1$, depth $D$ and head $H$, for each hidden layer in the transformer, we can construct four hidden layers as above. Then use such layers and the output layer which is the same as that in $\mathcal{F}$ to form a transformer $\mathcal{F}_I$, which has width $2W_1$, depth $4D$, head $H$, and $\mathcal{F}_I(x) = F(x)$ for any $x \in \Gamma^+$. So we prove the result. $\qquad\square$

## B  Preliminary results

We give two lemmas that will be used in the subsequent proofs.

**Lemma B.1.** *For any sentences $x$ and $z$, if the first $n$ symbols in $x$ and $z$ are the same, then for any transformer $\mathcal{F}$ and $j \in \mathbb{Z}_+$, the first $n$ rows in the outputs of the $j$-th hidden layers of $\mathcal{F}(x)$ and $\mathcal{F}(z)$ are the same.*

*Proof.* Firstly, we prove that for the $j$-th hidden layers $\mathcal{F}^j$ of $\mathcal{F}$, if $x_1$ and $z_1$ satisfy that the first $n$ rows in $x_1$ and $z_1$ are the same, then the first rows $n$ of $\mathcal{F}^j(x_1)$ and $\mathcal{F}^j(z_1)$ are the same.

Assume that $\mathcal{F}^j$ can be written as: $\mathcal{F}^j(x) = xw + \sum_{k=1}^{H} \text{softmax}(xQ_kV_kx^T + M)xK_k + \text{FNN}(xw + \sum_{k=1}^{H} \text{softmax}(xQ_kV_kx^T + M)xK_k)$.

In the residual layer, the calculations in this layer do not include the interactions between different rows, just do the same transformation to each row. So when $x_1$ and $z_1$ satisfy that the first $n$ rows in $x_1$ and $z_1$ are the same, $x_1w$ and $z_1w$ also satisfy that the first $n$ symbols are the same.

In the attention layer, from the definition of $M$, for any $i \in \mathbb{Z}_+$, the $i$-th row of $\text{softmax}(xQ_kV_kx^T + M)$ can be written as $(x_iQ_kV_kx_1^T, x_iQ_kV_kx_2^T, x_iQ_kV_kx_3^T, \ldots, x_iQ_kV_kx_i^T, 0, 0, \ldots, 0)$, where $x_i$ is the $i$-th row of $x$. So it is easy to see that, when $x_1$ and $z_1$ satisfy that the first $n$ rows in $x_1$ and $z_1$ are the same, the first $n$ rows of $\text{softmax}(x_1Q_kV_kx_1^T + M)x_1$ and $\text{softmax}(z_1Q_kV_kz_1^T + M)z_1$ are the same. Hence, because the other parts in the attention layer do not include the interactions between different rows, we have that the whole output of the attention also satisfies that the first $n$ symbols are the same when input $x_1$ and $z_1$.

By the above result, we find that the first $n$ rows of $x_1w + \sum_{k=1}^{H} \text{softmax}(x_1Q_kV_kx_1^T + M)x_1K_k$ and $z_1w + \sum_{k=1}^{H} \text{softmax}(z_1Q_kV_kz_1^T + M)z_1K_k$ are the same. Similarly to the residual layer, we know that the first $n$ rows of $\text{FNN}(x_1w + \sum_{k=1}^{H} \text{softmax}(x_1Q_kV_kx_1^T + M)x_1K_k)$ and $\text{FNN}(z_1w + \sum_{k=1}^{H} \text{softmax}(z_1Q_kV_kz_1^T + M)z_1K_k)$ are the same. Adding them, we can obtain the result.

Now, we can prove the lemma. We prove the result for the first layer. If the first $n$ symbols in $x$ and $z$ are the same, then the first $n$ rows of their embedding matrix are also the same. According to the above result, we see that the first $n$ rows in the output of the first hidden layer are the same for input $x$ and $z$.

If the first $n$ rows in the output of the $i$-th hidden layer are the same, according to the above result and the fact that the output of the $i$-th hidden layer is the input of the $(i + 1)$-th hidden layer, then the first $n$ rows in the output of the $(i + 1)$-th hidden layer are also the same. When input $x$ and $z$ to the transformer, we have proved that the result is valid for $i = 1$, so the result is valid for any $i$. We prove the lemma. $\qquad\square$

**Lemma B.2.** *Let $\mathcal{F} : \mathbb{R}^{1 \times n} \to \mathbb{R}^{1 \times m}$ be an FNN network with depth $D$ and width $W$. Then we have*

*(1) If $\mathcal{F}$ has no output layer, then we can find a transformer $\mathcal{F}_1$ without an output layer such that for any $k \in \mathbb{Z}_+$ and $x \in \mathbb{R}^{k \times n}$, it holds $\mathcal{F}_1(x) = (\mathcal{F}^T(x_1), \mathcal{F}^T(x_2), \ldots, \mathcal{F}^T(x_k))^T$, where $x_i$ is the i-th row of $x$. Hence, $\mathcal{F}_1$ has width $O(W)$ and depth $O(D)$, and has the same precision as $\mathcal{F}$.*

*(2) We can find a transformer $\mathcal{F}_1$ such that for any $k \in \mathbb{Z}_+$ and $x \in \mathbb{R}^{k \times n}$, it holds $\mathcal{F}_1(x) = \mathcal{F}(x_k)$, where $x_k$ is the last row of $x$. Hence, $\mathcal{F}_1$ has width $O(W)$ and depth $O(D)$, and has the same precision as $\mathcal{F}$.*

*Proof.* It is easy to see that (2) can be obtained by (1), so we just need to prove (1). To prove (1), we just need to show how to simulate an FNN layer by the transformer layer.

Let an FNN layer be written as $\mathrm{Relu}(xW+B) : \mathbb{R}^{1 \times n} \to \mathbb{R}^{1 \times m}$, where $W \in \mathbb{R}^{n \times m}$ and $B \in \mathbb{R}^{1 \times m}$. Then we can use three transformer layers with width $m + n$ to simulate it, which directly proves the lemma.

**The first layer $\mathcal{F}^1$.** Use $x \in \mathbb{R}^{k \times n}$ to calculate the $(x, 0) \in \mathbb{R}^{k \times (n+m)}$. Just need to take the residual layer in the transformer layer as $xw$, where $w = (I, 0) \in \mathbb{R}^{n \times (n+m)}$, and $I$ is the identity matrix. Other parameters in the attention layer and the FNN layer are all zero.

**The second layer $\mathcal{F}^2$.** Let the residual layer be $x$ and the parameters in the attention layer be all 0. Then the second layer can be written as: $x + \mathrm{FNN}(x) = x + \mathrm{Relu}(xw_1 \oplus b)w_2$.

Then, we define $w_1 \in \mathbb{R}^{(n+m) \times (n+m)}$ as: the last $m$ columns are $w_1' = (W^T, 0)^T$ and the first $n$ columns are 0. We define $b \in 1 \times (n + m)$ as: the last $m$ columns equal to $B$ and the first $n$ columns are 0. Then $w_2 \in \mathbb{R}^{(n+m) \times (n+m)}$ is an identity matrix.

It is easy to check that the $k$-th row of $\mathrm{Relu}(\mathcal{F}^1(x)w_1 \oplus b)w_2$ is $(0, \mathrm{Relu}(x_kW + B))$, where $x_k$ is the $k$-th row of $x$ and $k \in [n]$. So the $k$-th row of $\mathcal{F}^2(\mathcal{F}^1(x))$ is $(x_k, 0) + (0, F(x_k)) = (x_k, F(x_k))$.

**The third layer $\mathcal{F}^3$.** This layer satisfies $\mathcal{F}^3((x_k, F(x_k))) = F(x_k)$ for each $k \in [n]$. We just need to take the residual layer as $xw$ where $w = (0, I)^T \in \mathbb{R}^{(n+m) \times m}$ and other parameters are 0.

We prove the lemma. □

# C Proofs of results in Section 4.1

In this section, we will give proofs for Theorem 4.1 and Proposition 4.3.

## C.1 Position value

For a given sequence $s$ consisting of $a$ and $b$, such as $s = (a, a, b, a, b)$, let $|s|$ be the length of $s$. We define the position-$a$ value of $s$ as $V_a^\alpha(s, k)$ where $\alpha \in \mathbb{Z}_+$ and $k \leq |s|$, which is calculated as (similar for the position-$b$ value of $s$ as $V_b^\alpha(s, k)$):

(1) Let $n_a(s, k) = \sum_{i=1}^k I(s_i = a)$ be the number of $a$ in the first $k$ elements of $s$. Similarly, let $n_b(s, k) = \sum_{i=1}^k I(s_i = b)$, where $s_i$ is the $i$-th element of $s$.

(2) Let $q_a^\alpha(s, k) = \frac{n_a(s,k)}{e^\alpha n_b(s,k) + n_a(s,k)}$.

(3) Let $V_a^\alpha(s, k) = \frac{1}{k} \sum_{i=1}^k q_a^\alpha(s, i)$.

We call $V_a^\alpha$ the position value. It is easy to see that if $|s_1| = |s_2| = n$, then $V_a^\alpha(s_1, k) = V_a^\alpha(s_2, k)$ for any $k \in [n]$ if and only if $s_1 = s_2$. So the position value can be used to distinguish the sequences with the same length.

For the $s_1$ and $s_2$ with the different lengths, the position value can be used to distinguish them. We have the following lemma.

**Lemma C.1.** *Let $\alpha \in \mathbb{Z}_+$ and $s_1$, $s_2$ be sequences satisfying $|s_1| \neq |s_2|$. If $V_a^\alpha(s_1, |s_1|), V_a^\alpha(s_2, |s_2|)$ are not 0 and 1, then $V_a^\alpha(s_1, |s_1|) \neq V_a^\alpha(s_2, |s_2|)$.*

*Proof.* Without loss of generality, we assume that $|s_1| < |s_2|$. Assume that $V_a^\alpha(s_1, |s_1|)$ and $V_a^\alpha(s_2, |s_2|)$ are not 0 and 1, and $V_a^\alpha(s_1, |s_1|) = V_a^\alpha(s_2, |s_2|)$. We derive contradictions to prove the lemma in three steps.

**Step one:** We define two rational polynomials $\mathcal{F}_{a,s_1}(x) = \frac{1}{|s_1|} \sum_{k=1}^{|s_1|} \frac{n_a(s_1,k)}{xn_b(s_1,k)+n_a(s_1,k)}$ and $\mathcal{F}_{a,s_2}(x) = \frac{1}{|s_2|} \sum_{k=1}^{|s_2|} \frac{n_a(s_2,k)}{xn_b(s_2,k)+n_a(s_2,k)}$, then we have the following result:

$\mathcal{F}_{a,s_1}(x) = F_{a,s_2}(x)$ for any $x \in \mathbb{R}$.

This is because $\mathcal{F}_{a,s_1}(e^\alpha) = V_a^\alpha(s_1) = V_a^\alpha(s_2) = F_{a,s_2}(e^\alpha)$, and considering that $e^\alpha$ is not an algebraic number and $\mathcal{F}_{a,s_i}(x)$ is a rational polynomial whose coefficients are rational numbers. We thus have $\mathcal{F}_{a,s_1}(x) = F_{a,s_2}(x)$ for all $x \in \mathbb{R}$.

**Step two:** Assume that $P \leq |s_2|$ is the maximum prime number not more than $|s_2|$. Then we have the following result: The first $P$ elements of $s_2$ are the same.

If not, we have $n_a(s_2, P) \neq 0$ and $n_b(s_2, P) \neq 0$, so we consider the value of $\mathcal{F}_{a,s_1}(x)$ and $\mathcal{F}_{a,s_2}(x)$ for $x = -\frac{n_a(s_2,P)}{n_b(s_2,P)}(1 + \epsilon)$.

Because $P$ is the maximum prime smaller than $|s_2|$, so $P > |s_2|/2$. So, there exist no $Q \in [|s_2|]$ and $Q \neq P$ such that $-\frac{n_a(s_2,Q)}{n_b(s_2,Q)} = -\frac{n_a(s_2,P)}{n_b(s_2,P)}$. If not, there must be $P|n_a(s_2,Q) + n_b(s_2,Q)$, which implies $Q \geq 2P > |s_2|$, a contradiction with $Q \in [|s_2|]$.

So, when $x = -\frac{n_a(s_2,P)}{n_b(s_2,P)}(1 + \epsilon)$ and $\epsilon \to 0$, at most one sub-rational formula in $\mathcal{F}_{a,s_1}$ and $\mathcal{F}_{a,s_2}$ tends to $\infty$ (i.e. the $\frac{n_a(s_i,P)}{xn_b(s_i,P)+n_a(s_i,P)}$).

So if $n_a(s_1, P) = n_a(s_2, P)$, then $\mathcal{F}_{a,s_i}(x)/(\frac{1}{|s_i|} \frac{n_a(s_i,P)}{xn_b(s_i,P)+n_a(s_i,P)})$ tends to 1 when $\epsilon \to 0$ for $i = 1, 2$. Since $|s_1| < |s_1| + 0.5 < |s_2|$, we have $\mathcal{F}_{a,s_1}(x) < \frac{-1}{(\text{len}(s_1)+0.5)\epsilon} < F_{a,s_2}(x)$ when $\epsilon \to 0$, which is a contradiction to step one.

If $n_a(s_1, P) \neq n_a(s_2, P)$, then $\mathcal{F}_{a,s_1}(x)$ will not tend to $\infty$ when $\epsilon \to 0$, which is also a contradiction to step one. So we proved step two.

**Step three:** We now prove the lemma.

Because $V_a^\alpha(s_1, |s_1|)$ and $V_a^\alpha(s_2, |s_2|)$ are not 0 and 1, so there exists at least one $a$ and at least one $b$ in the $s_1$ and $s_2$. By step two, without loss of generality, we assume that the first $P$ elements in $s_2$ are all $a$, and the position of the first $b$ in $s_2$ is at $m_0 + 1$. It is easy to see that $m_0 \geq P$.

Then, we consider the values of $\mathcal{F}_{a,s_1}(x)$ and $\mathcal{F}_{a,s_2}(x)$ for $x = -\frac{n_a(s_2,m_0+1)}{n_b(s_2,m_0+1)}(1 + \epsilon)$.

Firstly, we show that no other $Q \neq m_0 + 1$ and $Q \in [s_2]$ satisfy $-\frac{n_a(s_2,Q)}{n_b(s_2,Q)} = -\frac{n_a(s_2,m_0+1)}{n_b(s_2,m_0+1)}$. Because $m_0 \geq P > |s|/2$ and $n_b(s_2, m_0 + 1) = 1$, so if $-\frac{n_a(s_2,Q)}{n_b(s_2,Q)} = -\frac{n_a(s_2,m_0+1)}{n_b(s_2,m_0+1)}$, there must be $n_a(s_2, Q) \geq 2n_a(s_2, m_0 + 1) \geq 2m_0 \geq 2P > |s_2|$, which is a contradiction.

Then, similar as before, we can prove that if $x = -\frac{n_a(s_2,m_0+1)}{n_b(s_2,m_0+1)}(1 + \epsilon)$ and $\epsilon \to 0$, then $\mathcal{F}_{a,s_1}(x) \neq F_{a,s_2}(x)$, which is a contradiction to step one, so we prove the lemma. $\qquad\square$

We have the following result for sequences of the same length.

**Lemma C.2.** *Let $\alpha \in \mathbb{Z}_+$. For any such sequences $s_1$ and $s_2$ where $\text{len}(s_1) = \text{len}(s_2) = n$ and $V_a^\alpha(s_1, n) = V_a^\alpha(s_2, n)$. We have*

*(1) The number of $k \in [n]$ such that $V_a^\alpha(s_1, k) = 1$ is equal to the number of $k \in [n]$ such that $V_a^\alpha(s_2, k) = 1$.*

*(2) The number of $k \in [n]$ such that $V_a^\alpha(s_1, k) = 0$ is equal to the number of $k \in [n]$ such that $V_a^\alpha(s_2, k) = 0$.*

*Proof.* We prove (1) first.

If $V_a^\alpha(s_1, n) = V_a^\alpha(s_2, n) = 1$, then we know that all elements in $s_i$ are all 1; if $V_a^\alpha(s_1, n) = V_a^\alpha(s_2, n) = 0$, then we know that all elements in $s_i$ are all 0.

Assume $V_a^\alpha(s_1, n) = V_a^\alpha(s_2, n) \neq 0$ and $V_a^\alpha(s_1, n) = V_a^\alpha(s_2, n) \neq 1$. We define two rational polynomials $\mathcal{F}_{a,s_1}(x) = \frac{1}{|s_1|} \sum_{k=1}^{|s_1|} \frac{n_a(s_1,k)}{x n_b(s_1,k) + n_a(s_1,k)}$ and $\mathcal{F}_{a,s_2}(x) = \frac{1}{|s_2|} \sum_{k=1}^{|s_2|} \frac{n_a(s_2,k)}{x n_b(s_2,k) + n_a(s_2,k)}$. Then, similar to the proof of Lemma C.1, we have the following result: $\mathcal{F}_{a,s_1}(x) = F_{a,s_2}(x)$ for any $x \in \mathbb{R}$.

It is easy to see that when $x \to \infty$, $\frac{n_a(s_1,k)}{x n_b(s_1,k) + n_a(s_1,k)} \to 0$ when $n_b(s_1, k) \neq 0$, and if $n_b(s_1, k) = 0$, there must be $\frac{n_a(s_1,k)}{x n_b(s_1,k) + n_a(s_1,k)} = 1$. So consider the $\mathcal{F}_{a,s_1}(x) = F_{a,s_2}(x)$ when $x \to \infty$, we know that there must be the same number of 1 in $\{\frac{n_a(s_1,j)}{x n_b(s_1,j) + n_a(s_1,j)}\}_{j \in [n]}$ and $\{\frac{n_a(s_2,j)}{x n_b(s_2,j) + n_a(s_2,j)}\}_{j \in [n]}$.

It is easy to see that $\frac{n_a(s_1,j)}{x n_b(s_1,j) + n_a(s_1,j)} = 1$ equals $\frac{n_a(s_1,i)}{x n_b(s_1,i) + n_a(s_1,i)} = 1$ for any $i \leq j$, similar for $s_2$. So the same number of 1 in $\{\frac{n_a(s_1,j)}{x n_b(s_1,j) + n_a(s_1,j)}\}_{j \in [n]}$ and $\{\frac{n_a(s_2,j)}{x n_b(s_2,j) + n_a(s_2,j)}\}_{j \in [n]}$ implies there exist the same number of $k \in [|s_1|]$ such that $V_a^\alpha(s_1, k) = V_a^\alpha(s_2, k) = 1$. Hence, we directly get (1) in the lemma.

For (2) in the lemma, just need to consider that $q_a^\alpha(s_k, j) + q_b^\alpha(s_k, j) = 1$ and $V_a^\alpha(s_k, j) + V_b^\alpha(s_k, j) = 1$ for any $k \in \{0, 1\}$ and $j \leq n$. Similar to the proof of (1), we get the result. $\qquad \square$

Then, we calculate the following value:
$$\min_{s_1, s_2, \mathrm{len}(s_1) \leq L, \mathrm{len}(s_2) \leq L, V_a^\alpha(s_1, |s_1|) \neq V_a^\alpha(s_2, |s_2|)} |V_a^\alpha(s_1, |s_1|) - V_a^\alpha(s_2, |s_2|)|.$$
Firstly, we have the following lemma.

**Lemma C.3.** *If $f$ is a nonzero integral coefficient polynomial whose coefficients have absolute values not more than $A$, then for any $s \in \mathbb{Z}_+$ satisfying $e^s > A + 1$, we have $|f(e^s)| > 1$.*

*Proof.* Because $e^s > A + 1$, so $\sum_{i=0}^n A e^{is} = A \frac{e^{ns+s} - 1}{e^s - 1} \leq e^{ns+s} - 1$.

So, letting $L = \deg(f)$, we have $|f(e^s)| > e^{sL} - \sum_{i=0}^{L-1} A e^{is} = e^{sL} - A \frac{e^{sL} - 1}{e^s - 1} > e^{sL} - (e^{sL} - 1) = 1$, this is what we want. $\qquad \square$

**Lemma C.4.** *If $e^\alpha \geq L^{2L+2} + 1$, then*
$$\min_{s_1, s_2, \mathrm{len}(s_1) \leq L, \mathrm{len}(s_2) \leq L, V_a^\alpha(s_1, |s_1|) \neq V_a^\alpha(s_2, |s_2|)} |V_a^\alpha(s_1, |s_1|) - V_a^\alpha(s_2, |s_2|)| \geq \frac{1}{e^{2\alpha L} L^{2L+4}}.$$

*Proof.* Firstly, without loss of generality, let $V_a^\alpha(s_1) > V_a^\alpha(s_2)$, we have that:

$$
\begin{aligned}
& V_a^\alpha(s_1, |s_1|) - V_a^\alpha(s_2, |s_2|) \\
=\ & \tfrac{1}{|s_1|} \sum_{k=1}^{|s_1|} q_a^\alpha(s_1, k) - \tfrac{1}{|s_2|} \sum_{k=1}^{|s_2|} q_a^\alpha(s_2, k) \\
=\ & \tfrac{1}{|s_1|} \sum_{k=1}^{|s_1|} \tfrac{n_a(s_1,k)}{e^\alpha n_b(s_1,k) + n_a(s_1,k)} - \tfrac{1}{|s_2|} \sum_{k=1}^{|s_2|} \tfrac{n_a(s_2,k)}{e^\alpha n_b(s_2,k) + n_a(s_2,k)} \\
=\ & \tfrac{1}{|s_1|} \tfrac{F_{s_1}(e^\alpha)}{H_{s_1}(e^\alpha)} - \tfrac{1}{|s_2|} \tfrac{F_{s_2}(e^\alpha)}{H_{s_2}(e^\alpha)} \\
=\ & \tfrac{|s_2| F_{s_1}(e^\alpha) H_{s_2}(e^\alpha) - |s_1| F_{s_2}(e^\alpha) H_{s_1}(e^\alpha)}{|s_1||s_2| H_{s_1}(e^\alpha) H_{s_2}(e^\alpha)}.
\end{aligned}
$$

Here $H_{s_i}(x) = \Pi_{i=1}^{|s_i|} (n_b(s_i, k)x + n_a(s_i, k))$ is an integral coefficient polynomial, whose coefficients are not more than $\Pi_{i=1}^{|s_i|}(n_b(s_i, k) + n_a(s_i, k)) \leq L^L$; $\mathcal{F}_{s_i}(x) = \sum_{i=1}^{|s_i|} \frac{n_a(s_i,k) H_{s_i}(x)}{n_b(s_i,k)x + n_a(s_i,k)}$ is an integral coefficient polynomial, whose coefficients are not more than $\mathcal{F}_{s_i}(x) \leq \sum_{i=1}^{|s_i|} n_a(s_i, k) L^{L-1} \leq L^{L+1}$.

So, we have that $|s_2| F_{s_1}(x) H_{s_2}(x)$ and $|s_1| F_{s_2}(x) H_{s_1}(x)$ are two positive integral coefficient polynomials, so $|s_2| F_{s_1}(x) H_{s_2}(x) - |s_1| F_{s_2}(x) H_{s_1}(x)$ is an integral coefficient polynomial, where the absolute values of coefficients are not more than $L \times L^{L+1} \times L^L = L^{2L+2}$. Considering Lemma C.3 and $e^\alpha > L^{2L+2} + 1$, we can prove that $|s_2| F_{s_1}(e^\alpha) H_{s_2}(e^\alpha) - |s_1| F_{s_2}(e^\alpha) H_{s_1}(e^\alpha) > 1$.

Hence, we have that $|s_1||s_2| H_{s_1}(e^\alpha) H_{s_2}(e^\alpha) \leq L^2 (L \times L^L e^{\alpha L})^2 = L^{2L+4} e^{2\alpha L}$. We prove the result. $\qquad \square$

## C.2 A lemma for FNN classification

For FNN, we have the following result.

**Lemma C.5.** *Let $f(x) = \text{Relu}((Ax + B)/10^C)$ where $A, B, C$ are integers. Then $f(x)$ can be expressed as a network with width 6 and depth $O(\lceil \ln(\max\{|A|, |B|\})/q\rceil + \lceil C/q\rceil)$ and precision $q$.*

*Proof.* Firstly, we define a function $h_i(m)$ for any integer $m$: when $m \geq 0$, $h_i(m) = [m/10^{iq}] - 10^q[m/10^{q+iq}]$; when $m < 0$, $h_i(m) = -h_i(-m)$. Then we have that: $m = \sum_{i=0}^{[\log_{10} m/q]} 10^{iq} h_i(m)$. Hence, let $H_j(m) = \sum_{i=[\log_{10} m/q]-j}^{[\log_{10} m/q]} 10^{(i+j-[\log_{10} m/q])q} h_i(m)$ when $j \leq [\log_{10} m/q]$. It is easy to see that $H_{[\log_{10} m/q]}(m) = m$. When $j > [\log_{10} m/q]$, let $H_j(m) = m$.

Then, we can calculate $\text{Relu}(Ax + B)$ as follows:

**The first layer has width 6:**

Use $x$ to calculate $\text{Relu}(x)$, $\text{Relu}(-x)$, $\text{Relu}(H_0(A)x)$, $\text{Relu}(-H_0(A)x)$, $\text{Relu}(H_0(B))$, and $\text{Relu}(-H_0(B))$. Because $H_0(A)$ and $H_0(B)$ are $q$-precision, so such a layer just needs $q$-precision.

**The $n$-th layer has width 6:**

Assume $n - 2 < [\log_{10} A/q]$. Since $H_{n-1}(A)x = 10^q H_{n-2}(A)x + h_{[\log_{10} m/q]-n+1}(A)x$ and $H_{n-2}(A)x = \text{Relu}(H_{n-2}(A)x) - \text{Relu}(-H_{n-2}(A)x)$, $x = \text{Relu}(x) - \text{Relu}(-x)$, we can use a layer with $q$-precision to calculate $\text{Relu}(H_{n-1}(A)x), \text{Relu}(-H_{n-1}(A)x)$ by $\text{Relu}(H_{n-2}(A)x), \text{Relu}(-H_{n-2}(A)x)$ and $\text{Relu}(x), \text{Relu}(-x)$. If $n - 2 \geq [\log_{10} A/q]$, then we just need to keep $H_{n-1}(A) = H_{n-2}(A)$. $H_{n-1}(B)$ can be calculated similarly.

So we can use

$$\text{Relu}(x), \text{Relu}(-x), \text{Relu}(H_{n-2}(A)x), \text{Relu}(-H_{n-2}(A)x), \text{Relu}(H_{n-2}(B)), \text{Relu}(-H_{n-2}(B))$$

to calculate the following values in the $q$ precision:

$$\text{Relu}(x), \text{Relu}(-x), \text{Relu}(H_{n-1}(A)x), \text{Relu}(-H_{n-1}(A)x), \text{Relu}(H_{n-1}(B)), \text{Relu}(-H_{n-1}(B)).$$

**At the $T = [\log_{10} \max\{A, B\}/q] + 2$ layer, calculate $\text{Relu}(Ax + B)$.**

Because $T - 2 \geq [\log_{10} A/q]$ and $T - 2 \geq [\log_{10} B/q]$, so $H_T(A) = A$ and $H_T(B) = B$.

In this layer, we use

$$\text{Relu}(x), \text{Relu}(-x), \text{Relu}(H_T(A)x), \text{Relu}(-H_T(A)x), \text{Relu}(H_T(B)), \text{Relu}(-H_T(B))$$

to calculate $\text{Relu}(Ax + B)$, just use $H_T(A) = A$, $H_T(B) = B$ and $\text{Relu}(Ax + B) = \text{Relu}(\text{Relu}(Ax) - \text{Relu}(-Ax) + \text{Relu}(B) - \text{Relu}(-B))$. We can obtain the result.

Finally, in the next $[C/q]$ layers, we just need to divide $\text{Relu}(Ax + B)$ by $10^q$ in each layer, and in the last layer, divide it by $10^{C-q[C/q]}$. Then we obtain $\text{Relu}((Ax + B)/10^C)$ and prove the lemma. $\square$

**Lemma C.6.** *For any given $0 < x_1 < x_2 < \cdots < x_N$ where $x_i < C$ and $|x_i - x_j| > c$, any given $y_i \in [m]$, there exists a network $f$ with precision $q$, width $O(1)$, and depth $O(N\lceil \frac{|\ln mC/c|}{q}\rceil)$ that satisfies $|f(x_i) - y_i| < 0.2$.*

*Proof.* To begin with, we demonstrate the case for the scenario without precision limitations. In this scenario, achieving these tasks requires a depth of $O(N)$ and a width of $O(1)$, as described below:

**The first part**: In this part, $f^1$ is used to calculate the label of $x_1$ and has three layers.

The first layer $f^{1,1}(x)$ has width 2, with $f_1^{1,1}(x) = \text{Relu}([10^{q_1} x_2] - 10^{q_1} x)$ and $f_2^{1,1}(x) = \text{Relu}(x)$, where $q_1$ is the minimum integer such that $10^{q_1} x_2 \geq 10^{q_1} x_1 + 2$. Since $x_2 \geq x_1 + c$, we have $10^{q_1} \leq O(1/c)$.

The second layer $f^{1,2}(x)$ has width 3: $f_1^{1,2}(x) = \text{Relu}(1 + f_1^{1,1}(x)/10^{q_2})$, $f_2^{1,2}(x) = \text{Relu}(10 \times (0.5 - f_1^{1,1}(x)))$ and $f_3^{1,2}(x) = \text{Relu}(f_2^{1,1}(x))$, where $q_2$ is the minimum integer such that $10^{q_2-2} > m$. It is easy to see that $q_2 \leq O(\log m)$.

The third layer $f^{1,3}(x)$ is the output of the first part, which has width 2: $f_1^1(x) = f_1^{1,3}(x) = $ Relu$(y_1(f_1^{1,2}(x) - f_2^{1,2}(x)))$ and $f_2^1(x) = f_2^{1,3}(x) = $ Relu$(f_3^{1,2}(x))$.

So when $i \neq 1$, we have $f_1^{1,1}(x_i) = 0$, so $f_1^{1,2} = 1$ and $f_2^{1,2} = 5$. Hence $f_1^1(x) = $ Relu$(y_1(1-5)) = 0$. When $i = 1$, using the definition of $q_1$, we have $20 > 10^{q_1}x_2 - 10^{q_1}x_1 \geq $ Relu$([10^{q_1}x_2] - 10^{q_1}x_1) = f_1^{1,1}(x_1) \geq $ Relu$(10^{q_1}x_2 - 1 - 10^{q_1}x_1) \geq 1$, so $1 \leq f_1^{1,2}(x) = 1 + f_1^{1,1}(x)/10^{q_2} \leq 1+0.2/m$ (use the value of $q_2$) and $f_2^{1,2}(x_1) = $ Relu$(10 \times (0.5 - f_1^{1,1}(x))) \leq $ Relu$(10 - (0.5-1)) = 0$. Hence, we have $f_1^1(x_1) = $ Relu$(y_1 f_1^{1,2}) \in [y_1, y_1 + 0.2]$, using the $|y_1| \leq m$. Finally, we have $f_1^1(x_i) \in [y_1, y_1 + 0.2]$ if and only if $i = 1$ and $f_2^1(x_i) = x_i$ for any $i \in [N]$ which is apparent.

**The $i$-th part, where $i \leq N$:** This part is used to calculate the label of $x_i$ and has five layers.

The input to $i$-th part is the output of $(i-1)$-th part. If the $(i-1)$-th part $f^{i-1}(x)$ satisfies: the output of $f^{i-1}(x)$ has width 2, $|f_1^{i-1}(x_j) - y_j| \leq 0.2$ when $j \leq i-1$, $f_1^{i-1}(x_j) = 0$ when $j > i-1$, and $f_2^{i-1}(x_j) = x_j$ for any $j \in [N]$, then we can make the output of the $i$-th part $f^i(x)$ to satisfy: $f^i(x)$ has width 2, $|f_1^i(x_j) - y_j| \leq 0.2$ when $j \leq i$, $f_1^i(x_j) = 0$ when $j > i$, and $f_2^i(x_j) = x_j$ for any $j \in [N]$.

To do this, the $i$-th part needs six layers:

The first layer and the second layer output $f^{i,2}(x)$ satisfying:

$$f_1^{i,2}(x) = f_1^{i-1}(x), f_2^{i,2}(x) = 2\text{Relu}(\lceil 10^{q_1}x_i \rceil/10^{q_1} - f_2^{i-1}(x)) + f_2^{i-1}(x) \text{ and } f_3^{i,2}(x) = f_2^{i-1}(x),$$

where $q_1$ is the minimum integer that satisfies $\lceil 10^{q_1}x_i \rceil/10^{q_1} < x_i + c/3$. It is easy to see that $10^{q_1} \leq O(1/c)$. Now we show that $f_2^{i,2}(x_p) - f_2^{i,2}(x_i) > c/3$ for any $p \neq i$.

We know that $f_2^{i,2}(x_j) = x_j$ when $j > i$, and $f_2^{i,2}(x_k) = 2\lceil 10^{q_1}x_i \rceil/10^q - x_k$ when $k \leq i$, so there must be $f_2^{i,2}(x_p) - f_2^{i,2}(x_i) \geq \min_{k<i, \, j>i}\{2\lceil 10^{q_1}x_i \rceil/10^q - x_k, x_j\} - (2\lceil 10^{q_1}x_i \rceil/10^{q_1} - x_i)$ for all $p \neq i$. Based on the definition of $q_1$, we know that $x_j - (2\lceil 10^{q_1}x_i \rceil/10^{q_1} - x_i) > c/3$ for any $j > i$ and $(2\lceil 10^q x_i \rceil/10^{q_1} - x_k) - (2\lceil 10^{q_1}x_i \rceil/10^{q_1} - x_i) > c$ for any $k < i$, so we get the result.

Then let the next three layers follow the first part, and use $f^{i,2}(x_j)$ to obtain $f^{i,5}(x)$ that satisfies: $f_1^{i,5}(x) = f_1^{i-1}(x)$; $|f_2^{i,5}(x_i) - y_i| \leq 0.2$, $f_2^{i,5}(x_j) = 0$ when $j \neq i$; and $f_3^{i,5}(x) = f_3^{i,2}(x) = f_2^{i-1}(x)$.

And the last layer is the output of the $i$-th part, where: $f_1^i(x) = $ Relu$(f_1^{i,5}(x) + f_2^{i,5}(x))$ and $f_2^i(x) = f_3^{i,5}(x)$, which is what we want.

**The output part:** Just output $f_1^n(x)$, and this is what we want.

It is easy to check that such a network has $O(N)$ nodes and has width $O(1)$, where each parameter is not greater than $O(mC/c)$. Now we just need to turn the parameters of the above network to be of $q$-precision. By Lemma C.5 and each layer defined before, if a node has parameters beyond precision, we can use a network with $O(1)$ width and $\lceil O(\ln mC/c) \rceil$ depth instead of it. So we obtain the result. $\qquad\square$

We have the following lemma which is used in the paper [25].

**Lemma C.7.** *For any $v \in \mathbb{R}^n$ and $T \geq 1$, let $u \in \mathbb{R}^n$ be uniformly randomly sampled from the hypersphere $S^{n-1}$. Then we have $P(|\langle u, v \rangle| < \frac{||v||_2}{T}\sqrt{\frac{8}{n\pi}}) < \frac{2}{T}$.*

So we can prove the following lemma.

**Lemma C.8.** *For any given $\{x_i\}_{i=1}^N \subset \mathbb{R}^n$ where $||x_i||_\infty \leq 1$ and $||x_i - x_j||_\infty > c$, any given $y_i \in [m]$, there exists a network $f$ with width $O(1)$ and depth $O(N\lceil \frac{\ln mNn/c}{q} \rceil)$ that can satisfy $|f(x_i) - y_i| < 0.2$.*

*Proof.* Firstly, we can find a vector $u \in \mathbb{R}^n$ that satisfies $||u||_2 = 1$ and $|ux_i - ux_j| > \Omega(\frac{c}{N^2 n})$, just using Lemma C.7.

Then, we can find a $u_r \in \mathbb{R}^n$ such that: $||u_r - u||_\infty \leq O(\frac{c}{N^2 n^2})$ and each weight of $u_r$ is a finite decimal with precision $O(\ln(\frac{N^2 n^2}{c})/q)$, so we have $|u_r x_i - u_r x_j| > \Omega(\frac{c}{N^2 n})$ and $|u_r x_i| \leq 2n$.

Then, we can use a layer to map $x_i$ to $u_r x_i + 2n$, and then use Lemma C.6 to find an FNN to memorize $\{(u_r x_i + 2n, y_i)\}$ and obtain the result. □

## C.3  Proof of Theorem 4.1

Now we prove Theorem 4.1.

*Proof.* The proof has five parts.

For any given $(x, y) \in S$, we define a sequence $s_{x,i} \in 2^{\{a,b\}}$ as follows: $s_{x,i}$ has the same length as $x$, and the $j$-th element of $s_{x,i}$ is $a$ if and only if $x[j] = \gamma_i$.

**Part One: The Embedding.**

We will embed in the following way:

The basic symbols $\gamma_i$ will embed in $(v_1^i, v_2^i, \ldots, v_T^i) \in \{0,1\}^{4T}$ where $v_j^i = (0,0,1,1)$ when $j \neq i$, $v_i^i = (0,1,0,1)$.

Hence, the input sentence $x$ whose $\text{len}(x) = n$ will be embedded in a vector $V_x \in \{0,1\}^{n \times 4T}$.

**Part Two: Use the $O(\lceil \ln(L \ln L)/q \rceil)$ layers to calculate the position value.**

Let $\alpha$ be the minimum positive integer such that $e^{10^\alpha} \geq L^{3L}$. It is easy to see that $\alpha \leq O(\ln(L \ln L))$.

**Step one:** Firstly, for any input $x$ whose embedding matrix is $V_x$, we use $\lceil \frac{\alpha}{q} \rceil$ hidden layer to make all the $(4i-1)$-th columns of $V_x$ expand $10^\alpha$ times, where $i \in [T]$.

To do this, in the first $\lceil \alpha/q \rceil$ layers, we make the parameters in the FNN layer and attention layer 0, but the transition matrix in the residual layer is defined as: the $(j,j)$-th weight of the transition matrix is 1 if $j \neq 4i-1$ for any $i \in [T]$; the $(4i-1, 4i-1)$-th weight is $10^q$; other weights are 0. Of course, to make the $(4i-1)$-th columns expand $10^\alpha$ times, in the $\lceil \alpha/q \rceil + 1$-th layer, the transition matrix in the residual layer is $10^{\alpha - q\lceil \frac{\alpha}{q} \rceil}$ at $(4i-1, 4i-1)$-th weights.

**Step two:** We use a hidden layer to calculate $q_a^{10^\alpha}(s_{x,i}, k)$.

Firstly, let the attention layer in this hidden layer have $T$ heads and let the $i$-th head in this attention layer be written as $\text{softmax}(xQ_i K_i x^T + M)xV_i$, where $Q_i \in \{0,1\}^{4T \times 4T}$, the $(4i, 4i-1)$-th weight of $Q_i$ is 1, others are 0; $K_i \in \{0,1\}^{4T \times 4T}$ and $K_i$ is the identity matrix; $V_i \in \{0,1\}^{4T \times 4T}$, where the $(4i-2, 4i-2)$-th weights of $V_i$ are 1 and others are 0.

Now, we consider the output of the $i$-th head when input $x$ into the transformer, assuming that the output of the previous step is $x'$. Firstly, by part one and the definition of $Q_i, K_i$, we know that the $k$-th row of $x' Q_i K_i x'^T$ is

$$
\begin{aligned}
& x'_k Q_i K_i x'^T \\
=~& x'_k Q_i x'^T \\
=~& ((x'_k)_{4i}(x'_1)_{4i-1}, (x'_k)_{4i}(x'_2)_{4i-1}, \ldots, (x'_k)_{4i}(x'_n)_{4i-1}) \\
=~& ((x'_1)_{4i-1}, (x'_2)_{4i-1}, \ldots, (x'_n)_{4i-1}) \\
=~& 10^\alpha (I(x[1] \neq \gamma_i), I(x[2] \neq \gamma_i), \ldots, I(x[n] \neq \gamma_i)).
\end{aligned}
$$

Hence, considering the definition of $M$, the weight in $k$-th row and $4i-2$-th column of $\text{softmax}(x' Q_i K_i x'^T + M)x'$ is $\frac{\sum_{j=1}^k I(x[1]=\gamma_i)}{e^{10^\alpha} \sum_{j=1}^k I(x[1] \neq \gamma_i) + \sum_{j=1}^k I(x[1]=\gamma_i)}$, which is equal to $q_a^{10^\alpha}(s_{x,i}, k)$.

Then, it is easy to check that $(4i-2)$-th column of $\text{softmax}(x' Q_i K_i x'^T)x' V_i$ is

$$
(q_a^{10^\alpha}(s_{x,i}, 1), q_a^{10^\alpha}(s_{x,i}, 2), q_a^{10^\alpha}(s_{x,i}, 3), \ldots, q_a^{10^\alpha}(s_{x,i}, \text{len}(x)))^T.
$$

By the definition of $V_i$, it is easy to know that other columns of $\text{softmax}(x' Q_i K_i x'^T)x' V_i$ are 0.

Hence, let the residual layer use a matrix $W \in \{0,1\}^{4T \times 4T}$, which is defined as: for the $i \in [T]$, $(4i-1, 4i-1)$-th and $(4i, 4i)$-th weights of $W_i$ are 1, and the other weights are 0. The FNN layer is

all 0. Then, assume that the whole layer will calculate a matrix $M_1(x) \in \mathbb{R}^{\text{len}(x) \times 4T}$ by inputting $x$ to the transformer. It is easy to check that the $(4i-2)$-th ($i \in [T]$) column of $M_1(x)$ is

$$(q_a^{10^\alpha}(s_{x,i}, 1), q_a^{10^\alpha}(s_{x,i}, 2), q_a^{10^\alpha}(s_{x,i}, 3), \ldots, q_a^{10^\alpha}(s_{x,i}, \text{len}(x)))^T.$$

Other columns are the same as $V_x$.

**Step three:** Next, we use a hidden layer to calculate $V_a^{10^\alpha}(s_{x,i}, k)$.

In this layer, the input is the $M_1(x)$ gotten by the above layer, so we can calculate the position value by $M_1(x)$.

Let the attention layer in this layer have $T$ heads, and in the $i$-th head ($i \in [T]$), we have $Q_i = K_i = 0$, $V_i \in \{0, 1\}^{4T \times 4T}$, and the $(4i-2, 4i-2)$-th weights of $V_i$ are 1, while others are 0.

Because $Q_i = K_i = 0$, it is easy to check that in the $i$-th head, it holds

$$\text{softmax}(M_1(x)Q_iK_iM_1^T(x) + M)M_1(x) = \text{softmax}(0 + M)M_1(x) = I_\downarrow M_1(x),$$

where $I_\downarrow$ is an under-triangle semi-matrix, and the $i$-th row is $(\frac{1}{i}, \frac{1}{i}, \ldots, \frac{1}{i}, 0, 0, \ldots, 0)$ (the first $i$ weights are $1/i$, while others are 0). Considering that $V_a^{10^\alpha}(s, k) = \frac{1}{k} \sum_{i=1}^{k} q_a^{10^\alpha}(s, i)$, based on the definition of $M_1(x)$, we know that the $4i-2$ columns of $i$-th head are

$$I_\downarrow((M_1(x)^T)_{4i-2})^T = (V_a^{10^\alpha}(s_{x,i}, 1), V_a^{10^\alpha}(s_{x,i}, 2), V_a^{10^\alpha}(s_{x,i}, 3), \ldots, V_a^{10^\alpha}(s_{x,i}, \text{len}(x)))^T.$$

By the definition of $V_i$, it is easy to know that other columns are 0.

Let the residual layer and the FNN layer in this layer be the same as in the layer in step two. So we can easily check that when we input the $x$ to the transformer, we can obtain the matrix $M_2(x)$ in this layer, whose $(4i-2)$-th column ($i \in [T]$) is

$$(V_a^{10^\alpha}(s_{x,i}, 1), V_a^{10^\alpha}(s_{x,i}, 2), V_a^{10^\alpha}(s_{x,i}, 3), \ldots, V_a^{10^\alpha}(s_{x,i}, \text{len}(x)))^T.$$

Other columns are the same as $M_1(x)$, which implies other columns are the same as $v_x$.

**Part Three: Use FNNs.**

Define the set $V_j = \{V_a^{10^\alpha}(s_{x,i}, j)\}_{(x,y) \in S, i \in [T]}$ where $j \in [L]$ and $V = \cup_{j \in [L]} V_j$.

We try to find an FNN $f_1$ such that:

(1) $|f_1(1)| < 0.2$ and $|f_1(0)| < 0.2$;

(2) If $z \in V_j / \{0, 1\}$, then there exists a $z_q \in [NT]$ such that $|f_1(z) - z_q(NTL)^{2j-2}| < 0.2$; moreover, when $z_1, z_2 \in v_j$, there exists $(z_1)_q \neq (z_2)_q$.

Since there exist at most $NT$ samples in $V_j$ and $V_j \cap V_i \subset \{0, 1\}$ when $i \neq j$ by Lemma C.1, such a network $f_1$ must exist. Hence, considering Lemma C.4 and Lemma C.6, we know that such an FNN $f_1$ just needs precision $q$ and $O(NLT\lceil L10^\alpha \ln LNT/q\rceil)$ layers and $O(1)$ width.

Based on the $M_2(x)$ and the above $f_1$, when input $x$ to the transformer, we define the matrix $M_3(x)$ which has the same size as $M_2(x)$ as: the $(4i-1)$-th columns of $M_3$ are $f_1((M_2(x))_{4i-2})$, where $(M_2(x))_{4i-2}$ is the $(4i-2)$-th column of $M_2(x)$; other columns are the same as those of $M_2(x)$.

Then by Lemma B.2, we can use a transformer with width $O(1)$ and depth $O(NLT\lceil 10^\alpha L \ln LNT/q\rceil)$ to obtain $M_3(x)$ by $M_2(x)$.

**Part Four.** In this part, we use several hidden layers to obtain a vector that is different from $x$ by $M_3(x)$.

In the first layer, this attention layer has $T$ heads, in the $i$-th head, $Q_i = 0$, $K_i = 0$ and $V_i \in \{0, 1\}^{4T \times 4T}$ such that the $(4i-1, 4i-1)$-th weight of $V_i$ is 1; others are 0.

The residual layer uses $W \in \{0, 1\}^{4T \times 4T}$ which is defined as: for the $i \in [T]$, $(4i-2, 4i-2)$-th weights of $W$ are 1, others are 0. The FNN layer is all 0.

Finally, we use $\lceil \frac{2L \ln NTL}{q} \rceil$ layers to reduce the $(4i-1)$-th columns ($i \in [T]$) of the output of the first layer by $(NTL)^{2L}$ times, in order to reduce the norm of the output for the above layer to no more than 1.

Let the last row for the output of the above layers be $M_l(x)$ when input $x$ with length $n$ to the transformer. Then based on the definition of $M_3(x)$, we have that:

(1) The $(4i-2)$-th weight of $M_l(x)$ is $V_a^{10^\alpha}(s_{x,i}, n)$ for any $i \in [n]$.

(2) The $(4i-1)$-th weight is $\frac{\sum_{j=1}^n f_1(V_a^{10^\alpha}(s_{x,i}, j))}{n(NTL)^{2L}}$ for any $i \in [n]$.

Firstly, by the definition of position value and $f_1$, it is easy to see that $||M_l(x)||_\infty \le 1$. Then we will prove that: for all $(x, y), (z, y_z) \in S$ that do not satisfy $\text{typ}(x) = \text{typ}(z) = \gamma_l$ for some $l \in [T]$, we have $||M_l(x) - M_l(z)||_\infty \ge \min\{\frac{1}{e^{2 \times 10^\alpha L} L^{2L+4}}, \frac{1}{L(NTL)^{2L}}\}$.

When $\text{typ}(x) = \text{typ}(z) = 1$, let $\gamma_i = \text{typ}(x)$. Then it is easy to see that $\gamma_i \notin \text{typ}(z)$. Now we consider the $(4i-2)$-th weights of $M_l(x)$ and $M_l(z)$. It is easy to see that the $(4i-2)$-th weight of $M_l(x)$ is 1 but the $(4i-2)$-th weight of $M_l(1)$ is 0. Which is what we want.

When $\text{typ}(x) > 1$, we consider two situations.

If $\text{len}(x) \ne \text{len}(z)$, by $\text{typ}(x) > 1$, there must be an $i$ such that $\gamma_i \in \text{typ}(x)$ and $\text{typ}(z) \ne \{\gamma_i\}$, so we consider the $(4i-2)$-th weights of $M_l(x)$ and $M_l(z)$. By Lemmas C.1 and C.4, for such an $i \in [T]$, there must be $|V_a^{10^\alpha}(s_{x,i}, \text{len}(x)) - V_a^{10^\alpha}(s_{z,i}, \text{len}(z))| > \frac{1}{e^{2 \times 10^\alpha L} L^{2L+4}}$.

If $\text{len}(x) = \text{len}(z)$, let $i \in [T]$ satisfy $s_{x,i} \ne s_{z,i}$. Such $i$ must exist, because $s_{x,i} = s_{z,i}$ implies that $\gamma_i$ has the same position in $x$ and $z$. Since it stands for any $i \in [T]$, there must be $x = z$. If $V_a^{10^\alpha}(s_{x,i}, \text{len}(x)) \ne V_a^{10^\alpha}(s_{z,i}, \text{len}(z))$, then we have similar results as before. If not, then we consider the $(4i-1)$-th weights of $M_l(x)$ and $M_l(z)$, which are $\frac{\sum_{j=1}^n f_1(V_a^{10^\alpha}(s_{x,i}, j))}{\text{len}(x)(NTL)^{2L}}$ and $\frac{\sum_{j=1}^n f_1(V_a^{10^\alpha}(s_{z,i}, j))}{\text{len}(z)(NTL)^{2L}}$.

Assume $k$ is the maximum one in $[n]$ satisfying $V_a^{10^\alpha}(s_{x,i}, k) \ne V_a^{10^\alpha}(s_{z,i}, k)$. Based on the definition of $f_1$ and Lemma C.2, we know that

$$
\begin{aligned}
&| \sum_{j=1}^n f_1(V_a^{10^\alpha}(s_{x,i}, j)) - \sum_{j=1}^n f_1(V_a^{10^\alpha}(s_{z,i}, j))| \\
> \quad & (NTL)^{2k-2} - 0.2 - (\sum_{j=0}^{k-1} NT(NTL)^{2j-2} + 0.2) \\
= \quad & (NTL)^{2k-2} - \frac{(NTL)^{2k-2} - 1}{NTL^2 - 1/(NT)} - 0.2L > 1.
\end{aligned}
$$

So we have

$$
|\frac{\sum_{j=1}^n f_1(V_a^{10^\alpha}(s_{x,i}, j))}{\text{len}(x)(NTL)^{2L}} - \frac{\sum_{j=1}^n f_1(V_a^{10^\alpha}(s_{z,i}, j))}{\text{len}(z)(NTL)^{2L}}| \ge \frac{1}{L(NTL)^{2L}}.
$$

The case of $\text{typ}(z) > 1$ is similar, so we obtain the result.

**Part Five.**

If $\text{typ}(x) = \text{typ}(z) = \gamma_i$, Proposition 4.3 shows that there must be $\mathcal{F}(x) = F(z)$. Considering the conditions of the theorem, $x$ and $z$ have the same label, so let $S_1 = \{(x, y) \in S : |\text{typ}(x)| = 1\} \subset S$ and $S_2 = \{(\gamma_i, y_i) : \exists (x, y_i) \in S, \text{typ}(x) = \gamma_i\}$. We just need a transformer to memorize $S_2 \cup S/S_1$.

In this part, we use an FNN to obtain the result. By Lemma C.8 and Part Four, since $10^\alpha = O(L \ln L)$, there exists an FNN with depth $O(N\lceil L^2 \ln^2 NTL/q\rceil)$ and width $O(1)$ which can classify $M_l(x)$ to $y$ for all $(x, y) \in S_2 \cup S/S_1$. Hence, by Lemma B.2, we can use a transformer with depth $(O(N\lceil L^2 \ln^2 NTL/q\rceil)$ and width $O(1)$ to simulate that FNN network. Adding all the above four parts, we can directly get the theorem. $\square$

### C.4 Proof of Proposition 4.3

*Proof.* Assume that $x$ satisfies $\text{typ}(x) = \{\gamma_i\}$. To prove the proposition, we need only to show $\mathcal{F}(x) = \mathcal{F}(\gamma_i)$ for any given transformer $\mathcal{F}$.

Firstly, we will show that, in each hidden layer $\mathcal{F}^j$ of $\mathcal{F}$, if the input of $\mathcal{F}^j$ ensures that each row is the same, then the output of $\mathcal{F}^j$ ensures that each row is the same.

We just need to prove it for $j = 1$; other layers are similar. Let $V_x$ be the embedding matrix of $x$ and easily see that the input of the first hidden layer is $V_x$ whose rows are the same.

Let the first layer be written as

$$\mathcal{F}^1(V_x) = V_x W_1 \quad + \quad \sum_{i=1}^{H} \text{softmax}(V_x Q_i K_i V_x^t + M) V_x V_i$$
$$+ \quad \text{FNN}(V_x W_1 + \sum_{i=1}^{H} \text{softmax}(V_x Q_i K_i V_x^t + M) V_x V_i).$$

In the attention layer, because each row of $V_x$ is the same, we have $\text{softmax}(V_x Q_i K_i V_x^t + M) V_x = I_\downarrow V_x$ where $I_\downarrow$ is a semimatrix under, and the $i$-th row is $(\frac{1}{i}, \frac{1}{i}, \ldots, \frac{1}{i}, 0, 0, \ldots, 0)$ (the first weights of $i$ are $1/i$, others are 0). Then the first hidden layer is

$$\mathcal{F}^1(V_x) = V_x W_1 + \sum_{i=1}^{H} I_\downarrow V_x V_i + \text{FNN}(V_x W_1 + \sum_{i=1}^{H} I_\downarrow V_x V_i).$$

By the definition of $I_\downarrow$, we have that $I_\downarrow V_x = V_x$, and by the definition of transformer, it is easy to see that all other parameter matrices in the attention layer and FNN layer are all right multiplied for $V_x$, which does the same transformation between all rows in the $V_x$. Thus all rows of $\mathcal{F}^1(V_x)$ are the same. Similar for the other layers.

To be convenient, let $\mathcal{F}$ have $l$ hidden layers and $\mathcal{F}^l(x)$ be the output of the last hidden layer of $\mathcal{F}(x)$. By the above result, we know that the rows in the output of the first hidden layer are all the same; hence we can obtain the output of the second hidden layer all the same, and so forth. Finally, we have that the rows of $\mathcal{F}^l(x)$ are all the same. Hence, by Lemma B.1, the first row of $\mathcal{F}^l(x)$ is equal to the first row, which is also the only row, of $\mathcal{F}^l(\gamma_i)$. So all the rows in the $\mathcal{F}^l(x)$ are equal to $\mathcal{F}^l(\gamma_i)$.

Now, we can prove the proposition. By the structure of the transformer, the output of the $\mathcal{F}(x)$ is a linear transformation on the last row of $\mathcal{F}^l(x)$. Because we have shown that each row of the $\mathcal{F}^l(x)$ is the same as $\mathcal{F}^l(\gamma_i)$, so $\mathcal{F}(x)$ is also equal to the linear transformation on the $\mathcal{F}^l(\gamma_i)$, which is equal to $\mathcal{F}(\gamma_i)$. So we get the result. $\square$

# D    Proofs of results in Section 4.2

## D.1    Proof of Theorem 4.4

First, we prove the sufficient condition.

*Proof.* The proof of the sufficient condition for Theorem 5.1 needs three parts.

**Part One:** In this part, we construct a new set $S_{ss}$ as follows.

For each $(x, y) \in S$, we select an $x_z \in S_x$ satisfying the following conditions to form a set $S_{ss} = \{x_z : (x, y) \in S\}$.

(c1) for any $k$, the $x_z[\text{len}(x) + 1]$ are the same for all $(x, y) \in S$ satisfying $\text{typ}(x) = \{\gamma_k\}$, and

(c2) $\text{len}(z) \leq L + 2$.

By the conditions given in the theorem, (c1) must be satisfied. Based on the definition of $S_x$, $z \in S_x$ only affects the first $L + 1$ symbols and the last symbol in $z$, and the subsequent symbols will not affect the overall satisfaction of the conditions in the definition of $S_x$. So, if $z$ is longer than $L + 2$, we just need to remove the $L + 2$ to the penultimate symbols in $z$, and it is still in the $S_x$. So, (c2) can be satisfied.

**Part Two: Define a new language.**

**First, we will use $S_{ss}$ to define a new language $S_n$.**

For any given $(x, y) \in S$ and $x_z \in S_x$, we define $\gamma_j^{x_z} = x_z[j]$ if and only if $j \leq \text{len}(x_z)$ and $\gamma_{\text{len}(x_z)+1}^{x_z} = \gamma_0$. Based on that, we define the sentence $x_z^k = (x, \gamma_{\text{len}(x_x)+1}^{x_z}, \ldots, \gamma_k^{x_z})$, and let it have the label $y_k^{x_z} = \gamma_{k+1}^{x_z}$, where $k \in \{\text{len}(x), \ldots, \text{len}(x_z)\}$.

Now we construct a set $S_n$:

$$S_n = \{(x_z^k, y_k^{x_z}) \| x_z \in S_{ss}, k \in \{\text{len}(x), \text{len}(x) + 1, \ldots, \text{len}(x_z)\}\}.$$

We make the following operation on $S_n$ until it stops: If there exists a $((x_a)_z^k, y_k^{(x_a)_z}), ((x_b)_z^k, y_k^{(x_b)_z}) \in S_n$ such that $(x_a)_z^k = (x_b)_z^k$ but $y^{(x_a)_z, k} \neq y^{(x_b)_z, k}$ for some $k$ and $\text{len}(x_a) < \text{len}(x_b)$, then remove $((x_a)_z^k, y_k^{(x_a)_z})$ from $S_n$.

**Second, we show that $S_n$ will be a language.** It suffices to show that after doing such an operation, we can ensure that for any $(x_1, y_1), (x_2, y_2) \in S_n$, $y_1 \neq y_2$ implies $x_1 \neq x_2$.

We just need to show that, if $(x_i, y_i) \in S_n$ where $i = 1, 2$ satisfy $y_1 \neq y_2$ but $x_1 = x_2$, the operation will not stop. To show that, we just need to prove that if $(x_a)_z^k = (x_b)_z^k$ is valid but $y_k^{(x_a)_z} \neq y_k^{(x_b)_z}$ for some $x_a \neq x_b$ and $k$, then there must be $\text{len}(x_a) \neq \text{len}(x_b)$. If not, we have $x_a = (x_1)_{[\text{len}(x_a)]} = (x_1)_{[\text{len}(x_b)]} = x_b$ where $x_1 = (x_a)_z^k$, which is a contradiction with $x_a \neq x_b$.

**Finally, we show $S_n$ is a language that can be memorized by a no-CoT-transformer.**

To prove this, we need only to show that $S_n$ satisfies the condition in Theorem 4.1. For any given $i$, if there exists a $(x_z^k, y_k^{x_z}) \in S_n$ such that $\text{typ}(x_z^k) = \{\gamma_i\}$, by the definition of $S_x$, there must be $k = \text{len}(x)$. Hence, we have $x_z^k = x$. Then considering (c1) of the definition of $S_{ss}$, we have that these $y_k^{x_z}$ are the same for any $x_z^k$ satisfied $\text{typ}(x_z^k) = \gamma_i$. This result implies that $S_n$ satisfies the condition in Theorem 4.1.

Moreover, since $S_n$ contains at most $O(L|S|)$ samples and each sentence in it has length $L + 2$, such a transformer has width $O(T)$, depth $O(NL^2 \lceil \ln^2(NTL)L^2/q \rceil)$, and heads $O(T)$.

**Part Three: Prove the result.**

Assume that $\mathcal{F}$ is a no-CoT-transformer that can memorize $S_n$. We will show that $\mathcal{F}$ can memorize $S$ as a CoT-transformer, which can directly prove Theorem 4.4.

**Step One:**

For any $(x, y) \in S$ satisfying the fact that any $(x_z^k, y_k^{x_z})$ does not removed from $S_n$, where $k \in \{\text{len}(x), \ldots, \text{len}(x_z)\}$, we show that $\widehat{F}_{\text{cot}}(x) = y$.

Because we do not remove any $(x_z^k, y_k^{x_z})$ from $S_n$ ($k \in \{\text{len}(x), \ldots, \text{len}(x_z)\}$) from $S_n$, there must be $\widehat{F}(x_z^k) = y_k^{x_z}$, and consider that sentence $(x_z^k, y_k^{x_z}) = x_z^{k+1}$ and $y_{\text{len}(x_z)}^{x_z} = \gamma_0$, so the CoT of $\mathcal{F}$ when input $x$ is $(\gamma_j^{x_z})_{j=\text{len}(x)+1}^{\text{len}(x_z)+1}$, and $\gamma_j^x = x_z[j]$ if and only if $j \leq \text{len}(z)$, and $\gamma_{\text{len}(x_z)+1}^{x_z} = \gamma_0$ meaning stop. Then by the definition of $\gamma_j^{x_z}$, we know that the symbol before $\gamma_0$ is $y$, so $\widehat{F}_{\text{cot}}(x) = y$. We get the result.

**Step Two:**

For any $(x, y) \in S$ satisfying the fact that some $(x_z^k, y_k^z)$ is removed from $S_n$, we show that $\widehat{F}_{\text{cot}}(x) = y$.

If not, let $(x, y) \in S$ be the sentence of maximum length such that $\widehat{F}_{\text{cot}}(x) \neq y$.

Let $k_m$ be the minimum value such that $(x_z^{k_m}, y_{k_m}^{x_z})$ has been removed from $S_n$. Then let $(x_1, y_1) \in S$ be the maximum length sentence such that $(x_1)_z^{k_m} = x_z^{k_m}$. Because $(x_z^{k_m}, y_{k_m}^{x_z})$ has been removed from $S_n$, so there exists at least one $(x_0, y_0) \in S$ such that $x_z^{k_m} = (x_0)_z^{k_m}$ and $\text{len}(x_0) > \text{len}(x)$, so we have $x_1 \neq x$ and $\text{len}(x_1) > \text{len}(x)$. Now we will show that $\widehat{F}_{\text{cot}}(x_1) = \widehat{F}_{\text{cot}}(x)$ and $y = y_1$, so we have $\widehat{F}_{\text{cot}}(x_1) \neq y_1$, which is a contradiction to the maximum of $\text{len}(x)$.

Firstly, we show that there must be $y = y_1$. Considering that $(x_z)_{[\text{len}(x_1)]} = x_1$ and $\text{len}(x_1) > \text{len}(x)$, by the definition of $S_x$, we know that $y = y_1$.

Secondly, by the minimum of $k_m$, similar to step one, we know that the first $k_m - \text{len}(x)$ step CoT of $\mathcal{F}$ when input $x$ is $(y_{\text{len}(x)}^{x_z}, y_{\text{len}(x)+1}^{x_z}, \ldots, y_{k_m-1}^{x_z}) = (\gamma_{\text{len}(x)+1}^{x_z}, \gamma_{\text{len}(x)+2}^{x_z}, \ldots, \gamma_{k_m}^{x_z})$. Considering that $k_m \geq \text{len}(x_1)$ and $(x_1)_z^k = x_z^k$ for any $k \leq k_m$, so by the minimum of $k_m$, we know that the first $k_m - \text{len}(x_1)$ step CoT of $\mathcal{F}$ when input $x_1$ is $(y_{\text{len}(x_1)}^{(x_1)_z}, y_{\text{len}(x_1)+1}^{(x_1)_z}, \ldots, y_{k_m-1}^{(x_1)_z}) = (\gamma_{\text{len}(x_1)+1}^{x_z}, \gamma_{\text{len}(x_1)+2}^{x_z}, \ldots, \gamma_{k_m}^{x_z})$.

It is easy to see that

$$(x, \gamma^z_{\text{len}(x)+1}, \gamma^z_{\text{len}(x)+2}, \ldots, \gamma^z_{k_m}) = (x_1, \gamma^z_{\text{len}(x_1)+1}, \gamma^z_{\text{len}(x_1)+2}, \ldots, \gamma^z_{k_m}),$$

which implies that $x$ adding the first $k_m - \text{len}(x)$ steps of CoT is equal to $x_1$ adding the first $k_m - \text{len}(x_1)$ steps CoT, according to the definition of CoT-transformer, there must be $\widehat{F}_{\text{cot}}(x_1) = \widehat{F}_{\text{cot}}(x)$. $\qquad\square$

Second, we prove the necessity of the condition.

*Proof.* We will show that if the conditions are not satisfied, then such a language cannot be memorized by any CoT-transformer.

**Part One.**

Firstly, we show that when $S_x = \phi$ for some $(x, y) \in S$, $S$ cannot be memorized by any CoT-transformer.

If not, let $S$ be memorized by a CoT-transformer $\mathcal{F}$. Then we can obtain a CoT for such an $x$ and write the CoT as $(x, \gamma_{i_1}, \gamma_{i_2}, \ldots, \gamma_{i_n}, y, \gamma_0)$.

Then we prove that $(x, \gamma_{i_1}, \gamma_{i_2}, \ldots, \gamma_{i_n}, y) \in S_x$, which is in contradiction to $S_x = \phi$ and thus prove the result.

We just need to verify that (1), (2), (3) in the definition of $S_x$ are correct.

First, it is easy to see that (1) in the definition of $S_x$ is correct.

For (2), if (2) is not correct, then $|\text{typ}(x, \gamma_{i_1})| = 1$. By Proposition 4.3, we know that $\widehat{F}((x, \gamma_{i_1})) = \widehat{F}(x) = \gamma_{i_1}$, so $\gamma_{i_2} = \gamma_{i_1}$. Similar to any $\gamma_{i_k}$ where $k \in [n]$. So we have $y = \gamma_{i_1} = \mathcal{F}((x, \gamma_{i_1}, \gamma_{i_2}, \ldots, \gamma_{i_n})) = \mathcal{F}((x, \gamma_{i_1}, \gamma_{i_2}, \ldots, \gamma_{i_n}, y)) = \gamma_0$, but based on the definition of $\gamma_0$, there must be $\gamma_0 \neq y$, which is contradictory.

For (3), if for some $(x_1, y_1) \in S$ such that $x_1 = (x, \gamma_{i_1}, \gamma_{i_2}, \ldots, \gamma_{i_m})$ for some $m \geq 1$, then by the definition of CoT-transformer, we have $\widehat{F}(x_1) = \widehat{F}(x)$, so there must be $y_1 = y$. So we prove the result.

**Part two.**

We show that if $\cap_{(x,y) \in S:\text{typ}(x)=\{\gamma_k\}} S^1_x = \phi$ for some $\gamma_k \in \Gamma$ which satisfies $\{(x, y) \in S : \text{typ}(x) = \{\gamma_k\}\} \neq \phi$, Then $S$ cannot be memorized by a CoT-transformer.

By Proposition 4.3, we know that for any sentences $(x_1, y_1), (x_2, y_2) \in S$ such that $\text{typ}(x_1) = \text{typ}(x_2) = \{\gamma_k\}$, the output of $\mathcal{F}(x_1)$ and $\mathcal{F}(x_2)$ are the same, which implies that if $S$ can be memorized by a CoT-transformer, the first symbol in CoT is the same for $\mathcal{F}$ when input $x_1$ or $x_2$. Considering the arbitrariness of $x_1$ and $x_2$, the above result implies that $\cap_{(x,y) \in S:\text{typ}(x)=\{\gamma_k\}} S^1_x \neq \emptyset$, so we can obtain the result. $\qquad\square$

### D.2 Proof of Proposition 4.5

*Proof.* We first prove (1) in the proposition. We show that if the set of the last elements of all sentences in $S$ is a proper subset of $\Gamma$, that is, $\{x[\text{len}(x)] : (x, y) \in S\} \subsetneq \Gamma$, then $S$ satisfies the conditions in Theorem 4.4.

**Part 1.1.** We need a simple result: Let $\gamma_i \in S/\{x[\text{len}(x)] : (x, y) \in S\}$, and $\gamma_i(L) = (\gamma_i, \gamma_i, \gamma_i \ldots, \gamma_i)$ be a sentence with length $L$ and all symbols in it are $\gamma_i$. Then for any $(x, y) \in S$ and $z \in \Gamma^+$, there must be $x_z = (x, \gamma_i(L), z, y) \in S_x$.

To prove the above result, we just need to verify the three conditions in the definition of $S_x$.

Condition (1) in the definition of $S_x$ is clearly valid.

For condition (2) in the definition of $S_x$, because $\gamma_i \notin \{x[\text{len}(x)] : (x, y) \in S\}$, so $(x_z)_{[len(x)+1]} > 1$.

For condition (3) in the definition of $S_x$. If $\text{len}(x_1) > \text{len}(x)$, considering that $\text{len}(x_1) \leq L$, the last symbol in $(x_z)_{[\text{len}(x_1)]}$ is $\gamma_i$ which must not be the last symbol in $x_1$. So $(x_z)_{[\text{len}(x_1)]} \neq x_1$ for any $\text{len}(x_1) > \text{len}(x)$, implying condition (3).

**Part 1.2.** Now we can prove (1) in the proposition. We just need to verify the conditions in Theorem 4.4.

Firstly, by the result in Part 1.1, we know that $S_x \neq \phi$ for any $(x, y) \in S$.

Secondly, for any $\gamma_j$ such that $\{(x, y) \in S : \mathrm{typ}(x) = \gamma_j\} \neq \emptyset$, we have $\gamma_j \in \{x[\mathrm{len}(x)] : (x, y) \in S\}$, so $j \neq i$ where $i$ is defined in Part 1.1. Then we know that $\gamma_i \in \cap_{(x,y)\in S, \mathrm{typ}(x)=\gamma_j} S_x^1$ by Part 1.1. This proves (1).

We now prove (2) in the proposition. We show that if $\mathrm{len}(x) = L$ for all $(x, y) \in S$, then $S$ satisfies the conditions in Theorem 4.4.

**Part 2.1.** We need a simple result: for any $(x, y) \in S$, if $\gamma_i \neq x[\mathrm{len}(x)]$, then for any $z \in \Gamma^+$, there must be $x_z = (x, \gamma_i, z, y) \in S_x$. In fact, by definition of $S_x$, this is obvious.

**Part 2.2.** Now we can prove (2) in the proposition. We just need to verify the conditions in Theorem 4.4.

Firstly, by the result in Part 2.1, we know that $S_x \neq \phi$ for any $(x, y) \in S$.

Secondly, for any $\gamma_j$ such that $\{(x, y) \in S : \mathrm{typ}(x) = \gamma_j\} \neq \emptyset$, we just need to take $i \neq j$, then we know that $\gamma_i \in \cap_{(x,y)\in S, \mathrm{typ}(x)=\gamma_j} S_x^1$ by Part 2.1. This proves (2). $\qquad\square$

# E   Proofs of results in Section 4.3

## E.1   Proof for Proposition 4.7

We just need to verify the conditions in Theorems 4.1 and 4.4 for language LCP.

*Proof.* It is easy to see that (2) and (3) in Proposition 4.7 can directly lead to (1), so we just need to prove (2) and (3) of Proposition 4.7.

### For no-CoT-transformer.

For $\mathrm{LCP}_n^{=1}$, we consider the sentence $x_i = (\gamma_1, \gamma_1, \ldots, \gamma_1)$ where $\mathrm{len}(x_i) = i$. It is easy to see that $x_i$ and $x_{i+1}$ have different labels by the definition of LCP, but $\mathrm{typ}(x_i) = \mathrm{typ}(x_{i+1}) = \{\gamma_1\}$. So $\mathrm{LCP}_n^{=1}$ does not satisfy the conditions in Theorem 4.1, and cannot be memorized by no-CoT-transformers.

For $\mathrm{LCP}_n^{>1}$. It is easy to see that $(x, y) \in \mathrm{LCP}_n^{>1}$ implies $|\mathrm{typ}(x)| > 2$, so $\mathrm{LCP}_n^{>1}$ satisfies the conditions in Theorem 4.1, and can be memorized by no-CoT-transformers.

### For CoT-transformer.

For $\mathrm{LCP}_n^{=1}$ and $(x, y) \in \mathrm{LCP}_n^{=1}$, let $\mathrm{typ}(x) = \gamma_x$. It is easy to check that $(x, \gamma_i, x_{cot}, y) \in S_x$ for any $\gamma_i \neq \gamma_x$ and $x_{cot} \in \Gamma^+$. So $\mathrm{LCP}_n^{=1}$ satisfies the conditions in Theorem 4.4, and can be memorized by CoT-transformers.

For $\mathrm{LCP}_n^{>1}$ and $(x, y) \in \mathrm{LCP}_n^{>1}$ such that $\mathrm{len}(x) < n$. If $x_z = (x, \gamma_i, x_{cot}, y) \in S_x$ for some $\gamma_i \in \Gamma$ and $x_{cot} \in \Gamma^+$, then by the fact $((x, \gamma_i), y_1) \in \mathrm{LCP}_n$, we have $(x_z)_{[\mathrm{len}(x)]+1} = (x, \gamma_i)$ but $y_1 \neq y$, which is contradictory with the definition of $S_x$. So $\mathrm{LCP}_n^{>1}$ does not satisfy the conditions in Theorem 4.4 and cannot be memorized by the CoT-transformer. $\qquad\square$

## E.2   Proof of Proposition 4.8

This proof also follows the proof of Theorem 4.1.

*Proof.* The proof has five parts. And we still define $s_{x,i}$ as that in the proof of Theorem 4.1.

### Part One: The Embedding.

We will embed in the following way:

The basic symbols $\gamma_i$ will embed in $(v_1^i, v_2^i, \ldots, v_T^i, 0) \in \{0, 1\}^{4T+\lceil \log_2 L \rceil}$ where $v_j^i = (0, 0, 1, 1)$ when $j \neq i$, and $v_i^i = (0, 1, 0, 1)$.

The position embedding for $n$-th position is $(0, 0, \ldots, 0, n_2) \in \{0,1\}^{4T+\lceil \log_2 L \rceil}$, where $n_2 \in \{0,1\}^{\lceil \log_2 L \rceil}$ is the binary representation of $n$, such as when $n = 11$, the $n_2 = (0, 0, \ldots, 0, 1, 0, 1, 1)$.

Hence, the input sentence $x$ satisfying $\mathrm{len}(x) = n$ will be embedded in a vector $V_x \in \{0,1\}^{n \times (4T+\lceil \log_2 L \rceil)}$.

**Parts Two, Three, Four.**

In these three parts, we perform the operation on the first $4T$ columns in $V_x$ the same way as that in the proof of Theorem 4.1, keeping the value of the last $\lceil \log_2 L \rceil$ columns unchanged throughout these parts.

Similar to that in Part Four of the proof of Theorem 4.1, we show that for any $(x, y_x), (z, y_z)$, we can obtain a different vector for $x$ and $z$.

If $x$ and $z$ do not satisfy $\mathrm{typ}(x) = \mathrm{typ}(z) = \{\gamma_l\}$ for some $l \in [T]$, then we consider the first $4T$ columns and follow the proof in part four in the proof of Theorem 4.1.

If $x$ and $z$ satisfy $\mathrm{typ}(x) = \mathrm{typ}(z) = \{\gamma_l\}$ for some $l \in [T]$, then there must be $\mathrm{len}(x) \neq \mathrm{len}(z)$, so we consider the last $\lceil \log_2 L \rceil$-th columns. Based on the definition of the position embedding, we know that the $\lceil \log_2 L \rceil$-th columns in the last row of the embedding matrix of $x$ and $z$ are different, and the $L_\infty$ norm of the difference between them is at least 1. Since the last $\lceil \log_2 L \rceil$-th columns are unchanged throughout these parts, we obtain the result.

**Part Five.**

Quite similar to Part Five in the proof of Theorem 4.1. □

### E.3 Proof for Proposition 4.10

*Proof.* We will show that, if $|\mathrm{typ}(x)| > 1$ for any $(x, y) \in S$ and the conditions in Theorem 4.4 are not satisfied, then such a language cannot be memorized by any CoT-transformer with position encoding.

Because $|\mathrm{typ}(x)| > 1$ for any $(x, y) \in S$, we need only to show that if $S_x = \phi$ for some $(x, y) \in S$, then $S$ cannot be memorized by any CoT-transformer with position encoding.

If not, assume that $S$ can be memorized by a CoT-transformer $\mathcal{F}$ with position encoding. Then we can obtain a CoT for such $x$, written as $(x, \gamma_{i_1}, \gamma_{i_2}, \ldots, \gamma_{i_n}, y, \gamma_0)$.

Then we prove that $(x, \gamma_{i_1}, \gamma_{i_2}, \ldots, \gamma_{i_n}, y) \in S_x$, which is a contradiction to $S_x = \phi$, and we establish the result.

We just need to verify (1), (2), (3) in the definition of $S_x$ are true.

First, it is easy to see that (1) in the definition of $S_x$ is true.

For (2), because $|\mathrm{typ}(x)| > 1$, it obviously stands.

For (3), for some $(x_1, y_1) \in S$ such that $x_1 = (x, \gamma_{i_1}, \gamma_{i_2}, \ldots, \gamma_{i_m})$ for some $n \geq m \geq 1$, even with position encoding, we have $\mathcal{F}(x_1) = F((x, \gamma_{i_1}, \gamma_{i_2}, \ldots, \gamma_{i_m}))$. Therefore, by the definition of CoT-transformer, we have $\widehat{F}(x_1) = \widehat{F}(x)$. Thus, there must be $y_1 = y$. So we prove the result. □

## F  Proofs of results in Section 5

### F.1  Proof for Theorem 5.1

*Proof.* Let $L_N = [\log_T N] + 1$. Then $T^{L_N} \geq N$, since $N, T \geq 3$. Therefore, $[\log_T N] + 1 \leq 2 \ln N$. We can arbitrarily select the $N$ sentences with length $L_N$. By Corollary 4.2 and Proposition 4.5, we know that the language composed of these sentences with arbitrary labels must meet the conditions of Theorem 4.1 and Theorem 4.4. It is easy to see that, for these $N$ sentences, there exist $T^N$ different situations to assign labels to them. Hence, $T^N$ different languages can be created by these $N$ sentences. We will show that at least one of these languages requires a no-CoT-transformer (CoT-transformer) with at least $\frac{N \ln T}{6q}$ parameters to memorize it, which is what we want.

It is easy to see that for a transformer with no more than $P$ parameters, its width, head, and depth are all smaller than $P$. Therefore, for any $T \geq 3$, there exist at most $P^3$ pairs of $W, D, H$ such that $\mathrm{para}(W, D, H, T) \leq P$. Hence, for any $W, D, H$ such that $\mathrm{para}(W, D, H, T) \leq P$, considering that each parameter has precision $q$, we have that each parameter has at most $10^{2q}$ different choices. So, there exist at most $(10^{2q})^P$ different transformers in $H^q_{W,D,H}$ (or $H^q_{W,D,H,\mathrm{cot}}$ for CoT transformer). So, there exist at most $P^3(10^{2q})^P$ situations for a no-CoT transformer (or CoT transformer) with $P$ parameters and precision $q$.

So to memorize all these $T^N$ languages created by such $N$ sentences, at least $T^N$ different transformers are required. So, if each of these languages can be memorized by a no-CoT-transformer (CoT-transformer) with no more than $P$ parameters, there must be $T^N \leq P^3(10^{2q})^P$, which implies $N \ln T \leq 3 \ln P + 2qP \ln 10 \leq 6qP$ ($q \geq 3$ is used here). The theorem is proved. $\qquad\square$

### F.2   Proof of Proposition 5.3

#### F.2.1   Proof of Proposition 5.3 for no-CoT transformers

In this section, we provide the proof of Proposition 5.3 for no-CoT transformers.

We present an easy lemma first.

**Lemma F.1.** *If* $\{a_i\}_{i=1}^n, \{b_i\}_{i=1}^n \subset \mathbb{R}_+$ *satisfy that:*

*(1) for any $i \neq j$, we have $|a_i - a_j| \geq 1$, $|b_i - b_j| \geq 1$ and*

*(2) for any $i, j \in [n]$, we have $|b_i - a_j| \geq 1$ when $a_i \neq b_j$.*

*Then it holds* $\left| \frac{1}{\sum_{i=1}^n e^{a_i}} - \frac{1}{\sum_{i=1}^n e^{b_i}} \right| \geq \frac{1}{(\sum_{i=1}^n e^{a_i})(\sum_{i=1}^n e^{b_i})}$.

*Proof.* Just need to prove that $\sum_{i=1}^n e^{a_i} - \sum_{i=1}^n e^{b_i} > 1$. Without loss of generality, let $a_i \geq a_{i+1}$ and $b_i \geq b_{i+1}$. Assume $k \in [n]$ is the minimum such that $a_k \neq b_k$, and let $a_k > b_k$. Then we have that: $\sum_{i=1}^n e^{a_i} - \sum_{i=1}^n e^{b_i} \geq e^{a_k} - \sum_{i=k}^n e^{b_i} \geq e^{a_k} - e^{b_k - n + k + 1} \frac{e^{n-k} - 1}{e - 1} > e^{a_k}(1 - 1/(e - 1)) > e - e/(e - 1) > 1$, which is what we want. $\qquad\square$

We now prove Proposition 5.3 for no-CoT transformers.

*Proof.* We follow the proof of Theorem 4.1. And we still define $s_{x,i}$ as that in the proof of Theorem 4.1.

**Part One: Embedding**

This is the same as Part One in the proof of Theorem 4.1.

**Part Two: Use the three layers to calculate the position value.**

This is the same as Part two in the proof of Theorem 4.1. But because there exists no precision limitation for the transformer, we just need three layers.

**Part Three.**

Define the set $V_j = \{V_a^{10^\alpha}(s_{x,i}, j)\}_{(x,y)\in S, i \in [T]}$ where $j \in [L]$ and $V = \cup_{j \in [L]} V_j$. Let $A \in \mathbb{R}_+$ satisfy $|Ax - Az| > 1$ for any $x, z \in V$.

In this proof, we define matrix $M_3(x)$ which has the same size as $M_2(x)$ as: the $(4i - 3)$-th column of $M_3$ is $A(M_2(x))_{4i-2}$, where $(M_2(x))_{4i-2}$ is the $(4i - 2)$-th column of $M_2(x)$; other columns are the same as $M_2(x)$. It is easy to see that a hidden layer with width $O(1)$ is enough to calculate $M_3(x)$ by $M_2(x)$.

Next, we use an FNN $f_1 : \mathbb{R} \to \mathbb{R}$ to map 0 and 1 to 1, but other values in $V$ to 0. It is easy to see that such a network $f_1$ need only the $O(1)$ layers and $O(1)$ width.

Based on such FNN $f_1$ and $M_3(x)$, we define a matrix $M_4(x)$ which has the same size as $M_3(x)$ as: the $(4i - 1)$-th column of $M_4(x)$ is $f_1((M_3)_{4i-2})$, where $(M_3)_{4i-2}$ is the $(4i - 2)$-th column of $M_3(x)$; other columns are the same as $M_3(x)$.

Then by Lemma B.2, we can use a transformer with width $O(1)$ and depth $O(1)$ to obtain $M_4(x)$ by $M_3(x)$.

**Part Four.**

In this part, we use two hidden layers to obtain a vector by $M_4(x)$, which is different from $x$.

In this layer, this attention layer has $T$ heads, and in the $i$-th head, $Q_i \in \mathbb{R}^{4T \times 4T}$ and the $(4i, 4i-3)$-th weight of $Q_i$ is 1, and others are 0; $K_i = I$ and $V_i \in \{0,1\}^{4T \times 4T}$ and the $(4i-1, 4i-1)$-th weights of $V_i$ are 1 and others are 0.

The residual layer uses $W \in \{0,1\}^{4T \times 4T}$ which is defined as: for $i \in [T]$, $(4i-2, 4i-2)$-th weights of $W$ is 1, others are 0. The FNN layer is 0.

Let the last row for the output of this layer be $M_l(x)$ when input $x$ with length $n$ to the transformer, then based on the definition of $M_4(x)$, we have that:

(1) The $(4i-2)$-th weight of $M_l(x)$ is $V_a^{10^\alpha}(s_{x,i}, n)$.

(2) The $(4i-1)$-th weight of $M_l(x)$ is $\frac{\sum_{j=1}^n e^{AV_a^{10^\alpha}(s_{x,i},j)} f_1(V_a^{10^\alpha}(s_{x,i},j))}{\sum_{j=1}^n e^{AV_a^{10^\alpha}(s_{x,i},j)}}$.

We will prove that for all $(x,y), (z,y_z) \in S$ that do not satisfy $\mathrm{typ}(x) = \mathrm{typ}(z) = \gamma_l$ for some $l \in [T]$, we have $||M_l(x) - M_l(z)||_\infty > 0$.

When $\mathrm{typ}(x) = \mathrm{typ}(z)$, the proof is similar to the proof of Theorem 4.1.

When $\mathrm{len}(x) \neq \mathrm{len}(z)$ and $\mathrm{typ}(x) > 2$, the proof is similar to the proof of Theorem 4.1.

When $\mathrm{len}(x) = \mathrm{len}(z)$ and $\mathrm{typ}(x) > 2$, let $i \in [T]$ satisfy $s_{x,i} \neq s_{z,i}$. If $V_a^{10^\alpha}(s_{x,i}, \mathrm{len}(x)) \neq V_a^{10^\alpha}(s_{z,i}, \mathrm{len}(z))$, then we have the same result as before; if not, by Lemma C.2, we know that there exist the same number of 1 or 0 in $\{V_a^{10^\alpha}(s_{x,i},j)\}_{j=1}^{\mathrm{len}(z)}$ and $\{V_a^{10^\alpha}(s_{z,i},j)\}_{j=1}^{\mathrm{len}(z)}$. Based on the definition of $f_1$, we know that

$$\sum_{j=1}^n e^{AV_a^{10^\alpha}(s_{x,i},j)} f_1(V_a^{10^\alpha}(s_{x,i},j)) = \sum_{j=1}^n e^{AV_a^{10^\alpha}(s_{z,i},j)} f_1(V_a^{10^\alpha}(s_{z,i},j)) \neq 0.$$

Hence, based on Lemmas F.1 C.1 and C.2, and considering the definition of $A$, we know that $|\frac{1}{\sum_{j=1}^n e^{AV_a^{10^\alpha}(s_{x,i},j)}} - \frac{1}{\sum_{j=1}^n e^{AV_a^{10^\alpha}(s_{z,i},j)}}| > 0$, so we prove the result.

**Part Five.** Quite similar to Part Five in the proof of Theorem 4.1. But because there exists no precision limit on the transformer, we just need $O(N)$ layers as shown in Lemma C.8. $\square$

### F.2.2 Proof of Proposition 5.3 for CoT transformers

In this section, we give the proof of Proposition 5.3 for CoT transformers.

*Proof.* The proof is based on the proof of Theorem 4.4.

**Part One.**

We need only to change the definition of $S_{ss}$.

For each $(x,y) \in S$, we put an $x_z \in S_x$ satisfying the following conditions into set $S_{ss} = \{x_z : (x,y) \in S\}$:

(c1) for any $k$, the symbols $x_z[\mathrm{len}(x)+1]$ are the same for all $(x,y) \in S$ such that $\mathrm{typ}(x) = \{\gamma_k\}$;

(c2) $\mathrm{len}(x_z) - \mathrm{len}(x) \leq N+1$.

We just need to consider (c2).

For a $(x,y) \in S$ such that $|\mathrm{typ}(x)| > 1$. Let $x_z \in S_x$ and $\mathrm{len}(x_z)$ be the minimum. If $\mathrm{len}(x_z) - \mathrm{len}(x) > N$, then there exists a $\mathrm{len}(x) < l < \mathrm{len}(z)$ such that there exists no $(z, y_z) \in S$ such that $y_z \neq y$ and $\mathrm{len}(z) = l$.

So we consider a sentence $x_l = ((x_z)_{[l-1]}, y)$. It is easy to check that $x_l \in S_x$ and $\mathrm{len}(x_l) < \mathrm{len}(x_z)$ which are contradictory to the minimum of $\mathrm{len}(x_z)$. So $\mathrm{len}(x_z) - \mathrm{len}(x) \leq N$.

For a $(x, y) \in S$ such that $\mathrm{typ}(x) = \{\gamma_i\}$. By the condition of Theorem 4.4, let $\gamma_j \in \bigcap_{(x,y) \in S, \mathrm{typ}(x) = \{\gamma_i\}} S_x^1$.

Then we let $x_z \in S_x$ satisfy that $x_z[\mathrm{len}(x) + 1] = \gamma_j$ and $\mathrm{len}(x_z)$ be the minimum. Similar to the above, we can show that $\mathrm{len}(x_z) - (\mathrm{len}(x) + 1) \leq N$. So we get the result.

**Parts two and three.** These two parts are similar to the proof of Theorem 4.4. Considering that the $S_n$ constructed by $S_{ss}$ defined in Part One has at most $O(N^2)$ samples in it, and using the result in Section F.2.1, we prove the result. $\qquad\square$

### F.3 Proof of Theorem 5.4

#### F.3.1 Proof of Theorem 5.4 for no-CoT-transformers

Firstly, we define the following language.

Let $\Gamma = \{\gamma_1, \gamma_2\}$, and $S_i$ be a sub-language of LCP with 2 samples $(x_j^i, y_j^i)$ where $j \in \{1, 2\}$ in it. We define that $x_1^i$ is a sentence with length $i + 1$, and the $i$-th element of $x_1^i$ is $\gamma_2$, the other elements are $\gamma_1$; the $x_2^i$ only contains $\gamma_1$ with length $i$; the labels are decided by LCP.

Now we can prove Theorem 5.4 for no-CoT-transformers, we show that for any $q$ and $P$, there exists an $i$ such that $S_i$ cannot be memorized by any transformer in $H_{P,P,P}^q$.

*Proof.* Assuming that for a pair of $P, q$, each $i \in \mathbb{Z}_+$, $S_i$ can be memorized by a non-CoT transformer in $H_{P,P,P}^q$, we can derive contradictions.

**First,** let $\mathcal{F}(x)_i$ be the $i$-th weight of $\mathcal{F}(x)$, then let $\min_{F \in H_{P,P,P}^q} |I(F(\gamma_1)_1 = F(\gamma_1)_2) + (F(\gamma_1)_1 - F(\gamma_1)_2)| = \epsilon$. Following the proof of Theorem 5.1, we know that there exist finite varieties of different transformers in $H_{P,P,P}^q$, so $\epsilon > 0$.

If $S_i$ is memorized by $\mathcal{F} \in H_{P,P,P}^q$, then by Proposition 4.3, we have $\mathcal{F}(x_2^i) = F(\gamma_1)$ for any $i \in \mathbb{Z}_+$, so $|\mathcal{F}(x_2^i)_1 - \mathcal{F}(x_2^i)_2| = |F(\gamma_1)_1 - F(\gamma_1)_2| \geq \epsilon$.

**We will prove that for any given transformer $\mathcal{F} \in H_{P,P,P}^q$, it holds $||F(x_1^i) - F(x_2^i)||_1 \to 0$ when $i \to \infty$.**

If $||\mathcal{F}(x_1^i) - \mathcal{F}(x_2^i)||_1 \to 0$ when $i \to \infty$ for any given $\mathcal{F} \in H_{P,P,P}^q$, then because there exist finite varieties of different transformers in $H_{P,P,P}^q$. So there exists an $i$ satisfying that $||\mathcal{F}(x_1^i) - \mathcal{F}(x_2^i)||_1 < \epsilon/3$ for any $\mathcal{F} \in H_{P,P,P}^q$, which implies

$$\mathcal{F}(x_1^i)_1 - \mathcal{F}(x_1^i)_2 \geq \mathcal{F}(x_2^i)_1 - |\mathcal{F}(x_2^i)_1 - \mathcal{F}(x_1^i)_1| - (\mathcal{F}(x_2^i)_2 + |\mathcal{F}(x_2^i)_2 - \mathcal{F}(x_1^i)_2|)$$
$$\geq \mathcal{F}(x_2^i)_1 - \mathcal{F}(x_2^i)_2 - 2\epsilon/3.$$

Similarly, it holds $\mathcal{F}(x_2^i)_1 - \mathcal{F}(x_2^i)_2 + 2\epsilon/3 \geq \mathcal{F}(x_1^i)_1 - \mathcal{F}(x_1^i)_2$. Since $|\mathcal{F}(x_2^i)_1 - \mathcal{F}(x_2^i)_2| \geq \epsilon$, any $\mathcal{F} \in H_{P,P,P}^q$ will give the same label of $x_1^i$ and $x_2^i$. Considering that $x_2^i$ and $x_1^i$ have different lengths, by the definition of LCP language, they have different labels. This is contradictory to the assumption and directly gets the result we want.

To show that, we need three parts, assume $\mathcal{F}$ is a transformer in $H_{P,P,P}^q$. Let $||\cdot||_{1,\infty}$ be the maximum $L_1$ norm of the row in a matrix.

**Part One: For any $j$, in the $j$-th hidden layer $\mathcal{F}_j$ of $\mathcal{F}$, the first $i - 1$ rows of $\mathcal{F}_j(x_1^i)$ and $\mathcal{F}_j(x_2^i)$ are equal to that $j$-th hidden layer in $\mathcal{F}(\gamma_1)$.**

This can be proved by using Lemma B.1 and Proposition 4.3.

**Part Two: For any $i, j$ and $x \in S_i$, in the $j$-th hidden layer $\mathcal{F}_j$ of $\mathcal{F}$, $||F_j(x)||_{1,\infty} \leq M_{P,q,j}$, where $M_{P,q,j}$ is a value that does not rely on $i$ but only depends on $P, q, j$.**

In the $j$-th hidden layer, if the input $z$ of the $j$-th hidden layer, which is also the output of the $(i-1)$-th hidden layer, satisfies that each row has the $L_1$ norm not more than $A$, then we can show that the output norm of the $j$-th hidden layer also depends only on $A$ and $P, q$. To show that, we just need to consider the attention layer, FNN and the residual layer respectively.

In the attention layer $\text{ATT}(z) = \sum_{i=1}^{P} \text{softmax}(zQ_iV_iz^T + M)zK_i$, since the $L_1$ norm of each row of $\text{softmax}(zQ_iV_iz^T + M)$ is 1, and we limited the precision of the transformer, it holds that each parameter in $K_i$ is not more than $10^q$. So we can show that each row of $\text{ATT}(z)$ will have a $L_1$ norm no more than $P(A \times P^2 \times 10^q)$.

Upon the residual layer $zw$, similar to before, each row has a $L_1$ norm of not more than $A \times P^2 \times 10^q$.

In the FNN layer $\text{FNN}(z + \text{ATT}(z))$, each row of $Z + \text{ATT}(z)$ goes through two transformer matrices, so the $L_1$ norm of each row has at most $O((P^2 10^q)^2 \|z + \text{ATT}(z)\|_{1,\infty})$. $\|\cdot\|_{1,\infty}$ means the maximum $L_1$ norm of the row in a matrix.

Adding them, the $L_1$ norm for each row of the output of the $j$-th hidden layer is at most $O(P(10^q P^2)^3 A)$.

If we take $A = M_{P,q,j-1}$, then we know that $M_{P,q,j} \leq O(P(10^q P^2)^3 M_{P,q,j-1})$. Considering that the input of the first hidden layer is only bound by $P$ and $q$, we have the result.

**Part Three: For any $j$, in the $j$-th hidden layer $\mathcal{F}^j$ of $\mathcal{F}$, $\|(F^j(x_1^i))_{i+1} - F^j(x_2^i)_i\| \to 0$ when $i \to \infty$. This directly leads to our result.**

To be convenient, we write $\mathcal{F}^j(x_1^i)$ as $f(j,1,i)$ and $\mathcal{F}^j(x_2^i)$ as $f(j,2,i)$, and $f^k(*,*,*)$ means the $k$-row of $f(*,*,*)$. And let $\|f^{i+1}(j-1,1,i) - f^i(j-1,2,i)\|_1 = \eta(j-1,i)$

For convenience, we define the input of the first hidden layer as the output of the 0-th hidden layer (i.e., $j = 0$ here. Consider that for any $i \in \mathbb{Z}_+$, the $i + 1$ row of $x_i^1$ and the $i$ row of $x_i^2$ are the same, so $\eta(0,i) = 0$ for the first hidden layer. We will prove that, if $\eta(j-1,i)$ satisfies $\eta(j-1,i) = 0$ when $i \to \infty$, then $\eta(j,i)$ also tends to 0 when $i \to \infty$.

Note that $f(j,1,i)$ can be calculated as: $f(j-1,1,i)W_{j-1} + \text{ATT}(f(j-1,1,i)) + \text{FNN}(f(j-1,1,i)W_{j-1} + \text{ATT}(f(j-1,1,i)))$.

Firstly, for the residual layer, we have $\|(f^{i+1}(j-1,1,i)-f^i(j-1,2,i))W_{j-1}\|_1 \leq 10^q P^2 \|f^{i+1}(j-1,1,i) - f^i(j-1,2,i)\|_1 = 10^q P^2 \eta(j-1,i)$. Based on the assumption of $\eta(j-1,i)$, when $i \to \infty$, it holds $\|(f^{i+1}(j-1,1,i) - f^i(j-1,2,i))W_{j-1}\|_1 \to 0$.

Secondly, by part one and Proposition 4.3, the first $i - 1$ rows of $f(j-1,1,i)$ and all rows of $f(j-1,2,i)$ are the same as $\mathcal{F}^{j-1}(\gamma_1)$. So in a head of attention layer in $\mathcal{F}_j$, written as $\text{softmax}(XQVX^T)XK$, we have that:

$$(\text{softmax}(f(j-1,1,i)QVf(j-1,1,i)^T + M)f(j-1,1,i))_{i+1}$$
$$= \frac{(i-1)e^s f^i(j-1,2,i)}{(i-1)e^s+e^v+e^w} + \frac{e^v f^i(j-1,1,i)}{(i-1)e^s+e^v+e^w} + \frac{e^w f^{i+1}(j-1,1,i)}{(i-1)e^s+e^v+e^w},$$

where $s = f^{i+1}(j-1,1,i)QV(f^{i-1}(j-1,1,i))^T, v = f^{i+1}(j-1,1,i)QV(f^i(j-1,1,i))^T, w = f^{i+1}(j-1,1,i)QV(f^{i+1}(j-1,1,i))^T$.

Considering that $(\text{softmax}(f(j-1,2,i)Q_iV_if(j-1,2,i)^T)f(j-1,2,i))_i = f^i(j-1,2,i)$, let $-(\text{softmax}(f(j-1,2,i)QVf(j-1,2,i)^T)f(j-1,2,i))_i + (\text{softmax}(f(j-1,1,i)QVf(j-1,1,i)^T)f(j-1,1,i))_{i+1} = c(i)$, we have that:

$$c(i) = \frac{(i-1)e^s f^i(j-1,2,i)}{(i-1)e^s+e^v+e^w} + \frac{e^v f^i(j-1,1,i)}{(i-1)e^s+e^v+e^w} + \frac{e^w f^{i+1}(j-1,1,i)}{(i-1)e^s+e^v+e^w} - f^i(j-1,2,i)$$
$$= \frac{-(e^v+e^w) f^i(j-1,2,i)}{(i-1)e^s+e^v+e^w} + \frac{e^v f^i(j-1,1,i)}{(i-1)e^s+e^v+e^w} + \frac{e^w f^{i+1}(j-1,1,i)}{(i-1)e^s+e^v+e^w}$$

Consider that the transformer has precision $q$ and part two, so we have $\frac{e^v+e^w}{(i-1)e^s+e^v+e^w} \leq \frac{2e^{M_{P,q,j-1}^2 P^2 10^{2q}}}{2e^{M_{P,q,j-1}^2 P^2 10^{2q}} + (i-1)e^{-M_{P,q,j-1}^2 P^2 10^{2q}}}$, similar to $\frac{e^v}{(i-1)e^s+e^v+e^w}$ and $\frac{e^w}{(i-1)e^s+e^v+e^w}$. Hence, by part two, we have that $\|c(i)\|_1 \leq O(M_{P,q,j-1}\frac{e^{P^2 10^{2q} M_{P,q,j-1}^2}}{2e^{P^2 10^{2q} M_{P,q,j-1}^2} + (i-1)e^{-P^2 10^{2q} M_{P,q,j-1}^2}})$, which tends to 0 when $i \to \infty$. And for the whole attention layer with $P$ heads and matrix $K$, we have $(\text{ATT}(f(j-1,1,i)))_{i+1} - (\text{ATT}((j-1,2,i)))_i \leq 10^q P^3 \|c(i)\|_1$, which is a value that tends to 0 when $i \to \infty$.

Finally, about the FNN layer, similar to Part two, for any given two vectors $x_1$ and $x_2$, we have that $\text{FNN}(x_1) - \text{FNN}(x_2) \leq O((10^q p^2)^2 \|x_1 - x_2\|_1)$. Consider that we have proved that $\|f^{i+1}(j-$

$1, 1, i)W_{j-1} - f^i(j-1, 2, i)W_{j-1}||_1 \to 0$ and $(\text{ATT}(f(j-1, 1, i)))_{i+1} - (\text{ATT}((j-1, 2, i)))_i \to 0$ when $i \to \infty$, so it holds $\text{FNN}(f^{i+1}(j-1, 1, i)W_{j-1} + (\text{ATT}(f(j-1, 1, i)))_{i+1}) - \text{FNN}(f^i(j-1, 2, i)W_{j-1} + (\text{ATT}(f(j-1, 2, i)))_i) \to 0$ when $i \to \infty$.

Adding the results of such three parts in the $j$-th hidden layer, we can obtain the result. □

### F.3.2 Proof of Theorem 5.4 for CoT-transformers

For the CoT-transformer, we consider the following example.

Let $\Gamma = \{\gamma_1, \gamma_2, \gamma_3, \gamma_4\}$, and $S_i'$ a sub-language of LCP with 10 samples $(x_j^i, y_j^i)$ where $j \in [10]$ in it. We define that:

(1) $x_1^i$ is a sentence with length $i$ and all elements in $x_1^i$ are $\gamma_1$;

(2) $x_j^i$ $(j = 2, 3, 4)$ is a sentence of length $i + 1$ and the first $i$ elements of $x_1^i$ are $\gamma_1$, the last one is $\gamma_{j-1}$;

(3) $x_j^i$ $(j = 5, 6, 7)$ is a sentence of length $i + 2$ and the first $i$ elements of $x_1^i$ are $\gamma_1$, the $(i+1)$-th element is $\gamma_4$, and the last is $\gamma_{j-3}$;

(4) $x_j^i$ $(j = 8, 9, 10)$ is a sentence with length $i + 3$ and the first $i$ elements of $x_1^i$ are $\gamma_1$, the $(i+1)$-th element is $\gamma_4$, the $(i+2)$-th element is $\gamma_1$, and the last one is $\{\gamma_1, \gamma_2, \gamma_4\}$, respectively.

Their labels are decided by LCP. Then we can prove Theorem 5.4 for CoT-transformers. We show that for any $q$ and $P$, there exists an $i$ such that $S_i'$ cannot be memorized by any transformer in $H_{P,P,P,cot}^q$.

*Proof.* Assume that for a pair of $P, q$, such that any $i \in \mathbb{Z}_+$, $S_i'$ can be memorized by $\mathcal{F} \in H_{P,P,P,cot}^q$, we can derive contradictions.

Firstly, if $S_i'$ can be memorized by an $\mathcal{F} \in H_{P,P,P,cot}^q$, we can show that the CoT created by $\mathcal{F}(x_1^i)$ is $(\gamma_4, \gamma_1, \gamma_3, \dots)$ for any $i \in \mathbb{Z}_+$.

Let the first symbol of the CoT created by $\mathcal{F}(x_1^i)$ be $\gamma_j$. If $j \in [3]$, then it holds $(x_1^i, \gamma_j) = x_{j+1}^i$, which implies $\mathcal{F}((x_1^i, \gamma_j)) = F(x_{j+1}^i)$, so that $\mathcal{F}$ will give the same label to $x_1^i$ and $x_{j+1}^i$. Because $x_1^i$ and $x_{j+1}^i$ have different lengths, based on the definition of LCP, they must have different labels. This is contradictory to $\mathcal{F}$ memorizes $S_i'$. So, we have proved that the first symbol of CoT created by $\mathcal{F}(x_1^i)$ is not $\gamma_1, \gamma_2, \gamma_3$. Thus, the first symbol of the CoT created by $\mathcal{F}(x_1^i)$ is $\gamma_4$. For the second and third symbols in CoT, in a similar way, we can show that the CoT created by $\mathcal{F}(x_1^i)$ is $(\gamma_4, \gamma_1, \gamma_3, \dots)$.

Secondly, by the definition of CoT-transformers and such CoT, we know that $\widehat{F}(x_1^i) = \gamma_4$ and $\widehat{F}((x_1^i, \gamma_4, \gamma_1)) = \gamma_3$. Moreover, because $x_1^i$ is a sentence with length $i$ and all elements in $x_1^i$ are $\gamma_1$, so by the Proposition 4.3, we have $\widehat{F}((x_1^i, \gamma_1)) = \widehat{F}(x_1^i) = \gamma_4$, which implies $\mathcal{F}$ can memorize $\{((x_1^i, \gamma_1), \gamma_4), ((x_1^i, \gamma_4, \gamma_1), \gamma_3)\}$ as a no-CoT-transformer.

Therefore, based on the assumption, we can deduce that for any $i$, the language $\{((x_1^i, \gamma_1), \gamma_4), ((x_1^i, \gamma_4, \gamma_1), \gamma_3)\}$ can be memorized by a $\mathcal{F} \in H_{P,P,P,cot}^q$. So, similar to the proof of no-CoT-transformers, we prove the result. □

### F.4 Proof of Corollary 6.1

Let $S_i$ be defined as in Section F.3.1, and $S_a = \cup_{i \in \mathbb{Z}_+} S_{2i} \subset \text{LCP}$. Let $S_i'$ be defined as in Section F.3.2, and $S_a' = \cup_{i \in \mathbb{Z}_+} S_{4i}' \subset \text{LCP}$. It is easy to check that $S_a(S_a')$ satisfies the conditions in Theorem 4.1(4.4).

*Proof.* **1. For no-CoT-transformer.** If a no-CoT-transformer $\mathcal{F}$ can memorize $S_a$, then we can prove that the result within two parts, which is similar to the proof of Theorem 5.4.

**Part One:** For any $x_2^{2i} \in S_a$, there exists an $\epsilon > 0$ such that $|\mathcal{F}(x_2^{2i})_1 - \mathcal{F}(x_2^{2i})_2| \geq \epsilon$ for any $i \in \mathbb{Z}_+$.

By Lemma 4.3, $|\mathcal{F}(x_2^{2i})_1 - \mathcal{F}(x_2^{2i})_2| = |F(\gamma_1)_1 - F(\gamma_1)_2|$ for any $i \in \mathbb{Z}_+$. If $F(\gamma_1)_1 = F(\gamma_1)_2$, then $\mathcal{F}$ cannot memorize $x_2^{2i}$, which is a contradiction with $\mathcal{F}$ can memorize $S_a$. So there exists an $\epsilon \in \mathbb{R}_+$ such that $\epsilon = |F(\gamma_1)_1 - F(\gamma_1)_2| = |\mathcal{F}(x_2^{2i})_1 - \mathcal{F}(x_2^{2i})_2|$ for any $i \in \mathbb{Z}_+$.

**Part Two:** We have $||\mathcal{F}(x_1^{2i}) - \mathcal{F}(x_2^{2i})||_1 \to 0$ when $i \to \infty$, which leads to our result. Consider that the value of parameters in $\mathcal{F}$ has an upper bound and a lower bound, so the proof of this part is similar to that in the proof of Theorem 5.4, and we omit it.

**2. For CoT-transformer.** If a CoT-transformer $\mathcal{F}$ can memorize $S_a'$, similar to that in the proof of Theorem 5.4, for any $x_1^{4i} \in S_a'$, we know that the CoT created by $\mathcal{F}(x_1^{4i})$ is $(\gamma_4, \gamma_1, \gamma_3, \dots)$ for any $i \in \mathbb{Z}_+$, which implies that $\mathcal{F}$ can memorize $\{((x_1^{4i}, \gamma_1), \gamma_4), ((x_1^{4i}, \gamma_4, \gamma_1), \gamma_3)\}$ for any $i \in \mathbb{Z}_+$ as a no-CoT-transformer. So, similar to before, we can obtain the result. $\square$

### F.5 Proof of Proposition 6.3

By Theorems 4.1 and 4.4, (1) in the proposition is apparent. So, we just need to find a sentence in $\mathbf{Arith}_p$ such that no-CoT- or CoT-transformers can solve it with confidence $c$.

Note that for $\mathbf{Arith}_p$, $\gamma_i = i$ when $i \in [p]$, $\gamma_{p+1}$ is the symbol $=$.

#### F.5.1 Proof of Proposition 6.3 for no-CoT transformers

*Proof.* Let $C_i$ be the sentence $'1 + 1 + 1 + \dots + 1'$ in $\mathbf{Arith}_{p,n}$, where there exist $i$ 1s in it. It is easy to see that $C_i$ has label $i \mod p$. We will show that, for any no-CoT-transformer $\mathcal{F}$, $\mathcal{F}$ cannot memorize some of these sentences with confidence $c$.

Let $\mathcal{F}$ be a no-CoT-transformer and $\mathcal{F}^l$ be the $l$-th hidden layer of $\mathcal{F}$.

**Firstly, we show that there exists a vector $C$ such that $||\mathcal{F}(C_i) - C||_2 \to 0$ when $i \to \infty$.**

Assume that the output of $\mathcal{F}^k$ when input $C_i$ to the transformer can be written as $l^k(i)$, and $l_j^k(i)$ means the $j$-th row of $l^k(i)$.

Assume that $\mathcal{F}^k$ can be written as

$$\mathcal{F}^1(x) = xw^k + \sum_{j=1}^{H} \text{softmax}(xQ_j^k V_j^k x^T + M)xK_j^k$$

$$+ \text{FNN}(xw^k + \sum_{j=1}^{H} \text{softmax}(xQ_j^k V_j^k x^T + M)xK_j^k).$$

We first consider $\mathcal{F}^1$. Let $v_+$ be the embedding vector of symbol $+$ and let $v_1$ be the embedding vector of symbol 1. We have that: $l_{2i-2}^1(i) = v_+ w^1 + \sum_{j=1}^{H} \frac{e^{s_j}v_+ + e^{t_j}v_1}{e^{s_j} + e^{t_j}}K_j^1 + \text{FNN}(v_+ w + \sum_{j=1}^{H} \frac{e^{s_j}v_+ + e^{t_j}v_1}{e^{s_j} + e^{t_j}}K_j^1)$, where $s_j = v_+ Q_j^1 V_j^1 v_+^T$ and $t_j = v_+ Q_j^1 V_j^1 v_1^T$. And we have $l_{2i-1}^1(i) = v_1 w + \sum_{j=1}^{H} \frac{(i-1)e^{s_j'}v_+ + ie^{t_j'}v_1}{(i-1)e^{s_i'} + ie^{t_i'}}K_j + \text{FNN}(v_1 w + \sum_{j=1}^{H} \frac{(i-1)e^{s_j'}v_+ + ie^{t_j'}v_1}{(i-1)e^{s_j'} + ie^{t_j'}}K_j)$, where $s_j' = v_1 Q_j^1 V_j^1 v_+^T$ and $t_j' = v_1 Q_j V_j v_1^T$.

It is easy to see that $l_{2i-2}^1(i)$ does not depend on $i$, and $l_{2i-1}^1(i)$ satisfies $l_{2i-1}^1(i) \to v_1 w + \sum \frac{e^{s_j'}v_+ + e^{t_j'}v_1}{e^{s_i'} + e^{t_i'}}K_j^1 + \text{FNN}(v_1 w + \sum \frac{e^{s_j'}v_+ + e^{t_j'}v_1}{e^{s_j'} + e^{t_j'}}K_j^1)$ when $i \to \infty$.

By the above result, we can see that the last two rows of $l^1(i)$ converge to a vector separately when $i \to \infty$. Then we will show that if $l_{2i-2}^{k-1}(i)$ and $l_{2i-1}^{k-1}(i)$ satisfy that converge to a vector separately when $i \to \infty$, it also stands for $l_{2i-2}^k(i)$ and $l_{2i-1}^k(i)$.

For $\mathcal{F}^k$, we have that: $l_{2i-2}^k(i) = l_{2i-2}^{k-1}(i)w^k + \sum_{j=1}^{H} \frac{\sum_{p=1}^{2i-2} e^{s_p^i(j)}l_p^{k-1}(i)}{\sum_{p=1}^{2i-2} e^{s_p^i(j)}}K_j^k + \text{FNN}(l_{2i-2}^{k-1}(i)w^k + \sum_{j=1}^{H} \frac{\sum_{p=1}^{2i-2} e^{s_p^i(j)}l_p^{k-1}(i)}{\sum_{p=1}^{2i-2} e^{s_p^i(j)}}K_j^k)$, where $s_p^i(j) = l_{2i-2}^{k-1}(i)Q_j^k V_j^k (l_p^{k-1}(i))^T$.

We will show that $l^k_{2i-2}(i)$ will converge to a vector, similar for $l^k_{2i-1}(i)$, so we can prove our conclusion.

Firstly, without loss of generality, assume that $\lim_{i\to\infty} l^{k-1}_{2i-2}(i) = \text{vec}_1$ and $\lim_{i\to\infty} l^{k-1}_{2i-1}(i) = \text{vec}_2$.

Secondly, by Lemma B.1, we have $l^{k-1}_p(i) = l^{k-1}_p([\frac{p+2}{2}])$ when $i \geq [\frac{p+2}{2}]$,

so $\sum_{j=1}^{H} \frac{\sum_{p=1}^{2i-2} e^{s^i_p(j)} l^{k-1}_p(i)}{\sum_{p=1}^{2i-2} e^{s^i_p(j)}} K^k_j = \sum_{j=1}^{H} \frac{\sum_{p=1}^{2i-2} e^{s^i_p(j)} l^{k-1}_p([\frac{p+2}{2}])}{\sum_{p=1}^{2i-2} e^{s^i_p(j)}} K^k_j$ and $s^i_p(j) = l^{k-1}_{2i-2}(i)Q^k_j V^k_j(l^{k-1}_p(i))^T = l^{k-1}_{2i-2}(i)Q^k_j V^k_j(l^{k-1}_p([\frac{p+2}{2}]))^T$.

Note that $s^i_{2p}(j) = l^{k-1}_{2i-2}(i)Q_j V_j(l^{k-1}_{2p}(p+1))^T$ tends to $t^j_1 = \text{vec}_1 Q_j V_j(\text{vec}_1)^T$ when $p, i \to \infty$ and $s^i_{2p+1}(j) = l^{k-1}_{2i-2}(i)Q_j V_j(l^{k-1}_{2p+1}(p+1))^T$ tends to $t^j_2 = \text{vec}_1 Q_j V_j(\text{vec}_2)^T$ when $i, p \to \infty$. Using

Lemma F.2, we have that $\lim_{i\to\infty} \frac{\sum_{p=1}^{2i-2} e^{s^i_p(j)} l^{k-1}_p([\frac{p+2}{2}])}{\sum_{p=1}^{2i-2} e^{s^i_p(j)}} = \frac{e^{t^j_1}\text{vec}_1 + e^{t^j_2}\text{vec}_2}{e^{t^j_1} + e^{t^j_2}}$. So $\lim_{i\to\infty} l^k_{2i-2}(i) =$

$\text{vec}_1 w^k + \sum_{j=1}^{H} \frac{e^{t^j_1}\text{vec}_1 + e^{t^j_2}\text{vec}_2}{e^{t^j_1} + e^{t^j_2}} K^k_j + \text{FNN}(\text{vec}_1 w^k + \sum_{j=1}^{H} \frac{e^{t^j_1}\text{vec}_1 + e^{t^j_2}\text{vec}_2}{e^{t^j_1} + e^{t^j_2}} K^k_j)$, and we get the result.

Finally, considering that the output of the transformer is only dependent on the the last row in the last hidden layer (i.e. $l^L_{2i-1}(i)$ when input $C_i$ to the transformer), which has a limitation when $i \to \infty$. So, we know that $\mathcal{F}(C_i)$ has a limitation when $i \to \infty$.

**Secondly, we prove the result.**

Let $C[k]$ be the $k$-th weight of $C$. Then if $C[k] > C[j]$ for some $k, j \in [p]$, then by the above result $\mathcal{F}(C_i) \to C$ when $i \to \infty$, there exists an $i_0$ such that for any $i > i_0$, it holds $(F(C_i))_k > (F(C_i))_j$. We take an $i_1$ such that $i_1 \mod p = j$ and $i_1 > i_0$. Then we have that $\mathcal{F}$ will not give $C_{i_1}$ the correct label $j$, because $(F(C_{i_1}))_k > (F(C_{i_1}))_j$, which implies $\mathcal{F}$ cannot memorize the $\mathbf{Arith}_{p,n}$.

If $C[1] = C[2] = C[3] = \ldots C[P]$, then there exists an $i_0$ such that when $i > i_0$, we have $|(F(C_i))_k - C[k]| < c/2$, which implies that $|(F(C_i))_k - (F(C_i))_j| < c$ for any $i > i_0$ and $k \neq j$, so $\mathcal{F}$ cannot memorize $\mathbf{Arith}_{p,n}$ with confidence $c$. $\qquad\square$

The above proof needs the following lemma.

**Lemma F.2.** *If a sequence of positive real numbers $\{x_i\}$ satisfies that $\lim_{p\to\infty} x_{2p} = a > 0$ and $\lim_{p\to\infty} x_{2p+1} = b > 0$, and a sequence of vectors $\{z_i\}$ satisfies that $\lim_{p\to\infty} z_{2p} = v_c$ and $\lim_{p\to\infty} z_{2p+1} = v_d$, then $\lim_{p\to\infty} \frac{\sum_{j=1}^{p} x_j z_j}{\sum_{j=1}^{p} x_j} = (av_c + bv_d)/(a+b)$.*

*Proof.* Assume that $i(\epsilon)$ satisfies that if $p > i(\epsilon)$, then $|x_{2p}-a| < \epsilon$, $|x_{2p+1}-b| < \epsilon$, $||y_{2p}-v_c||_2 < \epsilon$ and $||y_{2p+1} - v_d||_2 < \epsilon$.

Then we have that: $\frac{\sum_{j=1}^{p} x_j y_j}{\sum_{j=1}^{p} x_j} = \frac{\sum_{j=1}^{i(\epsilon)-1} x_j y_j}{\sum_{j=1}^{p} x_j} + \frac{\sum_{j=i(\epsilon)}^{p} x_j y_j}{\sum_{j=i(\epsilon)}^{p} x_j} \frac{\sum_{j=i(\epsilon)}^{p} x_j}{\sum_{j=1}^{p} x_j}$.

Firstly, it is easy to see that for any given $\epsilon$, when $p \to \infty$, we have $\frac{\sum_{j=1}^{i(\epsilon)-1} x_j y_j}{\sum_{j=1}^{p} x_j} \to 0$, because $a, b > 0$, and $\frac{\sum_{j=i(\epsilon)}^{p} x_j}{\sum_{j=1}^{p} x_j} \to 1$ also because the $a, b > 0$.

Secondly, we show that $\frac{\sum_{j=i(\epsilon)}^{p} x_j y_j}{\sum_{j=i(\epsilon)}^{p} x_j} \to (av_c + bv_d)/(a+b)$ when $\epsilon \to 0$ and $p - i(\epsilon) \to \infty$, which can directly prove our result.

We consider the $\frac{\sum_{j=i(\epsilon)}^{p} x_j y_j}{\sum_{j=i(\epsilon)}^{p} x_j} - \frac{v_c \sum_{j=i(\epsilon),j|2=0}^{p} x_j}{\sum_{j=i(\epsilon)}^{p} x_j} - \frac{v_d \sum_{j=i(\epsilon),j|2=1}^{p} x_j}{\sum_{j=i(\epsilon)}^{p} x_j}$ at first. We have that:

$$\left\| \frac{\sum_{j=i(\epsilon)}^{p} x_j y_j}{\sum_{j=i(\epsilon)}^{p} x_j} - \frac{v_c \sum_{j=i(\epsilon), j|2=0}^{p} x_j}{\sum_{j=i(\epsilon)}^{p} x_j} - \frac{v_d \sum_{j=i(\epsilon), j|2=1}^{p} x_j}{\sum_{j=i(\epsilon)}^{p} x_j} \right\|_2$$

$$= \left\| \frac{\sum_{j=i(\epsilon), j|2=0}^{p} x_j(y_j - v_c) + \sum_{j=i(\epsilon), j|2=1}^{p} x_j(y_j - v_d)}{\sum_{j=i(\epsilon)}^{p} x_j} \right\|_2$$

$$\leq \frac{\sum_{j=i(\epsilon), j|2=0}^{p} x_j \epsilon + \sum_{j=i(\epsilon), j|2=1}^{p} x_j \epsilon}{\sum_{j=i(\epsilon)}^{p} x_j}$$

$$= \epsilon.$$

So $\frac{\sum_{j=i(\epsilon)}^{p} x_j y_j}{\sum_{j=i(\epsilon)}^{p} x_j} \to \frac{v_c \sum_{j=i(\epsilon), j|2=0}^{p} x_j}{\sum_{j=i(\epsilon)}^{p} x_j} + \frac{v_d \sum_{j=i(\epsilon), j|2=1}^{p} x_j}{\sum_{j=i(\epsilon)}^{p} x_j}$ when $\epsilon \to 0$.

Hence, we show that we have $\frac{\sum_{j=i(\epsilon), j|2=0}^{p} x_j}{\sum_{j=i(\epsilon)}^{p} x_j} = \frac{a}{a+b}$ when $\epsilon \to 0$ and $p - i(\epsilon) \to \infty$.

Because $\frac{v_1^p(a-\epsilon)}{v_1^p(a+\epsilon) + v_2^p(b+\epsilon)} \leq \frac{\sum_{j=i(\epsilon), j|2=0}^{p} x_j}{\sum_{j=i(\epsilon)}^{p} x_j} \leq \frac{v_1^p(a+\epsilon)}{v_1^p(a-\epsilon) + v_2^p(b-\epsilon)}$, where $v_1^p(v_2^p)$ is the number of even (odd) in $[i(\epsilon), p]$. It is easy to see that when $\epsilon \to 0$ and $p - i(\epsilon) \to \infty$, we have $\frac{v_1^p(a-\epsilon)}{v_1^p(a+\epsilon) + v_2^p(b+\epsilon)} \to a/(a+b)$ and $\frac{v_1^p(a+\epsilon)}{v_1^p(a-\epsilon) + v_2^p(b-\epsilon)} \to a/(a+b)$. By the squeeze theorem, we get the result.

Similarly, $\frac{\sum_{j=i(\epsilon), j|2=1}^{p} x_j}{\sum_{j=i(\epsilon)}^{p} x_j} = \frac{b}{a+b}$ when $\epsilon \to 0$ and $p - i(\epsilon) \to \infty$.

Combining the above results, we have

$$\lim_{\epsilon \to 0, p-i(\epsilon) \to \infty} \frac{\sum_{j=i(\epsilon)}^{p} x_j y_j}{\sum_{j=i(\epsilon)}^{p} x_j}$$

$$= \lim_{\epsilon \to 0, p-i(\epsilon) \to \infty} \frac{v_c \sum_{j=i(\epsilon), j|2=0}^{p} x_j}{\sum_{j=i(\epsilon)}^{p} x_j} + \frac{v_d \sum_{j=i(\epsilon), j|2=1}^{p} x_j}{\sum_{j=i(\epsilon)}^{p} x_j}$$

$$= \frac{a v_c + b v_d}{a+b}$$

So we get the result. $\qquad\square$

### F.5.2 Proof of Proposition 6.3 for CoT-transformers

*Proof.* Let $C_i^j$ be the sentence $1 + 1 + 1 + \cdots + 1 + j$ in $\mathbf{Arith}_{p,n}$, where there exist $i$ 1s in it and $j \in [p]$. It is easy to see that, $C_i^j$ has label $(i+j) \mod p$. We will show that, for any CoT-transformer $\mathcal{F}$, $\mathcal{F}$ cannot give some of such sentences a correct CoT with confidence $c$.

Assume that $\mathcal{F}$ is a CoT-transformer and can solve the $\mathbf{Arith}_p$ with confidence $c$.

**First**, similar to the above subsection, we know that for such transformers $\mathcal{F}$ and $j$, when $i \to \infty$, it holds $\mathcal{F}(C_i^j) \to C(j)$ for a vector $C(j)$.

So, if $\mathcal{F}$ can solve $\mathbf{Arith}_p$ with confidence $c$, there must be $(C(j))[p+1] > (C(j))[t]$ for any $t \neq p+1$. If $(C(j))[p+1] \leq (C(j))[t]$ for some $t \neq p+1$, then the symbol $=$ will not appear at the start of the CoT with confidence $c$ for the $\mathcal{F}(C_i^j)$.

**Second**, we consider the CoT of $\mathcal{F}(C_i)$, where $C_i$ is defined in Section F.5.1.

From the definition of transformers, we know that if $\mathcal{F}$ can solve $\mathbf{Arith}_p$, then $\mathcal{F}(C_i)$ must output $=$ at the beginning of CoT. Now we consider the output of $\mathcal{F}((C_i, =))$. Similarly to the preceding subsection, we know that there exists a vector $C$ such that $\mathcal{F}((C_i, =)) \to C$ when $i \to \infty$.

Because $\mathcal{F}$ can solve $\mathbf{Arith}_p$ with confidence $c$, so as to meet the basic expression of the four arithmetic operations, the symbol $'='$ must be followed by a number in $[p]$, so we know there exists a $t \in [p]$ such that $C[t] > C[j]$ when $j \neq t$ and $j \in [p+7]$. If not, similar to the above section, we can show that $\mathcal{F}$ cannot get the correct CoT for $C_i$ with confidence $c$, which contradicts the assumption.

Then there exists an $i_0$ such that when $i > i_0$, $\widehat{\mathcal{F}}((C_i, =)) = t$.

**Third**, we consider $\mathcal{F}(C_i^t)$ and $\mathcal{F}((C_i, =, t))$. These are two sequences whose lengths are the same, which only differ at the penultimate symbols, one being $+$ and the other one being $=$. Then, because the parameters of $\mathcal{F}$ have the upper bound and lower bound, similar to that shown in Section F.3, there must be $\|\mathcal{F}((C_i^t)) - \mathcal{F}((C_i, =, t))\|_2 \to 0$ when $i \to \infty$. In part one, we showed that when

$i \to \infty$, we have $(\mathcal{F}(C_i^t))_{p+1} > (\mathcal{F}(C_i^t))_j$ for any $j \neq p+1$. So by the above results, there must be $(\mathcal{F}((C_i, =, t)))_{p+1} > (\mathcal{F}((C_i, =, t)))_j$ for any $j \neq p+1$ when $i \to \infty$, which implies that $\mathcal{F}((C_i, =, t))$ will output the symbol $=$ and create CoT $(=, t, =, \dots)$ for $C_i$.

Now we know that the CoT of $C_i$ must have the form $(=, t, =, \dots)$ when $i \to \infty$. If $i$ satisfies that $i$ mod $p \neq t$, then it is not a correct CoT to solve $C_i$, which is a contradiction to the fact that $\mathcal{F}$ can solve $\mathbf{Arith}_p$. So we prove the result. $\qquad\square$

## G  Impact Statement

We have theoretically demonstrated the power of CoT in the memorization capabilities of transformers. Our research does not include conclusions that have negative effects on the community.

