# OpenReview forum: "Analyzing the Power of Chain of Thought through Memorization Capabilities"
_NeurIPS.cc/2025/Conference — NeurIPS 2025 poster_

### Official Review · Reviewer_wb9P · 2025-07-01

**Clarity:** 3
**Significance:** 2
**Originality:** 3
**Rating:** 3
**Confidence:** 3

**Summary:**

This work studies whether CoT prompting can enhance the expressive power of Transformers in the context of memorization. Focusing on fixed-precision Transformers and finite datasets, the authors claim that CoT does not significantly improve a transformer's ability to memorize general datasets. They show that Transformers with CoT do not strictly dominate those without CoT in terms of memorization capacity. Additionally, the paper demonstrates that CoT does not enable Transformers to memorize certain infinite languages.

**Questions:**

See Weaknesses.

**Ethical Concerns:**

["NO or VERY MINOR ethics concerns only"]

**Final Justification:**

The authors' response partially addresses my concerns, but I still believe algorithmic structures or complexity classes are essential for properly analyzing CoT reasoning, even with the additional clarifications. Memorization, which considers the most general class, does not fully capture the reasoning aspects of CoT. Since this limitation remains unresolved, I maintain my original score.

**Quality:**

3

**Strengths And Weaknesses:**

Strengths
1. The paper presents an interesting and somewhat counter-intuitive result that CoT may not enhance the expressive power of Transformers from the perspective of memorization. This aspect seems to be largely overlooked in prior literature.
2. The theoretical analysis appears solid, covering both necessary and sufficient conditions as well as providing matching upper and lower bounds.

Weaknesses
1. In my understanding, CoT is primarily intended to to improve the capability of LLMs for reasoning tasks whose solving typically involves multiple algorithmic steps. From this perspective, the significance of analyzing CoT's impact on memorization is somewhat unclear.
2. The paper considers fixed-precision Transformers with flexible width and depth, in contrast to prior works that focus on log-precision, constant-depth Transformers when analyzing CoT’s effect on expressivity. In those earlier studies, constant depth is a critical constraint, where CoT serves to increase the model’s effective depth. While it is reasonable to explore a different setting, additional justification may be necessary to explain its relevance (e.g., better alignment with practical models) and to clarify how the differences in setup affect the interpretation and comparability of results with prior work.

---

> ### Author Rebuttal · Authors · 2025-07-29
>
> We thank the reviewer for acknowledging our theoretical contribution to the largely overlooked topic of expressive ability of CoT transformer in terms of memorization. We think that we have addressed the concerns raised in the Weakness clearly.
>
> **W1. In my understanding, CoT is primarily intended to to improve the capability of LLMs for reasoning tasks whose solving typically involves multiple algorithmic steps. From this perspective, the significance of analyzing CoT's impact on memorization is somewhat unclear.**
>
> **Answer:**
>
> We will explain the why studying Memorization CoT Transformer and, closely related, the relationship with previous Reasoning CoT from 3 aspects.
>
> 1, First, it should be noted that previous work on Reasoning CoT Transformer is also based on Memorization/Interpolation, since the accurate results are required for reasoning, so study the memorization/Interpolation is important for reasoning. We give precise description for the memorization ability for a LARGER class of problems than previous work on COT, the difference is: our Memorization CoT-Transformer interpolates data sampled from black-box algorithms (Line 301-302), while previous Reasoning CoT-Transformer can only to interpolate data obtained from KNOWN algorithms such as arithmetic problems. So, we consider a LARGER class of problems, because there is no KNOWN algorithms in most real-world situations like language translation. More precisely, previous CoT methods construct the transformer step by step to interpolate data according to the KNOWN algorithm to generate these data, such as give the results of an arithmetic expression step by step. While CoT memorization can be used to interpolate data obtained by black-box algorithms. The above information was given in Lines 31-36.
>
> 2, Whether CoT can enhance the transformer’s ability to solve back-box reasoning problems is apparently an important problem. Our work gives a complete answer to this problem.
>
> 3, We give the first result on interpolating infinite reasoning dataset by transformer, as shown in the “Section 5.3 Memorization of infinite language is hard”, while ALL previous work on CoT Reasoning focused on finite datasets as far as we know in.
>
> **W2. The paper considers fixed-precision Transformers with flexible width and depth, in contrast to prior works that focus on log-precision, constant-depth Transformers when analyzing CoT’s effect on expressivity. In those earlier studies, constant depth is a critical constraint, where CoT serves to increase the model’s effective depth. While it is reasonable to explore a different setting, additional justification may be necessary to explain its relevance (e.g., better alignment with practical models) and to clarify how the differences in setup affect the interpretation and comparability of results with prior work.**
>
> **Answer:**
>
> We answer this question from 3 aspects:
>
> 1. Fixed-precision Transformers are more close to practice, since semi-precision (FP16) is usually use to train models. Our results are more useful, especially when considering very long sectences or infinite dataset. Previous settings such as log-precision will fail in this case.
>
> 2. The reasons for using different structures are:
> Firstly, in the real world, due to the improvement of computing power, the depth of the LLM can be very large, which is more closed with our assumptions.
> Secondly, different problem is considered. As mentioned in answer to W1, we consider a more general problem than previous CoT works: we consider interpolating data sampled from black-box algorithms and previous work consider interpolating data obtained from KNOWN algorithms and the algorithm is used to construct the transformers. In our case, no algorithm is used for dataset, thus larger depth is needed. If we still consider shallow transformers, it will lead the conclusions that are not in line with reality.
>
> 3. The impact of our results on practice. We have studied a wider problem of CoT under a more realistic setting, and thus reached new conclusion: we give necessary and sufficient condition for CoT memorization and tight bound of number parameters of the memorization network. These conditions and bounds can be used to guide practitioners to train real models. For some infinte reasoning problem, we show that CoT is helpless.

---

> > ### Comment · Reviewer_wb9P · 2025-08-06
> >
> > Thank you for your detailed reply, which partially addresses my second question. However, I believe my first question has not been fully resolved.
> >
> > From my perspective, the essential distinction between reasoning and memorization lies in whether the question–answer mapping possesses some underlying algorithmic structure. In the context of reasoning, one typically considers a tractable complexity class (e.g., uniform $NC^0$ or $TC^0$) and analyzes whether all problems within that class can be solved by the model. In contrast, memorization involves instances that are "random" and lack such structure, corresponding to the most general complexity class.
> >
> > CoT is designed to enhance transformer models' ability to reason by encouraging algorithmic, step-by-step solutions to tractable problems. In this light, it seems somewhat unnatural to consider step-by-step memorization.
> >
> > The main concern is: Why should we assume the most general (potentially unstructured) complexity class when evaluating CoT, especially if this exceeds what we can reasonably expect for CoT? I would appreciate further clarification on this point, as I do not believe it has been fully addressed.

---

> > > ### Author Response · Authors · 2025-08-06
> > >
> > > Thank you for your reply. We apologize for not understanding the core of your previous question. Now, we will reply to your question again.
> > >
> > > 1: Why should we assume the most general (potentially unstructured) complexity class when evaluating CoT, especially if this exceeds what we can reasonably expect for CoT?
> > >
> > > The previous approach was indeed to analyze whether CoT could help solve a problem with specific algorithms (four budgets) or a Turing machine.
> > >
> > > But in reality, our expectations for CoT are not only to solve these simple problems. In actual reasoning tasks, we also use CoT to face more difficult problems, such as mathematical proofs (algebraic, geometric proofs, IMO, etc.), difficult computational problems (such as integral and limitation calculations), and so on. It is obvious that these problems do not exist with fixed algorithms to solve, even if such algorithms exist, they are extremely complex.
> > >
> > > However, limited exploration of issues such as P and NC0 is insufficient to study the help of CoT for these complex reasoning problems. So, to further explore the capabilities of CoT on reasoning, it is necessary to consider more complex issues, even 'potentially unstructured' ones.

---

> > > > ### Author Response · Authors · 2025-08-06
> > > > **More Answers**
> > > >
> > > > After carefully thinking your deep question, we add the following rebuttal on your questions and hope that we addressed your concerns more thoroughly.
> > > >
> > > > Comment 1. “From my perspective, the essential distinction between reasoning and memorization lies in whether the question–answer mapping possesses some underlying algorithmic structure.”
> > > >
> > > > Our memorization can be considered to interpolate finite dataset sampled from black-box algorithms (as mentioned in Line 301-302), while previous Reasoning CoT-Transformer can only to interpolate finite dataset obtained from KNOWN algorithms such as arithmetic problems. So, we consider a LARGER class of reasoning problems, because there is no KNOWN algorithms in most real-world tasks like language translation.
> > > >
> > > > Comment 2. “In the context of reasoning, one typically considers a tractable complexity class and analyzes whether all problems within that class can be solved by the model. In contrast, memorization involves instances that are "random" and lack such structure, corresponding to the most general complexity class.”
> > > >
> > > > Indeed, most existing work considered tractable complexity reasoning classes such as References [7,21] of our paper. On the other hand, a natural and essential question about CoT is:
> > > >
> > > > “Whether CoTs can enhance transformer’s ability of solving any reasoning problem”.
> > > >
> > > > Our paper give the answer to the above question. In particular, we demonstrated that there exist some reasoning tasks where CoT is equivalent to No-CoT, and CoT does not enhance the reasoning power of the transformer, as stated in Theorem 5.1. More precisely, from Theorems 4.1, 4.4, 5.1, \overline(N) is the necessary and sufficient number of parameters for a transformer to solve this reasoning task for both no-CoT and CoT transformers. Existence of such a reasoning task gives us new insights into the nature of CoT, that is, for some reasoning tasks CoT does not helps.
> > > >
> > > > Comment 3. “Why should we assume the most general (potentially unstructured) complexity class when evaluating CoT, especially if this exceeds what we can reasonably expect for CoT? I would appreciate further clarification on this point, as I do not believe it has been fully addressed.”
> > > >
> > > > In general, we believe that the reason is: “Many properties of CoT are still do not know and need to be explored.”
> > > >
> > > > Besides the questions we said in the our second Rebuttal and in Comment 2 above, we mention another unknown property of CoT:
> > > >
> > > > “Whether there exists a single CoT-trasnfomrer that can solve an infinite reasoning problem, like Arith_p.”
> > > > Also see “Conjecture5.8. Arith_p cannot be memorized by any fixed-precision CoT- or no-CoT-transformers” (line 404 ) of the paper.
> > > >
> > > > In the “Section 5.3 Memorization of infinite language is hard”, we give the first results on solving infinite reasoning problems by transformers, while ALL previous work on CoT Reasoning focused on finite datasets as far as we know. For instance, [7] showed that Arith_{p,n} can be solved by a CoT-transformer, where Arith_{p,n} is a finite set. In Proposition 5.7 of our paper, we show that Arith_p cannot be solved by any CoT-transformer with positive confidence, where Arith_p is an infinite set.

---

> > > ### Author Response · Authors · 2025-08-08
> > >
> > > We sincerely thank you for acknowledging the theoretical contribution of the paper. We also accept the limitations of our results raised by you. As the due time for author-reviewer is approaching, we want to summary the main contributions of the paper shortly for your considerations:
> > > 1. We give the lower bound and upper bound for CoT-Transformer and no CoT-Transformer to memorize N data sampled from general (black-box) reasoning tasks， which are both O(N). As a consequence, CoT does not enhance the reasoning power of transformers for certain reasoning tasks.
> > > 2. We give the FIRST result on solving infinite reasoning tasks by transformers, while ALL previous work on CoT Reasoning focused on finite datasets. In particular, we proved that arithmetic tasks in Z_p CANNOT be solved by any transformer with positive confidence (Proposition 5.7).

---

### Official Review · Reviewer_VN8V · 2025-07-03

**Clarity:** 3
**Significance:** 3
**Originality:** 3
**Rating:** 4
**Confidence:** 2

**Summary:**

This work studies whether CoT can enhance a model's memorization capability from a theoretical perspective. The authors first identified the necessary and sufficient conditions for Transformers to memorize a language. Then, the authors presented lower and upper bound on the number of model parameters required to memorize a given language with and without CoT. These bounds show that (1) CoT affects the languages that can be memorized by the Transformer (2) CoT does not increase the memorization capability of Transformers significantly.

**Questions:**

See weaknesses.

**Ethical Concerns:**

["NO or VERY MINOR ethics concerns only"]

**Final Justification:**

Given the concerns raised by other reviewers regarding the validity of the problem setup, I have lowered my rating to 4, as the impact of this work on understanding memorization in pre-trained Transformers might be limited. However, as someone who works on the empirical side of memorization, I do value theoretical work that tries to model memorization, even under toy settings.

**Limitations:**

Yes.

**Paper Formatting Concerns:**

N/A.

**Quality:**

3

**Strengths And Weaknesses:**

(Disclaimer: I do not work on formal language or learning theory, hence might not be the right person to judge the significance of this work)

**Strengths**
* This work studies an interesting problem: whether CoT enhance the memorization capacity of Transformers.
* This work includes a detailed analysis comparing memorization capacities of CoT and non-CoT Transformers under different language classes, position embeddings setup, etc.
* The paper is generally well written. The settings and assumptions are clearly stated in the paper.

**Weaknesses**
* This work studies one particular implementation of CoT Transformer (as defined on L154-158). Is it possible that another implementation of CoT Transformers could potentially lead to different memorization capabilities, especially for different languages?

---

> ### Author Rebuttal · Authors · 2025-07-29
>
> We thank the reviewer for acknowledging the study of an interesting problem and give a detailed analysis. We think that we have addressed the concerns raised in the Weakness clearly.
>
> **W1. This work studies one particular implementation of CoT Transformer (as defined on L154-158). Is it possible that another implementation of CoT Transformers could potentially lead to different memorization capabilities, especially for different languages?**
>
> **Answer:**
>
> The definition of CoT Transformer (L154-158) is the commonly used one in most papers and practical LLMs, which is also called autoregressive Transformer.
>
> Different types of CoT transformer require different theoretical support, and we believe that there are some other types of CoT transformer that can help improve the memorizaiton and reasoning capabilities of transformers.
>
> To study the memorization capability of other CoT Transformers will be listed as a future work in the paper.

---

> > ### Comment · Reviewer_VN8V · 2025-08-06
> >
> > Thanks for the explanation! I have read other reviewers comments and I understand that there are technical concerns regarding the validity of the setup and the significance of the proofs -- I would defer the judgement to reviewers who actually work on the theoretical aspect of Transformers. As someone who works on the empirical side of memorization, I generally would like to see more theory-grounded work on explaining Transformer's memorization behaviors, even in toy settings.

---

> > > ### Author Response · Authors · 2025-08-07
> > >
> > > Finally, thank you for reviewing our article and providing valuable comments.
> > >
> > > Best, author.

---

### Official Review · Reviewer_ZKA9 · 2025-07-03

**Clarity:** 2
**Significance:** 2
**Originality:** 3
**Rating:** 3
**Confidence:** 3

**Summary:**

This paper investigates the expressiveness of transformers without positional encoding, both with and without chain-of-thought (COT). It establishes a simplified theoretical framework, derives necessary and sufficient conditions for the languages expressible by these models, and compares their expressive power, illustrated through the LCP problem. Additionally, it provides bounds on parameter size estimates for valid languages. While the problem is of significant interest to the community, the paper is difficult to follow due to unclear notations and insufficient intuitive explanations of theorems.

**Questions:**

Major Problems:
1. What is the meaning of {K_k} in Theorem 4.1 and κ_j in Theorem 4.4? Is it a one-element set of κ_k or a k-element subset of κ?
2. How does a COT transformer memorize LCP^{=1}_{n}? By Theorem 4.1, a non-COT transformer only considers the set of distinct input symbols, not their positions or counts. For inputs x = A and x = A ···A, the COT process should be identical since they share the same typ() at each step, leading to the same output. How does the COT transformer produce different outputs based on input length?
3. Does Theorem 4.1 simply state that a non-COT transformer is a universal approximator for functions dependent only on typ(x), i.e., it can express all functions in UAT(typ(x))?
4. What mechanism allows a COT transformer to leverage input information beyond typ(x)?

Minor Problems:
1. The notation 2^κ, where κ is a set, is confusing. A more conventional notation would be κ^k, representing a length-k array with elements from κ.
2. What is the relationship between embedding dimension d and width W in Section 3?

These questions are adequately addressed by the authors in the responses.

**Ethical Concerns:**

["NO or VERY MINOR ethics concerns only"]

**Final Justification:**

My technical concerns have been adequately addressed. However, despite the authors' explanations, I still find the presentation overly abstract, lacking intuitive explanations or high-level proof sketches to enhance clarity. While the paper is technically sound, I cannot recommend it for acceptance.

**Limitations:**

yes

**Quality:**

2

**Strengths And Weaknesses:**

The paper addresses a compelling problem and provides theoretical insights into the expressiveness of transformers with and without COT. The analysis of expressive power differences and the application to the LCP problem are notable strengths. However, the paper suffers from insufficient explanations. Theorems lack intuitive interpretations, and key concepts are not adequately clarified, hindering readability.

---

> ### Author Rebuttal · Authors · 2025-07-29
>
> We thank the reviewer for acknowledge that our paper provides theoretical insights into the memorization expressiveness of transformers. The main concerns of the reviewer on our paper is about clarity. After carefully check the questions, we believe that the paper has formally and clearly defined these notions as to be explained below.
>
> Questions:
> Major Problems:
> **P1. What is the meaning of {K_k} in Theorem 4.1 and κ_j in Theorem 4.4? Is it a one-element set of κ_k or a k-element subset of κ?**
>
> **Answer:**  In Th4.1: '$typ (x1)=typ (x2)=\\{\kappa_k\\}$ for some $k \in [T]$' means that typ (x1)=typ (x2) and they all contain only one basic element. Here, {$\kappa_k$} is the set with a single element $\kappa_k$ but not a k-element subset of $\kappa$. Hence, $\kappa_k$ does not specifically refer to a certain basic element, but only refers to the basic element contained in typ (x1) and typ (x2). Please note that we mentioned 'for some $k \in[T]$' in Line 180, according the definition of $\kappa$ and typ(x) in the Line 95-97, the meaning of $\\{\kappa_k\\}$ is clear.
>
> In Th4.4：‘for any j ∈ [T] satisfying $\\{(x,y) \in S :typ(x) = \kappa_j\\}$ ≠ ∅‘ should be ‘for any j ∈ [T] satisfying $\\{(x,y) \in S :typ(x) = \\{\kappa_j\\}\\}$ ≠ ∅‘, we lose a '\{\}' here, but it does not affect the correct of theorem.
>
> Similarly, $\kappa_j$ is not a specific basic symbol but used to refer to any $j\in[T]$ that satisfies $\\{(x, y) ∈ S: typ (x)=\\{κ_j\\}\\}$ $\ne$∅. $\\{\kappa_j\\}$ is the set with a single element $\kappa_j$ but not a j-element subset of $\kappa$. Please note that we said 'for any j ∈ [T]' in Line 208. According the definition of $\kappa$ and typ(x) in the Line 95-97, the meaning of $\\{\kappa_j\\}$ is clear.
>
> **P2. How does a COT transformer memorize LCP^{=1}_{n}? By Theorem 4.1, a non-COT transformer only considers the set of distinct input symbols, not their positions or counts. For inputs x = A and x = A ···A, the COT process should be identical since they share the same typ() at each step, leading to the same output. How does the COT transformer produce different outputs based on input length?**
>
> **Answer:**
>
> 1. This result comes from Theorem 4.4 by verifying its hypothesis, as we did in the Proof of Proposition 4.7 in Lines 1057-1060.
>
> 2. We explain why a CoT transformer F can produce different results for x1=A and x2=AAAAAA.
>
> First by the definition of CoT transformer(line 154-157), we note that since each step of a CoT can be saw as once calculation process of no-CoT, we can use Prop 4.3 and Theorem 4.1 to each step of CoT.
>
> F(x1) and F(x2) are indeed the same in the first step of the CoT according to Prop 4.3, and let \widehat{F}(x1)=\widehat{F}(X2)=B.
>
> But, in the second step of the CoT, B will be attached to the ends of x1 and x2, which are the new inputs to F, that is, we will compute F(AB) and F(AAAAAAB). Now,  |typ(AB)|=|type(AAAAAAB)|=2. Since AB and AAAAAAB contain more than one element in $\kappa$, F can give different results for them according to Theorem 4.1.
>
> The same applies to every subsequent step, so CoT can solve the dataset with typ=1.
>
> **P3. Does Theorem 4.1 simply state that a non-COT transformer is a universal approximator for functions dependent only on typ(x), i.e., it can express all functions in UAT(typ(x))?**
>
> **Answer:**
> 1. Memorization means exactly equal to the value of the function. Then Theorem 4.1 is about exact interpolation of finite data sampled from black-box functions, NOT approximate the functions with a high accuracy. More precisely, Theorem 4.1 gives the necessary and sufficient conditions for using no-CoT transformers to exactly interpolate values of a black-box function on finite elements from $2^\kappa$.
> 2. So Theorem 4.1 is not about express functions over typ(x), but is about functions over $2^\kappa$. More precisely, about exact interpolation of finite data sampled from black-box functions over $2^\kappa$.
>
> **P4. What mechanism allows a COT transformer to leverage input information beyond typ(x)?**
>
> **Answer:** Beyond typ(x), a COT transformer can also use the autoregressive nature of CoT-Transformers to produce intermediate steps, which may result in different results than no CoT case. See our answer to P2.
>
> **P5. The notation 2^κ, where κ is a set, is confusing. A more conventional notation would be κ^k, representing a length-k array with elements from κ.**
>
> **Answer:** $2^\kappa$ is clearly defined in Line 96:  $2^\kappa$ is use to denote the set of all sequences with elements in kappa. What we want to clarify is that we did not impose any limit on the length of sentences throughout the paper, so we wanted to use a symbol to represent all sentences (regardless of length), which is why we chose an expression that is independent of length.
>
> **P6. What is the relationship between embedding dimension d and width W in Section 3?**
>
> **Answer:** In standard Transformers, d and W are the same. In some papers, W is less than d in order to use less parameters. In our paper, we relaxed this restriction so that they can be different. This more general assumption will not affect the rationality of the structure, as we introduced the situation where d and W are different in Line 130.

---

> > ### Comment · Reviewer_ZKA9 · 2025-08-05
> >
> > My technical concerns have been addressed, but I still have a few remaining questions:
> > 1. Memorization vs. Practical Performance of CoT: From the perspective of memorization capability, Chain-of-Thought (CoT) appears to offer only a limited advantage over non-CoT approaches. Yet, in practice, CoT consistently achieves strong performance on reasoning tasks. Does this work provide any insights into why CoT is so effective in practice despite this apparent gap?
> > 2. Intuition Behind Theorems 4.1 and 4.4: Could you provide a more intuitive explanation or a simple illustrative example for the proofs of Theorems 4.1 and 4.4? A high-level intuition or shortcut would be helpful for understanding the key ideas.

---

> > > ### Author Response · Authors · 2025-08-06
> > >
> > > Thank you for your reply. For the convenience of understanding, we will now use straightforward language to further explain your question. I hope it can solve your question.
> > >
> > > 1: Memorization vs. Practical Performance of CoT
> > >
> > > Regarding this issue, it can be answered by combining previous work:
> > >
> > > (1): There are indeed some tasks where CoT has advantages, such as the argument made in [7] of the References about the advantages of CoT in solving arithmetic problems. For these simple questions that can be solved through iterative algorithms, CoT can indeed demonstrate advantages, and these tasks are common in daily life, so CoT has demonstrated powerful capabilities in practice. Please note that this is not contradictory to our upper bound results(Th4.1 and 4.4), as our upper bound is demonstrated for any dataset rather than for specific tasks.
> > >
> > > (2): On the other hand, we demonstrate that there are indeed some tasks where CoT is equivalent to No-CoT ability, and CoT does not have an advantage, as stated in Theorem 5.1 and Prop3.7. It should be pointed out that these tasks are usually constructed and rare, so they are not common in daily life, but they do exist mathematically.
> > >
> > > 2. Intuition Behind Theorems 4.1 and 4.4
> > >
> > > The core of transformer's expressive power regarding this issue is to separate data with different labels through layer by layer in the transformer, thereby generating prediction results. As long as the data satisfies this property (can be separated layer by layer) under the given output method(CoT or no-CoT), the transformer can express it. However, some special data cannot satisfy this property and therefore cannot be expressed.

---

> > > > ### Comment · Reviewer_ZKA9 · 2025-08-08
> > > >
> > > > I would like to thank the authors for the careful clarifications. While I believe the paper is technically sound, I find the presentation overly abstract, with limited intuitive explanations or high-level proof sketches to aid understanding. As a result, while I cannot recommend acceptance, I am willing to raise my score to 3.

---

### Official Review · Reviewer_ueXx · 2025-07-05

**Clarity:** 2
**Significance:** 2
**Originality:** 3
**Rating:** 4
**Confidence:** 3

**Summary:**

CoT-enabled and standard fixed-precision autoregressive Transformers are formally compared, treating CoT as a scratch-pad workspace added at inference time.
Both models memorize any finite dataset with N parameters, though each obeys distinct necessary conditions; CoT does not lower the parameter count required for memorization.
Thus CoT alters which languages can be memorized but leaves raw memorization capacity unchanged, implying its gains come from reasoning structure rather than extra memory.

**Questions:**

Q1. Could you report empirical results on real LLMs to confirm the theoretical predictions?

Q2. Section 4 shows CoT and no-CoT memorize disjoint finite-language classes. Can you further characterize which natural NLP tasks map onto each class, or give examples where this distinction has concrete practical consequences?

Q3. Could you clarify what concrete practical or scientific insights are gained by measuring CoT’s memorization limits, and why those insights cannot be obtained more directly by studying CoT’s reasoning performance instead?
Given that Chain-of-Thought is primarily a reasoning aid, not a storage mechanism, the motivation for analysing its memorisation capacity feels under-justified.

Q4. As in Q3, your study focuses solely on whether finite datasets can be memorized, leaving untouched the core advantages of Chain-of-Thought—namely algorithmic reasoning difficulty and accuracy.
Could you justify why these reasoning metrics were excluded and provide evidence (theoretical or empirical) that the reported memorization bounds translate into meaningful gains—or at least no regressions—in actual CoT reasoning performance?

**Ethical Concerns:**

["NO or VERY MINOR ethics concerns only"]

**Final Justification:**

Since the authors addresses my concerns, I raise d the score.

**Limitations:**

Since no empirical experiments validate the theoretical results on real LLMs, their practical reliability remains untested.
The work considers only whether data can be memorized, ignoring CoT’s core strengths—algorithmic reasoning difficulty and accuracy.

**Paper Formatting Concerns:**

I didnot notice major formatting issues.

**Quality:**

2

**Strengths And Weaknesses:**

Strengths:
1. Gives exact necessary and sufficient conditions for memorization with and without CoT.
2. Proves tight N parameter bounds, showing CoT does not cut model size.
3. Identifies finite languages each model can memorize exclusively, clarifying expressive differences.

Weaknesses
1. Limits analysis to single-token labels, leaving sequence outputs unaddressed.
2. Omits positional encodings and LayerNorm, diverging from real-world Transformers.
3. Provides no empirical experiments, so theoretical claims are not validated in practice.

---

> ### Author Rebuttal · Authors · 2025-07-29
>
> We appreciate the reviewer's comments. We think that we have addressed the questions and concerns clearly.
>
> **W1. Limits analysis to single-token labels, leaving sequence outputs unaddressed.**
>
> **Answer:** You are correct on this which was also mentioned in lines 416-418. However, please note the following aspects: The single-token label case can also include many meaningful tasks, such as language recognition, topic classification and other single-token classification tasks; multiple choice questions such as mathematical multiple-choice questions, disease diagnosis choices, and color or place name recognition; single token code completion, single token mathematical computation, and other single token generation problems. We will add the above content in the revised paper. So, studying single-token memorization also has certain theoretical significance. Considering that our paper already has 41 pages, we list sequence memorization as a future research topic in line 417.
>
> **W2. Omits positional encodings and LayerNorm, diverging from real-world Transformers.**
>
> **Answer:** You are correct on this. However, please note the following aspects:
> 1. The effect of positional encodings on memorization was given. Lines 251-269 prove that “Position embedding is important for no-CoT transformer, but not for CoT-transformer”, which increases our understanding of positional encodings.
> 2. We include the main construction blocks of the Transformer. We acknowledge that analyzing the whole transformer is more important and difficult. Considering that this is the first paper on fixed precision CoT-Transformer memorization capability, we would like to thank the reviewer for understanding of this simplification.
> 3. Partial reasons why we do not consider positional encodings were given Remark3.3（Line 117）.
>
> **W3. Provides no empirical experiments, so theoretical claims are not validated in practice.
> and
> Q1. Could you report empirical results on real LLMs to confirm the theoretical predictions?**
>
> **Answer:**
>
> 1. We note that “learning theory” is a topic in the CFP and this paper belongs to this topic.
>
> 2. We think our main contribution is theoretical under practical background. We give necessary and sufficient condition for memorization with or no CoT, and tight bound of number parameters of the memorization network; further the more realistic fixed parameter precision model is considered for the first time. These conditions and bounds can be used to guide practitioners to train real models. Even the negative result that “CoT does not help memorization” is useful in practice: when solve problems obviously have no KNOWN algorithms, CoT will have no effect with high probability.
>
> 3. Due to time constraints, we cannot report experiment results.
>
> **Q2. Section 4 shows CoT and no-CoT memorize disjoint finite-language classes. Can you further characterize which natural NLP tasks map onto each class, or give examples where this distinction has concrete practical consequences?**
>
> **Answer:**
>
> 1. Firstly, We must clarify that we never say that CoT and no-CoT memorize disjoint finite-language classes. Proposition 4.7 shows that their scopes do not include each other (Line 232). We further said they are “both strong enough to memorize most languages” (Line 231).
> 2. Proposition 4.7 gives such examples: (2) of Prop 4.7 gives examples in the scope of Cot-Trans but not in that of no-CoT-Trans; (3) of Prop 4.7 gives examples in the scope of no-Cot-Trans but not in that of CoT-Trans;
>
>
>
> **Q3. Could you clarify what concrete practical or scientific insights are gained by measuring CoT’s memorization limits, and why those insights cannot be obtained more directly by studying CoT’s reasoning performance instead? Given that Chain-of-Thought is primarily a reasoning aid, not a storage mechanism, the motivation for analysing its memorisation capacity feels under-justified.**
>
> **Answer:** We thank the reviewer for raising this important topic about scientific insights and relation between previous works on reasoning CoT. Answers 1-3 below mainly address the scientific insights. Practical insights are addressed in Answer 4-6 for Q4.
>
> 1. First, it should be noted that previous work on Reasoning CoT Transformer is also based on Memorization/Interpolation, since the accurate results are required for reasoning, so study the memorization/Interpolation is important for reasoning.
> We give precise description for the memorization ability for a LARGER class of problems than previous work on COT, the difference is: our Memorization CoT-Transformer interpolates data sampled from black-box algorithms (Line 301-302), while previous Reasoning CoT-Transformer can only to interpolate data obtained from KNOWN algorithms such as arithmetic problems. So, we consider a LARGER class of problems, because there is no KNOWN algorithms in most real-world situations like language translation.
> More precisely, previous CoT methods construct the transformer step by step to interpolate data according to the KNOWN algorithm to generate these data, such as give the results of an arithmetic expression step by step. While CoT memorization can be used to interpolate data obtained by black-box algorithms. The above information was given in Lines 31-36.
> 2. Whether CoT can enhance the transformer’s ability to solve back-box reasoning problems is apparently an important problem. Our work gives a complete answer to this problem.
> 3. We give the first result on interpolating infinite reasoning dataset by transformer, as shown in the “Section 5.3 Memorization of infinite language is hard”, while ALL previous work on CoT Reasoning focused on finite datasets as far as we know in.
>
> **Q4. As in Q3, your study focuses solely on whether finite datasets can be memorized, leaving untouched the core advantages of Chain-of-Thought—namely algorithmic reasoning difficulty and accuracy. Could you justify why these reasoning metrics were excluded and provide evidence (theoretical or empirical) that the reported memorization bounds translate into meaningful gains—or at least no regressions—in actual CoT reasoning performance?
> Same concerns were raised in Limitions by the reviewer: Since no empirical experiments validate the theoretical results on real LLMs, their practical reliability remains untested. The work considers only whether data can be memorized, ignoring CoT’s core strengths—algorithmic reasoning difficulty and accuracy.**
>
> **Answer:** We continue our answers from Q3.
>
> 4. “your study focuses solely on whether finite datasets”.
>
> Actually, our paper seems to be first one on treating infinite dataset by transformer as far as we know. Section 5.3 of our paper “Memorization of infinite language is hard” considers infinite dataset, while ALL previous work on CoT Reasoning focused on finite datasets. See Lines 360-262.
>
> 5. “core advantages of CoT: algorithmic reasoning difficulty and accuracy”.
>
> First, accuracy seems not a problem, because both  our work and previous work which based on Memorization/Interpolation are accurately solved the problem. See Line 175-177 for definition. Our results are more useful in the sense, we consider fixed-precision parameters for the first time (Line 43-44).
> For algorithmic reasoning difficulty, our CoT Memorization measures the capability of CoT-Transformer to solve any language problem, while previous CoT applies to reasoning problems which are solved by KNOWN algorithms, as we know that there is no KNOWN algorithm “in most real-world situations like language processing”. For more, please refer to answer 1 for Q3.
>
> 6. “translate into meaningful gains” and practical implications.
>
> We give necessary and sufficient condition for memorization with or no CoT, and tight bound of number parameters of the memorization network; further the fixed parameter precision model is considered for the first time. These conditions and bounds can be used to guide practitioners to train real models. Even the negative result that “CoT does not help memorization” is useful in practice: when solve problems obviously have no KNOWN algorithms, CoT will have no effect with high probability.

---

### Note · Authors · 2025-08-12

We sincerely thank the reviewers for acknowledging our theoretical contributions. We also accept the limitations of our results such as lacking experimental results and clarity on the motivation.

We believe that this theoretical paper gives new and deep insights on the power of CoT-transformers:

1. We solved the problem: "Does CoT enhance the reasoning power of transformers for any reasoning task?"

We give the lower bound and upper bound for CoT-Transformer and no-CoT-Transformer to memorize N data sampled from general reasoning tasks，which are both overline(O)(N).  As a consequence, we proved that there indeed exist reasoning tasks for which CoT does not enhance the reasoning power of transformers.

2. We give the FIRST result on solving infinite reasoning tasks by transformers, while ALL previous work on CoT Reasoning focused on finite datasets. In particular, we proved that arithmetic tasks in Z_p CANNOT be solved by any CoT-transformer with positive confidence (Proposition 5.7).

3. We give the FIRST result on memorization for fixed-precision transformers, while all previous work focused on reasoning transformers whose parameter coefficients depend on the data or the model.

---

### Decision · Program_Chairs · 2025-09-17

**Decision:**

Accept (poster)

**Comment:**

**Summary of Claims and Findings**

This paper provides a theoretical analysis of the memorization capabilities of autoregressive transformers with parameters of fixed-precision, both with and without Chain of Thought (CoT). The authors establish necessary and sufficient conditions for these models to memorize a finite language and provide upper and lower bounds on the number of parameters required. Their central finding is that while the classes of languages memorizable by transformers with or without CoT are not subsets of each other, the number of parameters needed is asymptotically the same, specifically $\overline{O}(N)$ for a dataset of size $N$. This implies that CoT does not significantly enhance a transformer's raw memorization power. The paper also extends the analysis to infinite languages, arguing that CoT cannot help transformers memorize certain infinite arithmetic tasks.

**Strengths**

* **Novel Theoretical Contribution:** The paper addresses a novel and interesting theoretical question regarding the power of CoT from the perspective of memorization, an area that has been largely overlooked. The positive memorization results could be even stronger if considering practical usage of transformers, e.g., if a special [BOS] (beginning of sentence) token is added in the beginning, then the Non-CoT construction could memorize any dataset.

* **Rigorous Analysis:** The authors provide a solid theoretical framework, including necessary and sufficient conditions for memorization and tight bounds on the required number of parameters.

**(c) Weaknesses**

* **Simplifications:** The analysis omits key components of modern transformers like LayerNorm, positional encoding (in most parts of the paper) and focuses on single-token outputs, limiting its direct applicability. In particular, layerNorm could break some results such as Proposition 5.7(2) because Layernorm is non-lipschitz when input is around the zero and the current analysis seems to rely on the lipschitzness of the transformer layers.

Though this paper focuses on finite precision transformer, in the forward pass the transformer actually have infinite precision. This makes sense as intuitively the description length of transformer (measured by total number of bits used in parameters) upper bounds its memorization capability. However this also leads to questions about the practical relevance of the theoretical results presented in this paper, given the that the transformer forward pass are also done with finite precision and rounding.

* **Lack of Empirical Validation:** The submission is purely theoretical and lacks experiments on real-world models, making it difficult to assess the practical implications of its findings.

**Reasons for Recommendation**

This is a borderline paper, but its strengths ultimately outweigh its weaknesses. The primary reason for acceptance is the novelty and significance of the theoretical question it tackles. By formally analyzing CoT's impact on memorization, the paper provides a new and valuable lens through which to understand the mechanisms of large language models. While the practical implications are not yet clear and the theoretical model is simplified, the core findings that CoT alters *what* can be memorized but not *how much* and the analysis is a solid contribution to the theory of transformers.